# YTHDF2 regulates self non-coding RNA metabolism to control inflammation and tumorigenesis

Seungwon Yang [1,7] ✉, Yan-Hong Cui [1,7], Haixia Li[1,7], Jiangbo Wei [2,7], Gayoung Park[1], Ming Sun[1], Michelle Verghese[1], Emma Wilkinson[1], Teresa Nam[3], Linnea Louise Lungstrom [3], Xiaolong Cui[2], Tae Young Ryu[1], Jing Chen[4], Marc Bissonnette [5], Chuan He [2,6] & Yu-Ying He [1] ✉

The role of m⁶A RNA methylation of self non-coding RNA remains poorly understood. Here we show that m⁶A-methylated self U6 snRNA is recognized by YTHDF2 to reduce its stability and prevent its binding to Toll-like receptor 3 (TLR3), leading to decreased inflammatory responses in human and mouse cells and mouse models. At the molecular level, endosomal U6 snRNA binds to the LRR21 domain in TLR3, independent of m⁶A methylation, to activate inflammatory gene expression, a mechanism that is distinct from that of the best known synthetic TLR3 agonist poly I:C. Both U6 snRNA and YTHDF2 are localized to endosomes via the transmembrane protein SIDT2, where YTHDF2 functions to prevent the U6-TLR3 interaction. We further show that UVB exposure inhibits YTHDF2 by inducing its dephosphorylation and autophagic protein degradation in human keratinocytes and mouse skin. Skin-specific deletion of *Ythdf2* in mice enhanced the UVB-induced skin inflammatory response and promoted tumor initiation. Taken together, our findings demonstrate that YTHDF2 plays a crucial role in controlling inflammation by inhibiting m⁶A U6-mediated TLR3 activation, suggesting that YTHDF2 and m⁶A U6 are potential therapeutic targets for preventing and treating inflammation and tumorigenesis.

$N^6$-methyladenosine (m⁶A) RNA methylation is the most prevalent internal modification that occurs in messenger RNA (mRNA) and long non-coding RNA (lncRNA) of most eukaryotes[1–4]. m⁶A mRNA methylation regulates several aspects of RNA metabolism, including RNA decay, nuclear processing, translation, transcription, and RNA-protein interactions[1,3–7]. At the molecular level, m⁶A RNA modification is installed by writer complexes composed of factors including METTL3, METTL14, WTAP, and KIAA1429, or METTL16, and is removed by erasers FTO or ALKBH5[1]. m⁶A is recognized by m⁶A-binding proteins including YTHDF1-3, YTHDC1, YTHDC1-2, and IGF2BP1-3, also known as m⁶A readers, to regulate RNA fate[1,8]. Among the m⁶A writers, METTL16 is monomeric, and is distinct from METTL3/METTL14, which is an obligate heterodimer[9]. The METTL16 ortholog *mett-10* in *C. elegans* has been shown to deposit m⁶A on SAM synthase to inhibit its proper splicing[10]. Recent studies have demonstrated critical roles for METTL16 in the pathogenesis of leukemia and liver cancer in both

[1]Department of Medicine, Section of Dermatology, University of Chicago, Chicago, IL, USA. [2]Departments of Chemistry, Department of Biochemistry and Molecular Biology, Institute for Biophysical Dynamics, University of Chicago, Chicago, IL, USA. [3]The College, University of Chicago, Chicago, IL, USA. [4]Department of Medicine, Section of Hematology and Oncology, University of Chicago, Chicago, IL, USA. [5]Department of Medicine, Section of Gastroenterology, Hepatology & Nutrition, University of Chicago, Chicago, IL, USA. [6]Howard Hughes Medical Institute, University of Chicago, Chicago, IL, USA. [7]These authors contributed equally: Seungwon Yang, Yan-Hong Cui, Haixia Li, Jiangbo Wei. ✉e-mail: syang80research@gmail.com; yyhe@uchicago.edu

m[6]A-dependent and -independent mechanisms[11,12]. However, the function of METTL16 remains incompletely understood.

One m[6]A-modified non-coding RNA is the small nuclear RNA (snRNA) U6[13]. U6 snRNA is a non-coding RNA best known for its role in splicing. *U6* interacts with three snRNAs, pre-mRNA substrates, and more than 25 protein partners to form the catalytic core of the spliceosome during splicing[13]. Although commonly used as an internal standard to quantify the level of miRNA, U6 snRNA was recently recognized as a highly variably expressed gene in various human tissues including carcinoma tissues[14]. Notably, U6 snRNA levels are significantly higher in human carcinoma tissue than in the corresponding normal tissue[14,15]. Newly synthesized U6 appears transiently in the cytoplasm and undergoes maturation where it is accompanied by U6-associated proteins known as small nuclear ribonucleoprotein complexes, snRNPs, before returning to the nucleus[13,16]. Among the U6 snRNPs, loss of LSM6 or LSM7 induces a cytosolic accumulation of U6 snRNA[17]. Recently, U6 has been shown to be m[6]A-modified by METTL16 at A43[18,19]. In *Schizosaccharomyces pombe*, loss of the *Mettl16* ortholog *Mtl16* alters global splicing[20]. In contrast, in mouse embryos, *Mettl16* deletion had little effect on global splicing[21]. These contrasting functions suggest species-specific roles for METTL16 in splicing. Altogether, the functional role of m[6]A methylation on U6 snRNA in mammals as well as its reader remains unknown.

Emerging evidence has demonstrated that inflammation, originally recognized for its pivotal role in pathogen defense, also plays critical roles in a number of diseases including cancer[22] and autoimmune diseases[23], both of which can be induced or triggered by environmental factors such as UV radiation[24,25]. However, the molecular mechanisms that regulate inflammation remain incompletely understood. Here we show that YTHDF2 recognizes m[6]A-modified U6 snRNA to regulate U6 stability and binds to TLR3 in the context of inflammation and tumorigenesis, highlighting the crucial role of YTHDF2 and U6 m[6]A methylation in controlling inflammation.

## Results

### YTHDF2 controls inflammatory gene expression

Recently, we have shown that the m[6]A reader YTHDF1 regulates the repair of genome damage caused by UVB stress[26]. However, the role of other m[6]A readers, including YTHDF2, in stress responses remains poorly understood. To determine the functional role of YTHDF2, we performed RNA-seq in HaCaT cells, non-tumorigenic human keratinocytes, to identify pathways affected by YTHDF2 loss. Our analysis showed that YTHDF2 knockdown upregulates genes in several pathways, including the TNF and IL-17 signaling pathway, signaling by interleukins, TLR3 cascade, and antiviral pathways (Fig. 1A, Supplementary Fig. S1A), suggesting that YTHDF2 may act as a regulator of inflammation. Next, to determine whether YTHDF2 has a role in the UVB stress response, we performed mass spectrometric analysis to identify pathways associated with YTHDF2-interacting proteins in HaCaT cells (Supplementary Table S1). We compared these pathways with pathways of genes upregulated by UVB irradiation from our previous work in HaCaT cells (GSE145924)[26]. Analyzing both data sets, we found that genes up-regulated by UVB share several pathways with YTHDF2-interacting proteins, including metabolism of RNA and ribonucleoprotein complex biogenesis (Fig. 1B, Supplementary Fig. S1B), suggesting that YTHDF2 may play an important role in the UVB-induced stress response.

UVB damage induces inflammation, which clinically presents as sunburn. To determine whether YTHDF2 regulates UVB-induced inflammation in vivo, we generated a mouse model with skin-specific conditional knockout of *Ythdf2* (DF2 cKO) and assessed differences in UVB-induced histological alteration between wild-type (WT) and DF2 cKO mice. We noted that skin-specific YTHDF2 deletion increased epidermal thickness upon sham or UVB irradiation (Fig. 1C, D), suggesting that YTHDF2 loss enhances UVB-induced inflammation. Furthermore, flow cytometric analysis showed that skin-specific *Ythdf2* deletion specifically increased the number of CD45[+] cells and TCRγδ[+] T cells following UVB irradiation in the skin (Fig. 1E, Supplementary Fig. S2A, B); however, it did not significantly alter the number of other immune cells analyzed (Supplementary Fig. S2B–D). Next, we analyzed the effect of skin-specific *Ythdf2* deletion on the systemic immune system in the spleen, blood, and lymph node (LN). Skin-specific *Ythdf2* deletion did not significantly alter spleen weight or immune cell counts in the spleen or LN (Supplementary Fig. S3A–C). It is worthwhile to note that skin-specific *Ythdf2* deletion increased the number of CD11b[+] myeloid cells in both the blood and spleen, PMN-MDSCs and Ly6C high cells in the blood, and M-MDSC cells in the spleen, while it decreased Ly6C low cells in the blood (Supplementary Fig. S4A–C). These findings indicate that skin *Ythdf2* deletion augments skin inflammation and likely alters the systemic inflammatory response following UVB irradiation.

Given the role of YTHDF2 in modulating skin and systemic inflammation, we hypothesized that its expression may be altered in autoimmune disorders, where inflammation plays a central role in disease progression[23]. Indeed, *YTHDF2* expression was decreased in systemic lupus erythematosus (SLE) and type I diabetes as compared with healthy controls (Fig. 1F, Supplementary Fig. S5A). In contrast, we did not observe consistent changes in other autoimmune diseases (Supplementary Fig. S5B–D). These findings implicate that YTHDF2 may play an important, yet selective, role in autoimmune diseases, likely associated with its role in inflammation. Further investigations are warranted to define the role of YTHDF2 in the pathogenesis of SLE and type I diabetes.

Prompted by the observed role of YTHDF2 in skin inflammation in vivo, we hypothesized that keratinocyte-intrinsic YTHDF2 controls inflammatory gene expression and suppresses UVB-induced inflammation. Indeed, we found that knockdown of *YTHDF2* increased the expression of inflammatory mediators such as *TNF-α*, *IL-6*, *COX-2/PTGS2*, and *IL-1β* in both non-tumorigenic human HaCaT keratinocytes and normal human epidermal keratinocytes (NHEK) following UVB irradiation; it also increased the baseline expression of multiple inflammatory genes in A431 skin cancer cells (Fig. 1G–Q and Supplementary Fig. S6A–C). Furthermore, we found that skin-specific *Ythdf2* deletion increased COX-2 expression upon sham or UVB irradiation (Supplementary Fig. S6D), supporting our findings in human cells that YTHDF2 loss enhances UVB-induced inflammatory gene expression. Conversely, forced overexpression of YTHDF2 inhibited UVB-induced expression of these inflammatory mediators (Fig. 1R–V). As an mRNA m[6]A reader, YTHDF2 destabilizes m[6]A-containing mRNAs[27]. Intriguingly, although *YTHDF2* knockdown increased mRNA levels of pro-inflammatory genes, it had no effect on the mRNA stability of *IL-8, TGFβ, VEGF, COX-2, GM-CSF, IL-6, IL-1α,* or *IL-1β*, while it decreased the mRNA stability of *TNF-α* and *IL-16* in HaCaT cells (Supplementary Fig. S6E). These findings suggest that YTHDF2 loss may not directly inhibit mRNA decay of pro-inflammatory genes, but rather may indirectly enhance pro-inflammatory gene expression by activating upstream pathways that promote the expression of these genes.

### YTHDF2 binds to m[6]A-methylated self U6 snRNA to induce U6 decay

To determine the molecular mechanism by which YTHDF2 regulates pro-inflammatory gene expression, we analyzed mRNA m[6]A modification levels in polyadenylated RNAs using ultra-high-performance liquid chromatography-tandem mass spectrometry (UHPLC-MS/MS). Indeed, loss of *YTHDF2* increased mRNA m[6]A enrichment in HaCaT cells (Fig. 2A). Next, we performed m[6]A-seq to identify potential transcriptome-wide m[6]A-modified gene targets for YTHDF2. *YTHDF2* knockdown had little effect on m[6]A enrichment in the 5′UTR, 3′UTR, CDS, or the total peak distribution in HaCaT cells (Fig. 2B, C). Sequence motif analysis of m[6]A peaks showed enrichment for the previously

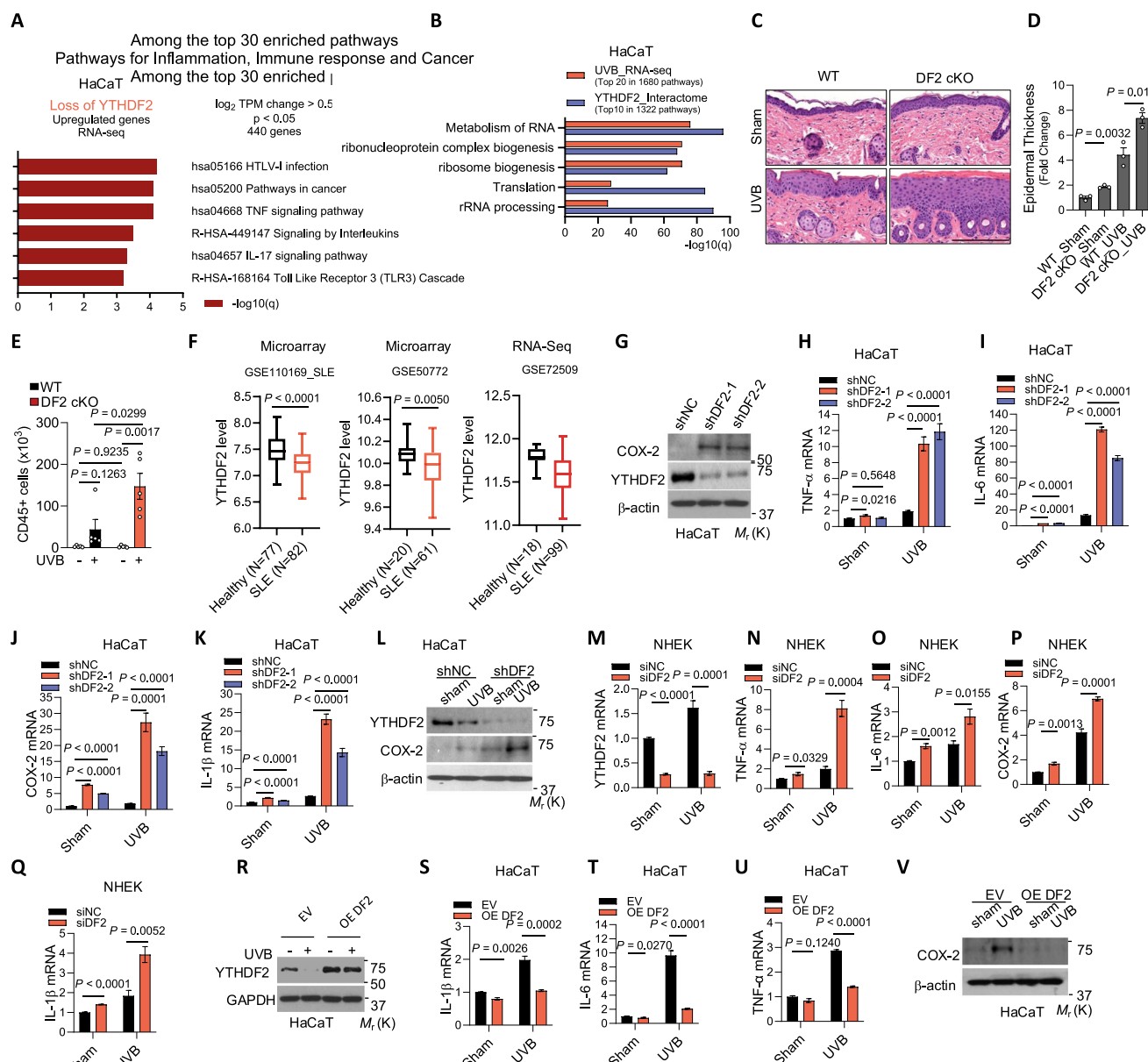

**Fig. 1 | YTHDF2 controls inflammatory gene expression. A** Pathway enrichment of genes upregulated upon YTHDF2 knockdown in HaCaT cells (RNA-seq, $q < 0.05$, DESeq2). **B** Overlap between the top 20 UVB-induced pathways (RNA-seq) and the top 10 pathways of YTHDF2-interacting proteins (mass spectrometry, 1 h post-sham or -UVB). **C** Representative H&E staining of skin sections 24 h after sham or UVB exposure in WT (YTHDF2[flox/flox]) and DF2 cKO (K14Cre; YTHDF2[flox/flox]) mice. Scale bar, 200 µm. $n = 3$ mice per group. **D** Quantification of epidermal thickness ($n = 3$). **E** CD45[+] cell counts in skin of sham- and UVB-irradiated WT and DF2 cKO mice ($n = 5$). **F** Box plots of YTHDF2 expression across systemic lupus erythematosus (SLE) datasets. Centre line, median; box, interquartile range; whiskers, minimum to maximum. Statistical analyses were conducted using a two-tailed unpaired Student's $t$-test, with $P$ values indicated (ns, $P > 0.05$). **G** Immunoblot of COX-2 and YTHDF2 in HaCaT cells with or without YTHDF2 knockdown. **H–K** qPCR analysis of TNF-α, IL-6, COX-2 and IL-1β mRNAs in HaCaT cells transduced with shNC (short hairpin RNA negative control), shDF2-1 or shDF2-2 (short hairpin RNAs targeting YTHDF2), with or without UVB. **L.** Immunoblot analysis of YTHDF2 and COX-2 in HaCaT cells with or without YTHDF2 knockdown, with or without UVB. **M–Q** qPCR analysis of YTHDF2, TNF-α, IL-6, COX-2 and IL-1β in NHEK cells transfected with siNC (small interfering RNA negative control) or siDF2 (siRNA targeting YTHDF2), with or without UVB. **R** Immunoblot of YTHDF2 in HaCaT cells expressing EV (empty vector) or OE DF2 (YTHDF2 overexpression), with or without UVB. **S–U.** qPCR analysis of IL-1β, IL-6 and TNF-α in cells as in R. **V** Immunoblot of COX-2 in cells as in R. β-actin served as internal control for qPCR (**H–K**, **M–Q**, **S–U**). Statistical analyses were conducted using a two-tailed unpaired Student's $t$-test, with $P$ values indicated (ns, $P > 0.05$). Data are shown as mean ± SE ($n = 5$ for **E**; $n = 4$ for **H–K**, **M–Q**, **S–U**) or mean ± SD ($n = 3$ for D). All experiments were conducted using biologically independent samples.

identified m[6]A target site motif (GGACU)[28] (Fig. 2D). Overall, the levels of over 3800 genes were changed in cells with knockdown of *YTHDF2* (Supplementary Fig. 7A). The downstream targets of YTHDF2 were selected following 3 criteria in YTHDF2 knockdown cells as compared with control cells: (1) increased RNA level, (2) increased m[6]A enrichment, and (3) related to inflammation or tumorigenesis. From this analysis, we identified the top 30 upregulated genes that are potential

m[6]A targets of YTHDF2 (Supplementary Fig. S7B), consistent with YTHDF2's known role in promoting decay of m[6]A-modified mRNA. Finally, we selected and verified four genes – *FOS*[29], *JUN*[22], *SOX4*[30], and *SOX9*[30] – as potential YTHDF2 targets, based on increased mRNA levels, stability, and increased m[6]A enrichment upon YTHDF2 knockdown, and as well as YTHDF2 binding to the mRNAs in HaCaT cells (Supplementary Fig. S7C–F, S8A, B). These findings indicate that *FOS*,

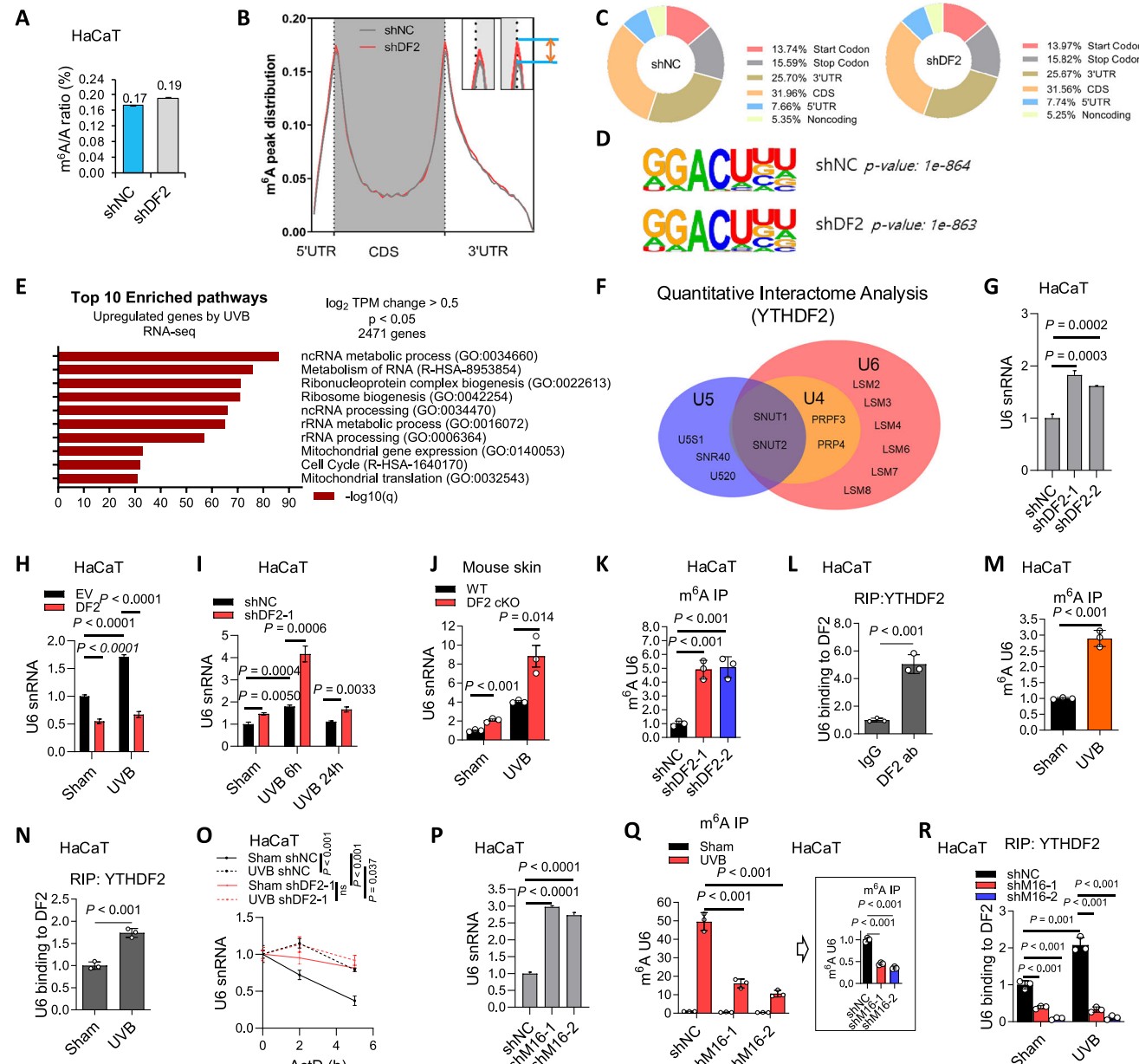

**Fig. 2 | YTHDF2 binds to m⁶A-methylated self U6 snRNA to induce U6 decay.**
**A** UHPLC-MS/MS for m⁶A enrichment in mRNA in HaCaT cells with or without YTHDF2 knockdown ($n = 2$). No statistical analysis was performed. **B** Distribution of m⁶A peaks across 5′UTR, CDS and 3′UTR. **C** Proportion of m⁶A peaks in the indicated regions across all mRNAs. **D** Consensus m⁶A motif identified by HOMER. **E** Pathway analysis of UVB-induced upregulated genes in HaCaT cells (q < 0.05, DESeq2). **F** Identification of U5-, U4- and U6-associated RNPs among YTHDF2-interacting proteins by mass spectrometry in HaCaT cells. **G–I** qPCR of U6 snRNA in HaCaT cells with or without YTHDF2 knockdown, with or without YTHDF2 over-expression, and with or without UVB irradiation. **J** qPCR of U6 snRNA in mouse skin with or without YTHDF2 deletion after sham or chronic UVB for 23 weeks ($n = 3$ mice). **K** m⁶A-IP qPCR for enrichment of U6 snRNA in HaCaT cells with or without YTHDF2 knockdown. **L** RIP of YTHDF2–U6 snRNA interaction in HaCaT cells.

**M** m⁶A-IP qPCR of U6 snRNA in HaCaT cells with or without UVB (30 min). **N** RIP of YTHDF2–U6 snRNA interaction in HaCaT cells with or without UVB (30 min). **O** qPCR of U6 snRNA stability in HaCaT cells with or without YTHDF2 knockdown, treated with actinomycin D (ActD), with or without UVB (4 h time course). **P** qPCR of U6 snRNA in HaCaT cells with METTL16 knockdown (shNC, shM16-1, shM16-2). **Q** m⁶A-IP qPCR of U6 snRNA in HaCaT cells with or without METTL16 knockdown, with or without UVB (30 min). **R** RIP of YTHDF2–U6 snRNA interaction in HaCaT cells with or without METTL16 knockdown, with or without UVB (30 min). 18S rRNA served as internal control (**G–J**, **O–P**). Statistical analyses were performed using a two-tailed unpaired Student's t-test. Data are presented as mean ± SE ($n = 4$ for **G–I**, **O–P**) or mean ± SD ($n = 2$ for **A**; $n = 3$ for **J–N**, **Q–R**). All experiments were conducted using biologically independent samples.

*JUN*, *SOX4*, and *SOX9* are downstream m⁶A targets for YTHDF2. Further investigation is needed to elucidate the functional importance of these genes in YTHDF's function. Interestingly, *YTHDF2* knockdown had little effect on the m⁶A enrichment across the transcripts for *TNF-α* and *IL-1β*, while it increased the m⁶A enrichment across the transcripts for *IL-6* and *COX-2* in HaCaT cells (Supplementary Fig. S9A–D). Since YTHDF2 knockdown had no effect on the mRNA stability of COX-2 or

IL-6 (Supplementary Fig. S6E), future investigation is warranted to elucidate the role of m⁶A methylation of *COX-2* and *IL-6* in their gene expression, including but not limited to translation.

As we did not observe canonical inflammatory genes in the candidate target list for YTHDF2 by m⁶A-seq, we elected to re-examine the pathways for UVB-induced genes from our previous RNA-seq data (GSE145924)[26]. Indeed, UVB irradiation induced the expression of

genes related to (non-coding RNA) ncRNA metabolic processes and ncRNA processing in HaCaT cells (Fig. 2E), suggesting a role for ncRNA in YTHDF2 regulation of the UVB stress response. In addition, our mass spectrometric analysis of YTHDF2-interacting proteins showed that YTHDF2 binds to several U6 snRNA-associated small nuclear ribonucleoproteins (U6 snRNPs) such as U6 snRNA-associated Sm-like protein LSM4, LSM6, U4/U6 small nuclear ribonucleoprotein Prp4 (PRPF4), U4/U6 small nuclear ribonucleoprotein Prp3 (PRPF3), and U4/U6.U5 tri-snRNP-associated protein 1/2 (SNUT1/2), as well as several U5-associated ribonucleoproteins in HaCaT cells (Fig. 2F, Supplementary Fig. S10A). These findings led us to hypothesize that YTHDF2 acts as a reader for m⁶A-methylated self U6 snRNA.

Indeed, we found that *YTHDF2* knockdown increased U6 snRNA levels in HaCaT cells (Fig. 2G). Similarly, UVB irradiation also increased U6 levels in HaCaT cells by qPCR and Fluorescence In-Situ Hybridization (FISH) analysis (Fig. 2H-I, Supplementary Fig. S10B). Both baseline and UVB-induced U6 levels were decreased by YTHDF2 overexpression, while they were further increased by YTHDF2 knockdown in HaCaT cells (Fig. 2H-I). Skin-specific *Ythdf2* deletion in mice increased U6 levels in the skin, which were further increased by UVB irradiation (Fig. 2J). Moreover, *YTHDF2* knockdown increased m⁶A enrichment in U6 snRNA in HaCaT cells and primary mouse keratinocytes (Fig. 2K, Supplementary Fig. S10C). Furthermore, using RNA immunoprecipitation (RIP) analysis, we found that YTHDF2 binds to U6 snRNA in HaCaT cells (Fig. 2L). At 30 min in HaCaT cells, UVB irradiation increased m⁶A enrichment in U6 snRNA (Fig. 2M) and enhanced the interaction of U6 with YTHDF2 (Fig. 2N). We then hypothesized that YTHDF2 regulates U6 stability. Indeed, both *YTHDF2* knockdown and UVB stress increased U6 snRNA stability in HaCaT cells (Fig. 2O). Notably, the combination of UVB irradiation and *YTHDF2* knockdown had a similar effect on U6 stability as either UVB irradiation or YTHDF2 knockdown alone (Fig. 2O), suggesting that UVB irradiation increases U6 stability through inhibiting YTHDF2. Notably, the half-life of U6 in control cells is approximately 4 h, substantially shorter than the ~24 h half-life reported previously[31,32]. It is possible that actinomycin D (ActD) is not specific for RNA Pol III, as it inhibits all three eukaryotic polymerases (I, II, and III) by binding to DNA[33]. Therefore, the shorter half-life observed in our data suggests that the effect of ActD on other RNA polymerases might influence U6 levels and stability, as other RNA polymerases may modulate upstream regulators involved in U6 transcription, maturation, and/or stability.

To determine whether binding of YTHDF2 to U6 snRNA is dependent on m⁶A methylation, we assessed the effect of knocking down the known m⁶A methyltransferase METTL16 on U6 snRNA levels[18,19]. *METTL16* knockdown increased U6 levels in HaCaT cells (Fig. 2P, Supplementary Fig. S10D), while it decreased both baseline and UVB-induced m⁶A enrichment of U6 in HaCaT cells (Fig. 2Q). Furthermore, *METTL16* knockdown reduced YTHDF2 binding to U6 under baseline condition and UVB stress in HaCaT cells (Fig. 2R). These results demonstrate that YTHDF2 binds to m⁶A-modified U6 snRNA to mediate U6 decay.

## YTHDF2 controls inflammation by regulating U6 snRNA

To determine the role of U6 in YTHDF2's function, we assessed whether U6 knockdown rescues the effect of YTHDF2 knockdown on inflammatory gene expression. Indeed, U6 (*RNU6-1*) knockdown drastically inhibited the effect of *YTHDF2* knockdown on the expression of cytokine genes in HaCaT cells (Fig. 3A-C and Supplementary Fig. S11A). Prior literature has shown that UVB exposure directly damages the snRNA U1 to promote inflammation[34]. To determine whether the effect of U6 on cytokine expression occurs through direct UVB damage to U6 snRNA, we assessed the effect of UVB-irradiated synthetic U6 snRNA. tRNA was used as a negative control RNA[34]. However, we found that in vitro UVB-irradiated U6 snRNA had no effect on *IL-6* expression when transfected into HaCaT cells (Fig. 3D, E). Next, we

assessed whether the effect of YTHDF2 inhibition is mediated by the direct damage on U6 snRNA by UVB irradiation. We examined the difference in inflammatory response in HeLa cells, a cell line with high transfection efficiency, with or without YTHDF2 inhibition transfected with sham- or UVB-irradiated synthetic non-m⁶A-modified (U6)- and m⁶A-modified U6 (m⁶A U6) in vitro. Consistently, in vitro UVB irradiation of U6 or m⁶A U6 molecules had no effect on IL-6 expression compared to sham (Fig. 3F). In WT HeLa cells, while both U6 and m⁶A U6 induced *IL-6* expression, an inflammatory gene robustly upregulated by UVB irradiation and/or YTHDF2 inhibition (Figs. 2I and 3B), U6 lead to a higher induction than m⁶A U6 (Fig. 3F). We also observed that *YTHDF2* deletion in HeLa cells increased *IL-6* expression and abolished the difference between U6 and m⁶A U6 (Fig. 3F).

To explore the role of m⁶A methylation of U6 in YTHDF2's function in skin cancer, we next assessed the consequence of METTL16 inhibition on the expression of cytokines in the human A431 skin carcinoma cells. *METTL16* knockdown enhanced the expression of *IL-6* and *TNF-α*, two inflammatory genes robustly upregulated by UVB irradiation and/or *YTHDF2* inhibition (Figs. 2I and 3B), under baseline conditions in A431 cells (Fig. 3G–H, Supplementary Fig. S11B). The effect of *METTL16* knockdown was reduced by U6 knockdown in A431 cells (Fig. 3I-L), indicating that METTL16 suppresses cytokine expression through regulating U6. It is noteworthy that as compared with HaCaT and HeLa cells, A431 cells showed reduced basal METTL16 expression, which may render A431 cells sensitive to *METTL16* knockdown-induced cytokine expression (Supplementary Fig. S11C). *METTL16* knockdown increased UVB-induced COX-2 expression in HaCaT cells (Fig. 3M). However, dual knockdown of both *METTL16* and *YTHDF2* mimicked the effect of *YTHDF2* knockdown alone in HaCaT cells (Fig. 3N–Q). Similarly, *YTHDF2* knockout abolished the effect of *METTL16* knockdown in HeLa cells (Fig. 3R–W, Supplementary Fig. S11D, E). Finally, to determine whether METTL16 inhibition affects dsRNA levels, as previously observed following METTL3 inhibition[35,36], and to assess whether U6 snRNA contributes significantly to the dsRNA pool, we compared dsRNA levels in A431 cells under control, *METTL16* knockdown, and/or U6 knockdown conditions using immunofluorescence analysis with the anti-J2 antibody. However, *METTL16* knockdown, U6 knockdown, or the combined knockdown had no effect on dsRNA levels in A431 cells (Fig. S11F-S11G), suggesting that METTL16 regulates inflammatory response via a distinct mechanism from METTL3 and that the effect of U6 depletion on inflammatory gene expression is not mediated by secondary dsRNA fragment accumulation. Together, these results demonstrate that YTHDF2 suppresses inflammatory gene expression, at least in part, by controlling the metabolism of m⁶A methylated U6.

## YTHDF2 binding to m⁶A U6 prevents m⁶A U6-mediated TLR3 activation

Our RNA-seq data analysis showed that genes in the TLR3 cascade are upregulated by *YTHDF2* depletion (Fig. 1A), suggesting that TLR3 signaling may contribute to the inflammatory response induced by YTHDF2 loss. TLR3 is a pattern recognition receptor best known for detecting dsRNA[37] and structured RNAs resembling dsRNA to activate immune responses[38,39]. Indeed, a small-scale siRNA screening for *TLR3*, *TLR7*, *TLR8*, *MDA5*, and *RIG-I*, which are all known sensors for RNA[40,41], showed that knockdown of TLR3 at least partially reverses the effect of YTHDF2 knockdown in HaCaT cells (Fig. 4A, B). In addition, *TLR7/8* knockdown also partially reversed the effect of *YTHDF2* knockdown in HaCaT cells (Fig. 4A, B), suggesting an important role of these receptors in YTHDF2's function and warrants further investigation. Moreover, inhibition of TLR3 at least partially reversed the effect of either *YTHDF2* knockdown or *METTL16* knockdown both under UVB stress in HaCaT cells and under baseline condition in A431 cells (Fig. 4C–G and Supplementary Fig. S12A–J). Together, these data demonstrate that the

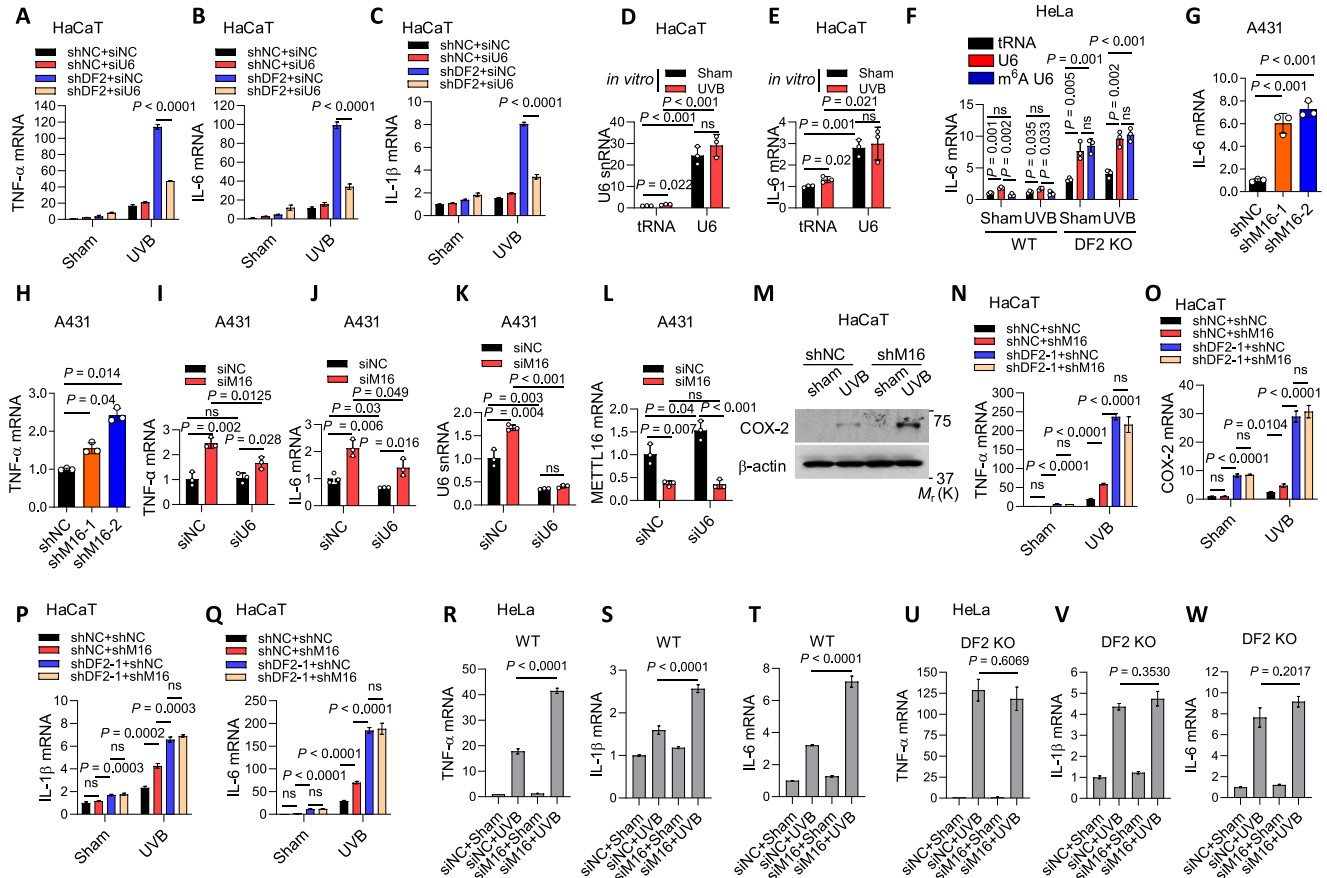

**Fig. 3 | YTHDF2 controls inflammatory gene expression through U6 m⁶A methylation. A–C** qPCR of TNF-α, IL-6 and IL-1β mRNAs in HaCaT cells with or without YTHDF2 or U6 knockdown, with or without UVB irradiation. siU6, small interfering RNA targeting U6 snRNA. qPCR of U6 snRNA (**D**) and IL-6 mRNA (**E**) in HaCaT cells 30 h after transfection with sham- or UVB-irradiated purified tRNA or synthetic U6 snRNA. **F** qPCR of IL-6 mRNA in wild-type (WT) or YTHDF2 knockout (KO) HeLa cells 30 h after transfection with sham- or UVB-irradiated purified tRNA, synthetic U6 snRNA, or synthetic m⁶A-modified U6. **G, H** qPCR of TNF-α and IL-6 mRNAs in A431 cells transduced with shNC (short hairpin RNA negative control), shMETTL16-1 or shMETTL16-2. **I–L** qPCR of TNF-α, IL-6, METTL16 and U6 snRNA in A431 cells with or without siMETTL16 or siU6. **M** Immunoblot of COX-2 in

HaCaT cells with or without METTL16 knockdown, with or without UVB irradiation. **N–Q** qPCR of TNF-α, COX-2, IL-1β and IL-6 mRNAs in HaCaT cells with or without YTHDF2 or METTL16 knockdown, with or without UVB irradiation. **R–W** qPCR of TNF-α, IL-1β and IL-6 mRNAs in WT and YTHDF2 knockout (DF2 KO) HeLa cells with or without METTL16 knockdown, with or without UVB irradiation. siM16, small interfering RNA targeting METTL16. Housekeeping genes used were HPRT1 (**A–C,N–W**), 18S rRNA (**D, K**) and GAPDH (**E–J, L**). Statistical analyses were performed using a two-tailed unpaired Student's *t*-test. Data are presented as mean ± SE (*n* = 4 for **A–C, N–W**) or mean ± SD (*n* = 3 for **D–L**). All experiments were conducted using biologically independent samples.

effect of YTHDF2 on inflammation is mediated, at least in part, through TLR3.

To further determine whether the effect of YTHDF2 inhibition on the inflammatory response is mediated through U6/TLR3, we assessed the effect of U6 knockdown, *TLR3* knockdown, and their combined depletion. We found that either U6 knockdown or TLR3 knockdown decreased *IL-6* expression, and that combined depletion of TLR3 and U6 had a similar effect to single knockdown of either U6 (*RNU6-1*) or *TLR3* in A431 cells (Fig. 4H–J), suggesting that the effect of U6 on inflammation is mediated through TLR3 signaling. In addition, U6 knockdown had no effect on the dsRNA level in A431 cells (Fig. S11F), suggesting that U6 depletion does not affect secondary dsRNA fragment accumulation in regulating inflammatory response.

Given the well-established role of U6 snRNA in regulation of splicing[13], we aimed to assess whether splicing inhibition affects the inflammatory response driven by YTHDF2 loss through U6 upregulation. We found that the splicing inhibitor Pladienolide B (PlaB) decreased the expression of the proinflammatory cytokines in both control and *YTHDF2*-knockdown cells (Supplementary Fig. S12K), mimicking the effect of U6 knockdown (Fig. 3). In parallel, PlaB also decreased the U6 snRNA level, which correlates with the decrease in

cytokine expression caused by PlaB (Supplementary Fig. S12K). These findings raise the possibility that splicing inhibition by PlaB reduces global transcription or alters RNA metabolism, leading to decreased expression of inflammatory genes and U6 snRNA.

To determine whether UVB-induced DNA damage also contributes to the induction of inflammation, we assessed the effect of *TLR9* knockdown or cGAS inhibition. We found that either *TLR9* knockdown or inhibition of cGAS had no effect on UVB-induced expression of inflammatory genes (Fig. S12L–S12M). These data suggest that UVB-induced inflammation is not mediated by UVB-induced DNA damage.

To determine the specific mechanism by which YTHDF2 regulates U6 and TLR3 signaling, we performed several in vitro binding and pull-down assays. Consistent with observations in mRNA, we found that recombinant YTHDF2 preferentially binds to m⁶A U6 more than U6 in vitro (Supplementary Fig. S13A). In contrast, recombinant TLR3 bound to both U6 and m⁶A U6 with a similar affinity in vitro (Fig. 4K). An m⁶A immunoprecipitation assay showed that 16.5% U6 snRNA is m⁶A modified in A431 cells (Fig. 4L), similar to previous findings in mouse embryonic fibroblasts[42]. Additionally, pull-down assays showed that the U6 36-60 sequence, where the

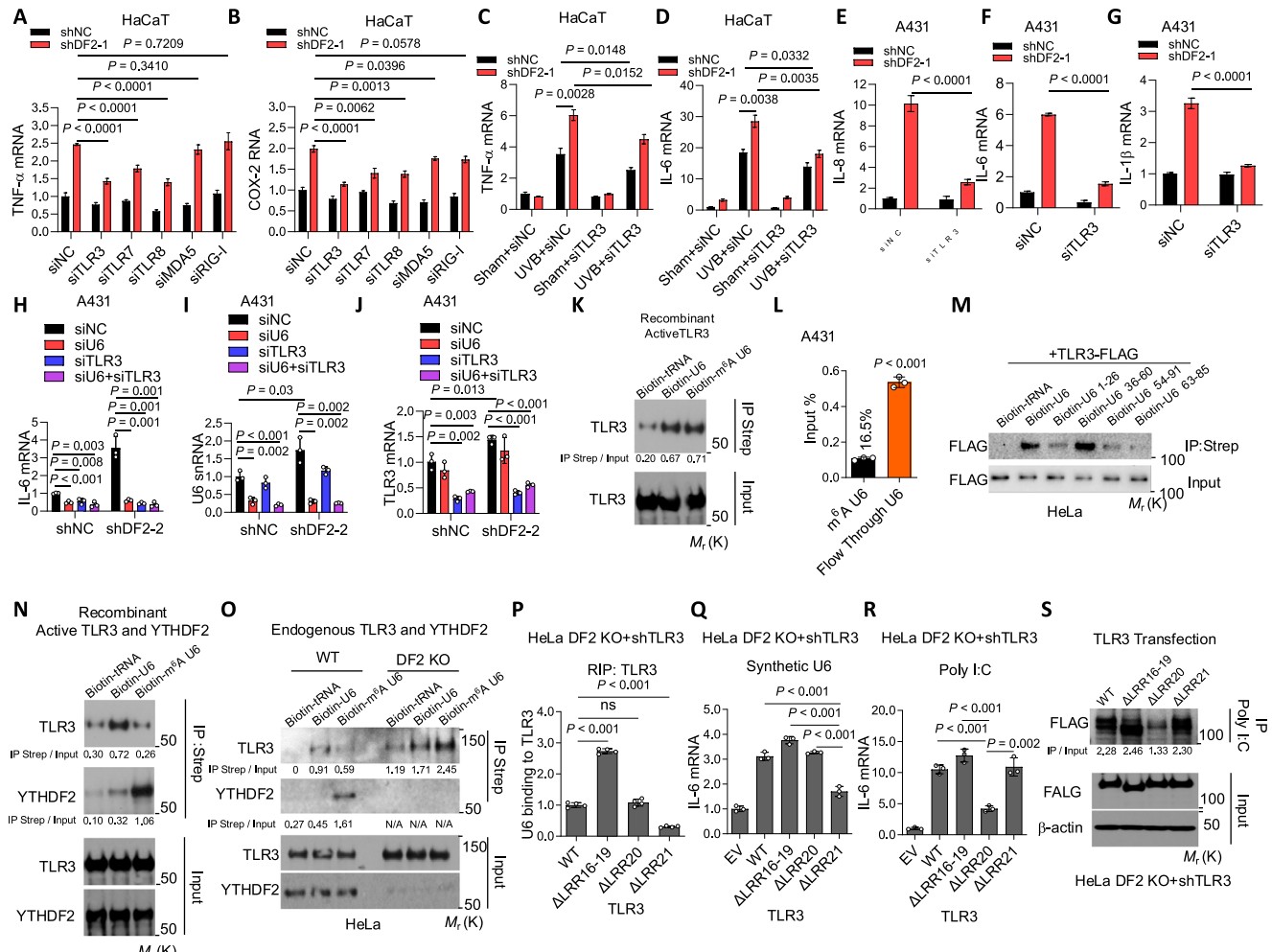

**Fig. 4 | YTHDF2 interacts with m⁶A U6 and thus inhibits m⁶A U6 binding to TLR3.** **A, B** qPCR of TNF-α and COX-2 mRNAs in shNC and shDF2 HaCaT cells transfected with siNC, siTLR3, siTLR7, siTLR8, siMDA5 or siRIG-I (siRNAs targeting TLR3, TLR7, TLR8, MDA5 and RIG-I, respectively). **C, D** qPCR of TNF-α and IL-6 mRNAs in shNC and shDF2 HaCaT cells transfected with siNC or siTLR3, 6 h after sham or UVB irradiation. **E–G.** qPCR of IL-8, IL-6 and IL-1β mRNAs in shNC and shDF2 A431 cells. **H–J** qPCR of IL-6, U6 snRNA and TLR3 in A431 cells with or without U6 and/or TLR3 knockdown. **K** Pull-down showing interaction between biotin-labelled U6 or m⁶A-U6 and recombinant TLR3. **L** qPCR of U6 snRNA in m⁶A-IP or flow-through fractions in A431 cells, showing the proportion of m⁶A-modified U6. **M** Pull-down of biotin-labelled tRNA, U6 or truncated U6 oligos (1–26, 36–60, 54–91, 63–85) with TLR3 in lysates of HeLa cells overexpressing TLR3–FLAG. **N** Pull-down

of biotin-labelled tRNA, U6 or m⁶A-U6 with recombinant YTHDF2 and TLR3. **O.** Pull-down of biotin-labelled tRNA, U6 or m⁶A-U6 with YTHDF2 or TLR3 in lysates from WT and DF2 KO HeLa cells overexpressing TLR3–FLAG. **P** RIP of U6 interaction with TLR3 (WT and mutants) in DF2 KO HeLa cells. **Q–R** qPCR of IL-6 mRNA in DF2 KO HeLa cells with TLR3 knockdown and overexpression of WT or mutant TLR3, following treatment with synthetic U6 or poly I:C (polyinosinic–polycytidylic acid, synthetic dsRNA analog). **S** Pull-down of biotin-labelled poly I:C with WT or mutant TLR3 in DF2 KO HeLa cells with TLR3 knockdown and overexpression of WT or mutant TLR3–FLAG. Housekeeping genes: HPRT1 (**A–G**, **Q–R**), GAPDH (**H, J**) and 18S rRNA (**I**). Statistical analyses were performed using a two-tailed unpaired Student's t-test. Data are shown as mean ± SE (n = 4 for **A–G**) or mean ± SD (n = 3 for **H–J, L, P–R**). All experiments used biologically independent samples.

A43 m⁶A motif is located, preferentially binds with TLR3 in HeLa cells (Fig. 4M, Supplementary Fig. S13B, C). When recombinant TLR3 was incubated with U6 or m⁶A U6 in the presence of recombinant YTHDF2, we found that U6 exhibited preferred binding with TLR3 (Fig. 4N). In comparison, m⁶A U6 preferentially bound with YTHDF2, but not TLR3 (Fig. 4N). In wild-type (WT) HeLa cells, pull-down assays showed that U6 binds with TLR3 but not YTHDF2, while m⁶A U6 binds with YTHDF2, not TLR3 (Fig. 4O), confirming the findings from in vitro binding assays (Fig. 4N). However, in *YTHDF2* knockout cells, both U6 and m⁶A U6 bound with TLR3 with m⁶A U6 showing more prominent binding with TLR3 than U6 (Fig. 4O), suggesting that TLR3 prefers to bind with m⁶A U6 in vivo, distinct from the findings from in vitro binding assays (Fig. 4K), possibly due to structural differences between endosomal TLR3 in vivo and recombinant TLR3 in vitro. Together, these findings demonstrate that YTHDF2 specifically inhibits TLR3 binding to m⁶A U6.

TLR3 recognizes its ligand, dsRNA, through its N-terminal ecto-domain, which contains 23 leucine-rich repeats (LRRs)[43]. Mutational analysis has shown that the LRR20 motif of TLR3 binds to poly I:C[44], a synthetic analog of dsRNA and known TLR3 ligand. To determine the specific LRR motif of TLR3 that binds to U6, we generated various LRR deletion mutants of motifs (ΔLRR16-19, ΔLRR20, and ΔLRR21) that are in the C-terminus of TLR3 around the poly I:C-binding LRR20 motif[44] to determine which mutants show loss of binding to U6 and activation by U6. A RIP assay showed that deletion of LRR21 leads to loss of binding to U6, while deletion of LRR20 had no effect in HeLa cells with *YTHDF2* knockout and *TLR3* knockdown (Fig. 4P). Intriguingly, deletion of LRR16-19 enhanced binding of TLR3 to U6 (Fig. 4P), suggesting that these motifs suppress the TLR3-U6 interaction, warranting future investigation to determine the underlying mechanism for such sup-pression. Further analysis showed that deletion of the LRR21 motif, but not the LRR20 motif, resulted in loss of induction of *IL-6* by U6 in HeLa

cells with YTHDF2 knockout and TLR3 knockdown (Fig. 4Q). In comparison, deletion of the LRR20 motif, but not the LRR21 or LRR16-19 motifs, led to loss of induction of *IL-6* expression by Poly I:C in HeLa cells with YTHDF2 knockout and TLR3 knockdown (Fig. 4R), consistent with previous studies[44]. Furthermore, a pull-down assay confirmed that LRR20 is required for binding to poly I:C in HeLa cells with *YTHDF2* knockout and *TLR3* knockdown (Fig. 4S). Together, these results demonstrate that the m[6]A motif sequence of U6 snRNA interacts with and activates TLR3 via the LRR21 motif.

## YTHDF2 and U6 snRNA localize to endosomes to mediate inflammation

TLR3 is an RNA sensor localized in endosomes[40,41]. To determine the mechanism by which YTHDF2 regulates TLR3 activation, we assessed whether U6 and YTHDF2 also localize in the cytoplasm, particularly in endosomes, and whether YTHDF2, U6, and TLR3 interact. First, we found that both *YTHDF2* knockdown and UVB irradiation increased the cytoplasmic U6 level in HeLa, HaCaT, and A431 cells (Fig. 5A and Supplementary Fig. S14A–E). Second, UVB irradiation increased both total level and endosomal proportion of U6 snRNA in HeLa cells (Fig. 5B). Third, we performed RNA fluorescence in situ hybridization (FISH) of U6 in combination with immunofluorescence analysis of the endosome marker Rab7 to confirm U6 localization in endosomes. While U6 snRNA was mainly detected in the nucleus, we observed that UVB increases U6 snRNA levels in the cytoplasm and U6 colocalization with the late endosome marker Rab7 in HaCaT cells, HeLa cells, and mouse skin (Fig. 5C, Supplementary Fig. S14F–S14G), supporting U6 localization in endosomes. YTHDF2 inhibition in HaCaT cells, mouse skin, or A431 cells increased U6 snRNA levels in the nucleus and cytoplasm and increased U6 colocalization with Rab7 (Fig. 5C, Supplementary Fig. S14F–S14I), while U6 knockdown decreased them in A431 and HaCaT cells (Supplementary Fig. S14H–S14I). These orthogonal validations further suggest a role for endosomal U6 snRNA in the inflammatory response. Next, we demonstrated YTHDF2 colocalization with endosomes using multiple approaches, including endosome fractionation in A431 cells and Endo-IP in 293 T cells, which were selected for their high plasmid transfection and protein expression (Fig. 5D, E). Endo-IP enabled rapid isolation of early/sorting endosomes through affinity capture of the early endosome-associated protein EEA1[45]. Further, confocal imaging analysis in HeLa cells confirmed that YTHDF2 localization in endosomes, as shown by the colocalization of YTHDF2 with the late endosome marker Rab7 (Fig. 5F).

To determine the mechanism by which U6 and YTHDF2 translocate to endosomes, we assessed the role of dynamin, a key factor in the endocytosis pathway that mediates biomolecule trafficking[46]. Indeed, the dynamin inhibitor Dynasore reduced the endosomal proportion of U6 and the endosomal YTHDF2 level in HeLa cells and reduced U6 stability in A431 cells (Fig. 5G–I, supplementary Fig. S14J), indicating a critical role of endocytosis in the localization of U6 and YTHDF2 to endosomes. To further understand the molecular mechanism by which U6 and YTHDF2 are trafficked to endosomes, we assessed the role of SIDT2, a RNA transporter localized in late endosomes and lysosomes[47,48]. Indeed, knockdown of *SIDT2* reduced the total and endosomal proportion of U6, endosomal YTHDF2 levels, and U6 stability in A431 cells (Fig. 5J–L and supplementary Fig. S14K–S14L). These results mimic the effect of Dynasore and indicate that SIDT2 is required for the localization of U6 and YTHDF2 to endosomes. To examine whether YTHDF2 or U6 enters endosomes as a complex or individually, we assessed the effect of knockdown of U6, YTHDF2, or METTL16 on endosomal localization. Knockdown of U6 reduced the YTHDF2 levels in endosomes in A431 cells (Fig. 5M and supplementary Fig. S14M), indicating that U6 is required for YTHDF2 localization to endosomes. We found that knockdown of either YTHDF2 or METTL16 increased both endosomal U6 levels and total U6 levels in HeLa cells (Fig. 5N, O), indicating that neither YTHDF2 nor m[6]A methylation is

required for U6 entry into endosomes. In addition, we found that the C-terminus of YTHDF2 is required for its localization to endosomes, as wild-type (DF2 (WT)) full-length and C-terminal YTHDF2 (DF2 (C)), but not N-terminus YTHDF2 (DF2 (N)), were detected in endosomes (Fig. 5P, Q). Together, our data demonstrate that SIDT2 transports either the U6-YTHDF2 complex altogether or U6 alone into endosomes, while YTHDF2 entry into endosomes requires U6 snRNA.

Next, to characterize whether YTHDF2 or U6 snRNA is trafficked intracellularly or extracellularly to endosomes, we assessed whether cells uptake either YTHDF2 or U6 snRNA secreted by other cells. However, when cultured with conditioned medium from control cells, no YTHDF2 or U6 snRNA was detected in endosomes of HeLa cells with *YTHDF2* deletion or A431 cells with U6 snRNA knockdown, respectively (Fig. S14N–S14O), suggesting that intracellular trafficking pathways deliver YTHDF2 and U6 snRNA into endosomes. Lastly, to determine whether U6 snRNA entry into endosomes is required for the increased expression of inflammatory genes, we assessed the effect of Dynasore, an inhibitor of U6 endosomal entry (Fig. 5G), on cytokine expression. We found that Dynasore reduced expression of inflammatory genes induced by U6 in HeLa cells, or by *YTHDF2* knockdown in A431 cells at baseline condition (Fig. 5R, S) and in HaCaT and HeLa cells treated with UVB irradiation (Fig. S14P, S14Q). In control cells with YTHDF2 expression, Dynasore also slightly reduced inflammatory gene expression in A431 cells and *IL-6* expression in UVB-irradiated HaCaT cells, but not in UVB-irradiated HeLa cells (Fig. 5S and Fig. S14P, S14Q), suggesting a cell-type-dependent, YTHDF2-independent effect of Dynasore. Such effect of Dynasore may be due to the reduction of endosomal localization of non-m[6]A-modified U6, which can be YTHDF2-independent, as non-m[6]A-modified U6 is bound by TLR3 but not by YTHDF2 (Fig. 4K, O). Future investigation is needed to elucidate the specific molecular mechanism for regulating U6 snRNA metabolism and function. Taken together, these findings demonstrate that the endocytosis pathway mediates the localization of U6 and YTHDF2 to endosomes, thus regulating TLR3 activation and inflammation.

## YTHDF2 is inhibited by UVB irradiation

Since both YTHDF2 loss and UVB treatment similarly increase U6 stability (Fig. 2O), we next asked whether UVB inhibits YTHDF2 by modulating YTHDF2 post-translational modifications. Mass spectrometric analysis showed that at 1 h, UVB induced dephosphorylation of YTHDF2 at serine 39 (S39) in HaCaT cells (Fig. 6A, Supplementary Fig. S15A). To confirm this observation, we generated an antibody specific for YTHDF2 phosphorylation at S39 and found that UVB decreases YTHDF2 S39 phosphorylation post-irradiation in HaCaT cells (Fig. 6B). We next re-examined YTHDF2-interacting proteins and found that UVB inhibits the binding of YTHDF2 with several kinases and CNOT1, the subunit for the CCR4−NOT deadenylase complex that mediates m[6]A-methylated RNA decay[49], but promotes YTHDF2 binding to MYPT1 (myosin phosphatase target subunit 1) (Supplementary Fig. S15B–D), a regulatory subunit for myosin phosphatase that is a member of protein phosphatase type 1[50]. We hypothesized that UVB might induce YTHDF2 dephosphorylation by the MYPT1-phosphatase complex to inhibit the YTHDF2-CNOT1 interaction and thus U6 decay. Coimmunoprecipitation (Co-IP) analysis demonstrated that UVB induces the YTHDF2-MYPT1 interaction in HeLa cells (Supplementary Fig. S15E). Further co-IP analysis demonstrated that the phosphomimetic S39D YTHDF2 mutant shows enhanced binding to MYPT1 as compared with WT YTHDF2 and the non-phosphorylatable S39A mutant in HeLa cells with YTHDF2 knockout (Fig. 6C), suggesting a potential role of the MYPT1−phosphatase complex in dephosphorylating YTHDF2 at S39. In addition, as compared with WT YTHDF2, we found that the phosphomimetic S39D YTHDF2 mutant shows increased binding to CNOT1, while the S39A mutant shows decreased binding in HeLa cells with

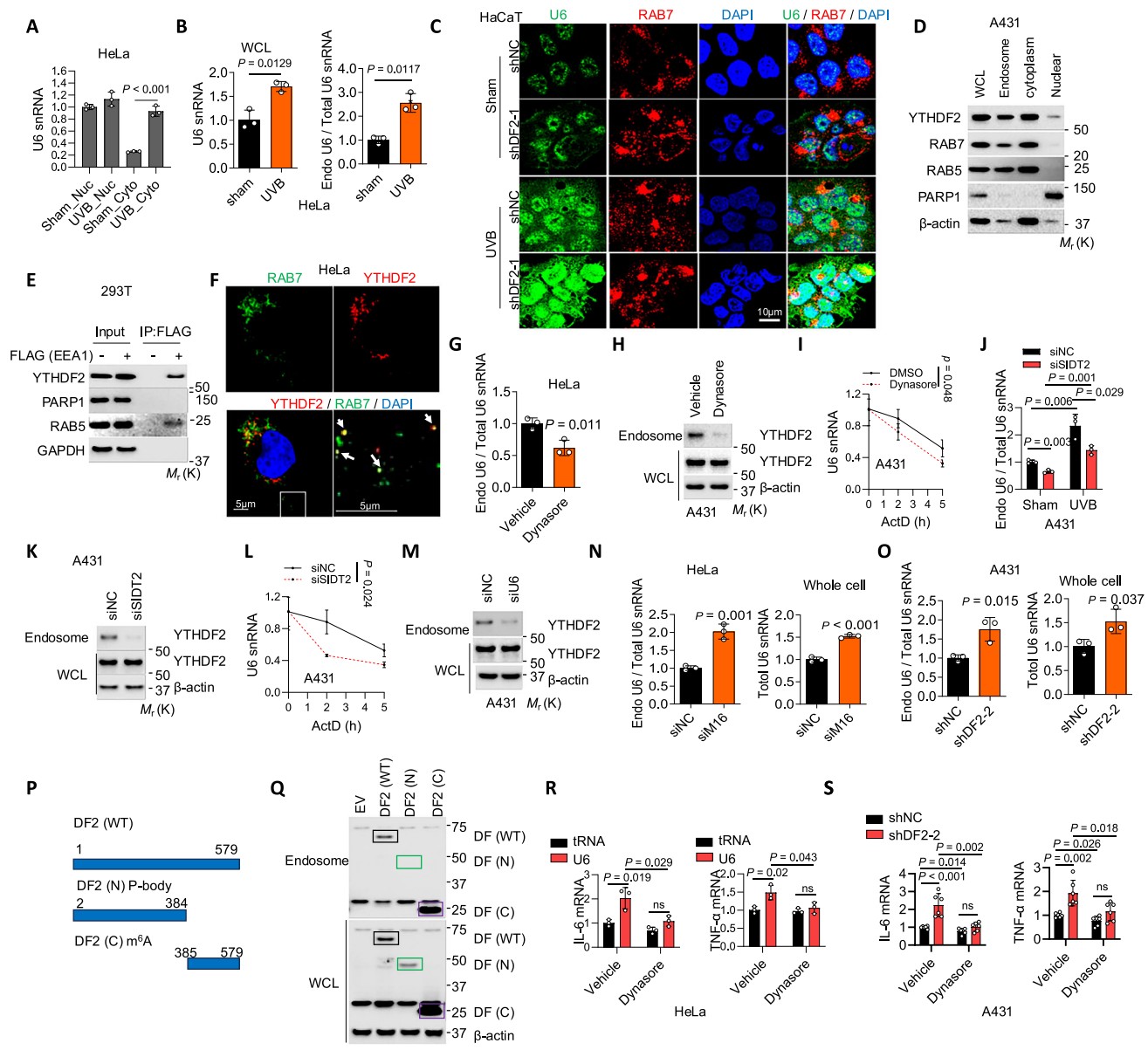

**Fig. 5 | Both U6 and YTHDF2 are localized in endosomes. A** qPCR of nuclear (nuc) and cytoplasmic (cyto) fractions in HeLa cells with or without UVB irradiation (30 mJ/cm², 6 h). **B**. qPCR in HeLa cells with or without UVB irradiation (30 mJ/cm², 6 h). **C** Confocal images of HaCaT cells with or without YTHDF2 knockdown, 6 h after sham or UVB irradiation (20 mJ/cm²). **D** Immunoblot in A431 cells. **E** Endo-IP of endosomal fractions in 293 T cells. **F** Confocal images of HeLa cells; arrows indicate colocalization of YTHDF2 and Rab7. **G** qPCR in endosomes of HeLa cells with or without Dynasore. **H** Immunoblot in A431 cells with or without Dynasore. **I** qPCR of mRNA stability in A431 cells treated with or without Dynasore. **J** qPCR in endosomes in A431 cells with or without SIDT2 knockdown. **K** Immunoblot of WCL and endosomes in A431 cells with or without SIDT2 knockdown. **L** qPCR of mRNA stability in A431 cells with or without SIDT2 knockdown. **M** Immunoblot of WCL and

endosomes in A431 cells with or without U6 knockdown. **N** qPCR in endosomes of HeLa cells with or without METTL16 knockdown. **O** qPCR in endosomes of A431 cells with or without YTHDF2 knockdown. **P** Schematic of human YTHDF2 and truncations (N- or C-terminal). **Q** Immunoblot of WCL and endosomes in HeLa cells transfected with EV, DF2 (WT), DF2 (N) or DF2 (C). Unmarked bands >25 kDa are nonspecific. qPCR of mRNAs in HeLa cells transfected with tRNA or U6 and treated with vehicle or Dynasore (**R**), and in A431 cells with or without YTHDF2 knockdown treated with vehicle or Dynasore (**S**). Housekeeping genes: 18S rRNA (**B, I, L, N, O**) and GAPDH (**R–S**). Statistical analyses were performed using a two-tailed unpaired Student's t-test. Data are shown as mean ± SD (n = 3 for **A–B, G, I, J, L, N, O, R**; n = 6 for **S**). All experiments used biologically independent samples.

YTHDF2 knockout (Fig. 6D), indicating that S39 phosphorylation of YTHDF2 is required for the YTHDF2-CNOT1 interaction. Consistently, WT YTHDF2, but not the S39A mutant, decreased U6 snRNA level and U6-induced-TNF-α expression, while both WT and S39A mutant YTHDF2 showed similar binding affinity to U6 in HeLa cells with *YTHDF2* knockout (Fig. 6E, Supplementary Fig. S15F−I).

Next, we assessed the role of CNOT1 in U6 decay. Indeed, we found that CNOT1 binds to U6 in HeLa cells (Fig. 6F) and CNOT1 knockdown inhibits U6 decay in both HeLa and A431 cells (Fig. 6G, H,

and Supplementary Fig. S15J, K). To elucidate the mechanism of YTHDF2-mediated U6 decay, we examined the role of YTHDF2 and METTL16-mediated m⁶A methylation in CNOT1 binding to U6 snRNA. We found that either *YTHDF2* deletion or *METTL16* knockdown drastically reduces CNOT1 binding to U6 in HeLa cells (Fig. 6I, J and Supplementary Fig. S15L, M), supporting a critical role of YTHDF2 and U6 m⁶A methylation in CNOT1 binding to U6 snRNA. To determine whether UVB irradiation affects CNOT1 expression, we assessed the effect of UVB irradiation on CNOT1 protein levels. We found that UVB

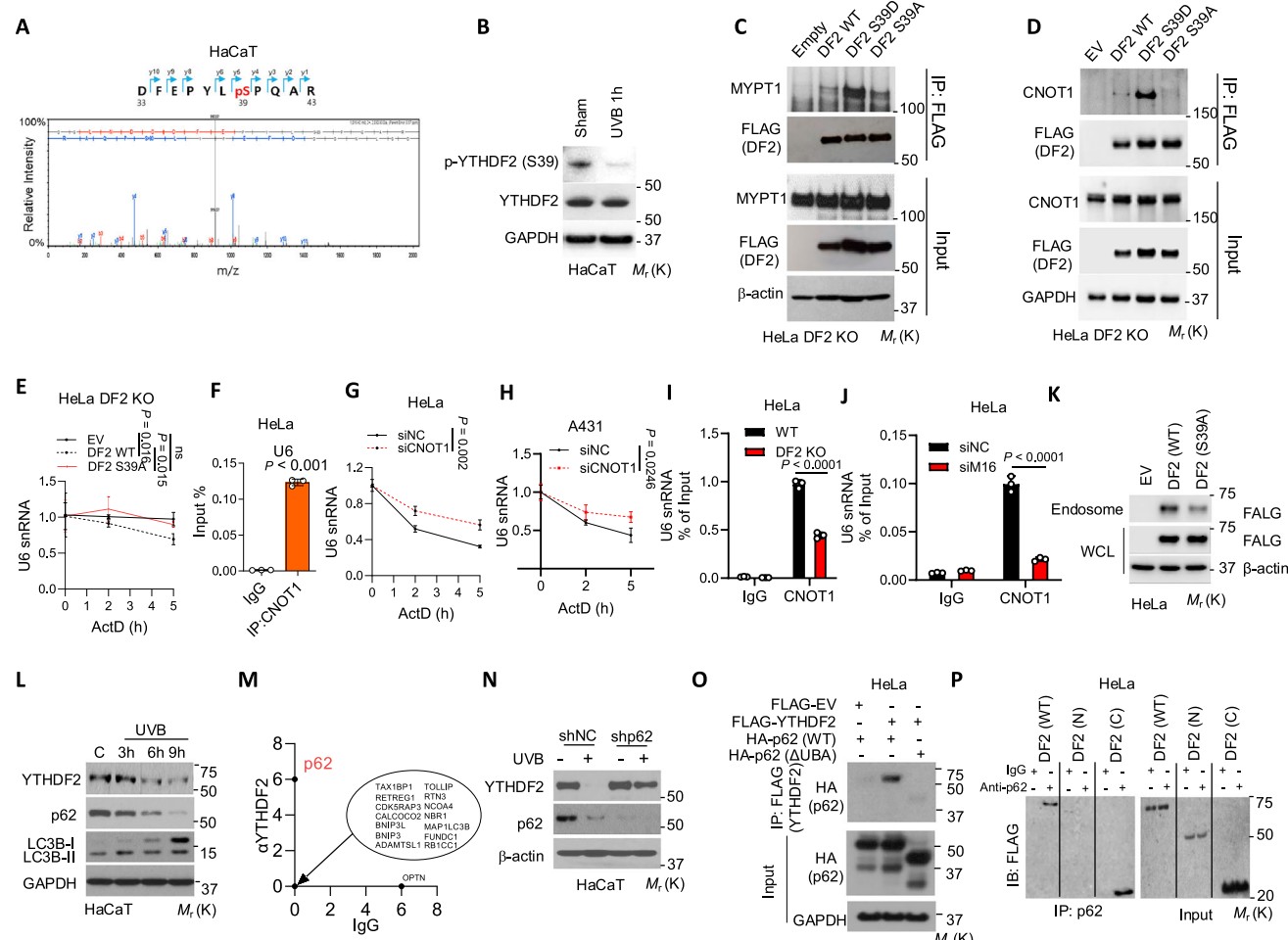

**Fig. 6 | YTHDF2 is inhibited by UVB irradiation. A** Loss of YTHDF2 phosphorylation at serine 39 (S39) in HaCaT cells after UVB irradiation identified by mass spectrometry. **B** Immunoblot of p-YTHDF2 (S39) and YTHDF2 in HaCaT cells with or without UVB irradiation (1 h). **C** Co-IP of MYPT1 with FLAG-tagged YTHDF2 (EV, WT, S39D or S39A) in DF2 KO HeLa cells. **D** Co-IP of CNOT1 with FLAG-tagged YTHDF2 (EV, WT, S39D or S39A) in DF2 KO HeLa cells. **E** qPCR of U6 snRNA stability in DF2 KO HeLa cells transfected with DF2 WT or DF2 S39A. **F** RIP of U6 with CNOT1 in HeLa cells. qPCR of U6 snRNA stability in HeLa (**G**) and A431 (**H**) cells with or without CNOT1 knockdown. RIP of U6 with CNOT1 in HeLa cells with or without YTHDF2 knockout (**I**) or METTL16 knockdown (**J**). **K** Immunoblot of FLAG-tagged YTHDF2 in WCL and endosomes from HeLa cells transfected with EV, DF2 WT or

DF2 S39A. **L** Immunoblot analysis of YTHDF2, p62, and LC3-I/II in HaCaT after sham (**C**) or UVB irradiation. **M** Unique spectrum counts for autophagy receptors including p62 f from mass spectrometry of YTHDF2-interacting proteins as compared with IgG. **N** Immunoblot of YTHDF2, p62, and LC3-I/II in HaCaT cells expressing shNC or shp62, 6 h after sham or UVB irradiation. **O** Co-IP of FLAG-tagged YTHDF2 with HA-tagged p62 (WT or ΔUBA) in HeLa cells. **P** Co-IP of FLAG-tagged YTHDF2 (WT, N-terminus or C-terminus) with p62 in HeLa cells. Housekeeping gene: 18S rRNA (**E, G, H**). Statistical analyses were performed using a two-tailed unpaired Student's t-test. Data are shown as mean ± SD ($n = 3$ for **E–J**). All experiments were conducted using biologically independent samples.

irradiation had no effect on CNOT1 protein expression in either HeLa or HaCaT cells (Supplementary Fig. S15N). These results demonstrate that CNOT1 binds to U6 snRNA and mediates U6 decay through interacting with YTHDF2 in an m6A-methylation dependent manner. Furthermore, the S39A mutant reduced endosomal localization in HeLa cells (Fig. 6K). These findings demonstrate that CNOT1 mediates U6 decay and dephosphorylation of YTHDF2 inhibits U6 decay and endosomal localization.

In addition to inhibiting YTHDF2 phosphorylation, we observed that UVB down-regulates YTHDF2 protein in multiple cell lines and primary keratinocytes including HaCaT, CHL-1 (melanoma cells), HeLa, and NHEK cells (Fig. 6L, and Supplementary Fig. S16A–C). This occurred in parallel with autophagy induction, as shown by the downregulation of the selective autophagy receptor p62 and the induction of LC3-II, despite the increase in the *YTHDF2* mRNA level in HaCaT cells (Fig. 6L, and Supplementary Fig. S16A-D).

UVB irradiation induces an acute inflammatory response as observed by increased expression of inflammatory genes (Fig. 1). To

determine whether the inflammatory response correlates with YTHDF2 protein abundance, we assessed YTHDF2 protein levels at different time points. Indeed, as compared with sham-irradiation, YTHDF2 protein level is decreased at 6 h post-UVB irradiation, while it was recovered at 24 h post-UVB irradiation in HaCaT cells (Supplementary Fig. S16D), likely due to increased *YTHDF2* mRNA levels (Supplementary Fig. S16D). The temporal regulation of YTHDF2 protein expression by UVB irradiation negatively correlated with the increases in U6 snRNA levels (Fig. 2I) as well as inflammatory gene expression[51] at 6 h followed by a decrease at 24 h post-UVB irradiation, further supporting an inhibitory role of YTHDF2 in UVB-induced inflammation. Importantly, UVB irradiation had little effect on other m6A readers in HaCaT cells (Supplementary Fig. S16E). Furthermore, UVB irradiation also down-regulated YTHDF2 protein in mouse skin (Supplementary Fig. S16F, G).

Next, to determine whether YTHDF2 down-regulation induced by UVB radiation is mediated through autophagy, a cellular self-eating process that degrades excessive or damaged proteins and organelles[52],

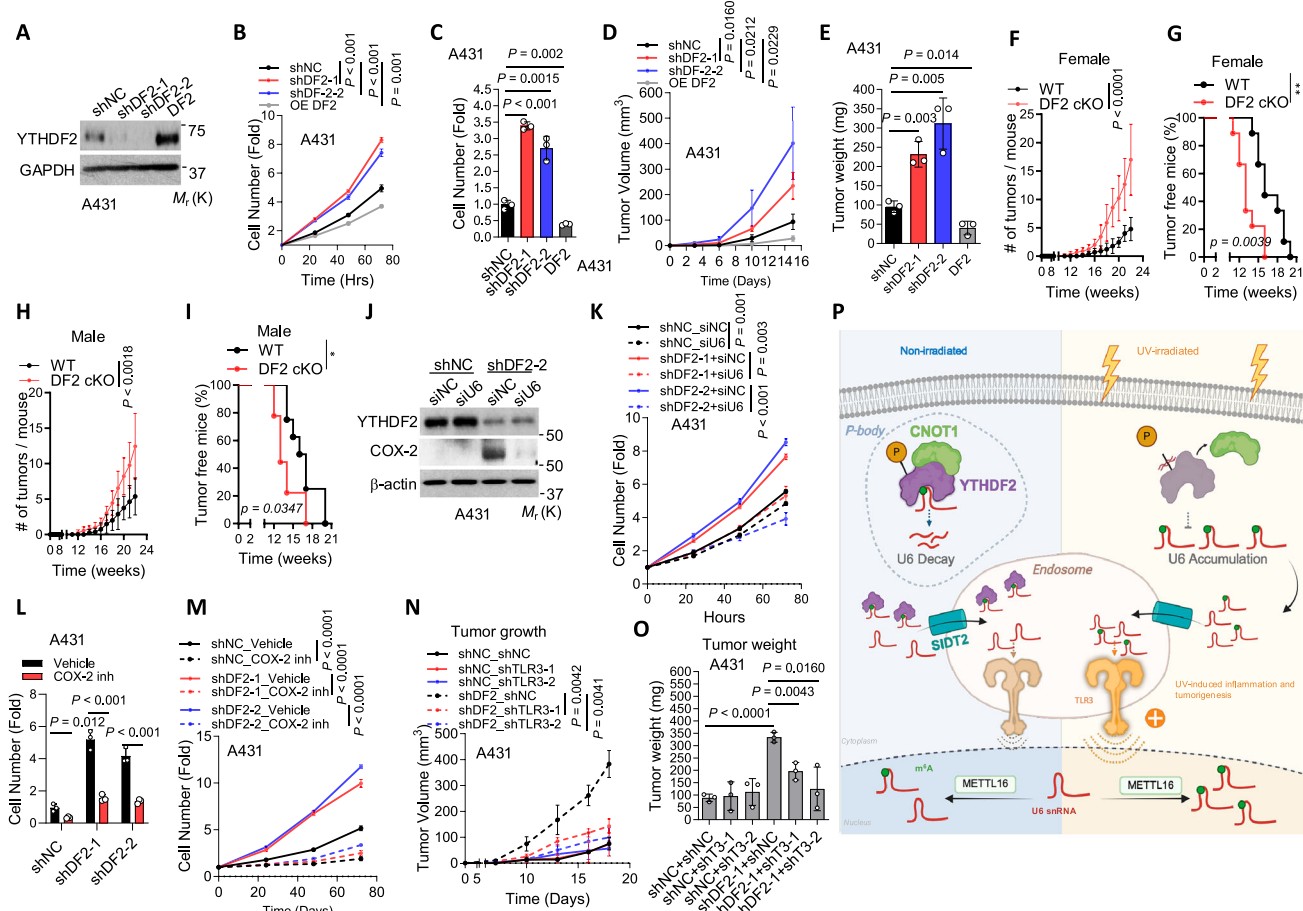

**Fig. 7 | YTHDF2 suppresses skin tumorigenesis in mice through inhibiting TLR3. A** Immunoblot confirming YTHDF2 knockdown and over-expression in A431 cells. **B** Cell proliferation of cells as in A. **C** Cell migration of cells as in A. **D** Tumor growth of A431 xenografts in nude mice. **E** Tumor weight for A431 xenografts. **F-I** Number of tumor per mouse (F and G) and percentage of tumor-free mice (H and I) in WT and DF2 cKO mice following chronic UVB irradiation. **J** Immunoblot of YTHDF2 and COX-2 in shNC and shDF2 A431 cells with or without U6 knockdown. **K** Cell proliferation of shNC and shDF2 A431 cells with or without U6 knockdown. Cell migration (**L**) and proliferation (**M**) in A431 cells with or without YTHDF2 knockdown, treated with or without Celecoxib (20 μM, COX-2

inhibitor, 24 h). Tumor volume (**N**) and tumor weight (**O**) in nude mice injected with shNC or shDF2 A431 cells with or without TLR3 knockdown. **P** Schematic of the role of YTHDF2 in regulating U6 snRNA decay and interaction with TLR3 to control UVB-induced inflammation and tumorigenesis. f Created in BioRender. Verghese, M. (https://BioRender.com/k4uvdti). Statistical analyses were conducted using a two-tailed unpaired Student's *t*-test (**B–F**, **H**, **K–O**) and log-rank test (**G**, I). Data are shown as mean ± SD ($n = 3$ for **B–E**, **K**, **L–O**; $n = 9$ for each group in **F**, **G**; $n = 8$ [WT] and $n = 9$ [DF2 cKO] in **H**, I). All experiments used biologically independent samples.

we sought to establish whether autophagy inhibition reversed the effect of UVB irradiation on YTHDF2 and U6 levels. Using HaCaT cells with or without knockdown of the essential autophagy gene *ATG5* or *ATG7*, we found that inhibition of either ATG5 or ATG7 prevented UVB-induced down-regulation of YTHDF2 and U6 up-regulation (Supplementary Fig. S16H–J). To identify the autophagy receptor responsible for UVB-induced autophagy of YTHDF2, we re-examined the list of YTHDF2-interacting proteins from our mass spectrometry analysis. We observed that YTHDF2 preferentially binds with the autophagy receptor p62 (also known as SQSTM1) in HaCaT cells (Fig. 6M, and Supplementary Fig. S16K). Indeed, knockdown of p62 prevented UVB-induced YTHDF2 down-regulation and U6 up-regulation in HaCaT cells (Fig. 6N and Supplementary Fig. 16L). Furthermore, in mouse skin, skin-specific conditional knockout (cKO) of *Atg5*, *Atg7*, or *Sqstm1*(*p62*) prevented UVB-induced YTHDF2 down-regulation (Supplementary Fig. S16M, N). Co-IP showed that p62 binds to YTHDF2 in HeLa and HaCaT cells. Additionally, the p62-YTHDF2 interaction requires p62's ubiquitin-associated (UBA) domain in HeLa cells (Fig. 6O and Supplementary Fig. S16O), which is known to interact with ubiquitinated proteins[53]. As both p62 and YTHDF2 can bind to RNA[27,54], we assessed whether the p62-YTHDF2 interaction is dependent on RNA. We found

that removal of RNA had no effect on the interaction between p62 and YTHDF2 in HaCaT cells (Supplementary Fig. S16P). In addition, we found that the C-terminus, but not the N-terminus, of YTHDF2 is required for YTHDF2 binding to p62 (Fig. 6P). These results demonstrate that YTHDF2 is degraded via p62-dependent selective autophagy in response to UVB stress.

### YTHDF2 suppresses tumorigenesis by inhibiting TLR3 activation

As inflammation can promote tumorigenesis[22] and the role of YTHDF2 in established cancers is cancer-type-dependent[55–57], we next investigated the function of YTHDF2 in skin tumorigenesis. Indeed, our RNA-seq data analysis showed that genes in the Pathways in cancer are upregulated by *YTHDF2* knockdown in HaCaT cells (Fig. 1A), suggesting a role for YTHDF2 in skin cancer. In addition, we found that knockdown of YTHDF2 in A431 cells increases cell proliferation and migration in vitro, whereas forced overexpression of YTHDF2 in A431 cells decreases cell proliferation and migration (Fig. 7A–C, and Supplementary Fig. S17A). A431 xenografts in nude mice revealed that knockdown of YTHDF2 increased tumor growth, whereas forced overexpression of YTHDF2 decreased tumor growth (Fig. 7D, E, and Supplementary Fig. S17B). Furthermore, skin-specific *Ythdf2* deletion

accelerated skin tumorigenesis induced by chronic UVB irradiation in both male and female mice, demonstrating an inhibitory role of YTHDF2 in tumor initiation (Fig. 7F–I). Histological analysis showed that UVB irradiation induces skin tumor formation in both WT and DF2 cKO mice, while squamous skin carcinoma was detected in the DF2 cKO mice (Supplementary Fig. S17C). This was supported by immunofluorescence analysis that indicated expression of the basal keratinocyte marker KRT14 (K14) in both basal layer of mouse skin and mouse tumors, while the differentiation marker KRT10 (K10) was only detected in mouse skin, hair follicles, tumor from WT mice, but not in the tumor from DF2 cKO mice (Supplementary Fig. S17C).

Next, we assessed whether YTHDF2 protein level is altered in human skin cancer. Immunofluorescence analysis showed that YTHDF2 was lower in human skin squamous cell carcinoma samples (both stage 1 and stage 2) than in normal human epidermis (Supplementary Fig. S17D, E). Furthermore, YTHDF2 was lower in stage 2 SCC than stage 1 SCC (Supplementary Fig. S17D, E), suggesting an active role for YTHDF2 in skin tumorigenesis as well as tumor progression.

To determine the role of U6 in YTHDF2's functional role, we assessed whether U6 inhibition rescues the effect of YTHDF2 knockdown on cell proliferation. U6 knockdown drastically inhibited the effect of YTHDF2 knockdown on COX-2 expression and cell proliferation in A431 cells (Fig. 7J, K and Supplementary Fig. S17F). Next, we assessed the importance of inflammatory genes in YTHDF2 function. Among these inflammatory mediators regulated by YTHDF2 (Fig. 1), COX-2 is a UVB-inducible enzyme that promotes inflammation and tumorigenesis[58]. Indeed, the COX-2 inhibitor Celecoxib drastically suppressed the proliferation and migration of YTHDF2 knockdown and control cells; it also diminished the effect of *YTHDF2* knockdown in A431 cells (Fig. 7L, M, Supplementary Fig. S17G). These findings indicate that YTHDF2 loss promotes tumor cell proliferation by increasing U6/COX-2. In addition, inhibition of TLR3 also reversed the tumor-promoting effect of YTHDF2 knockdown in mice from A431 cells (Fig. 7N, O and Supplementary Fig. S17H, I). These findings demonstrate that YTHDF2 suppresses tumorigenesis through inhibiting the TLR3 pathway.

## Discussion

Despite emerging investigations into the critical roles of m⁶A mRNA methylation in inflammation and cancer[55–57,59], the regulatory and functional role of m⁶A methylation of self non-coding RNA in inflammation and tumorigenesis remains largely unknown. Here, we show that YTHDF2 recognizes m⁶A methylation of non-coding U6 snRNA to control TLR3 activation in inflammation and tumorigenesis (Fig. 7P). m⁶A-methylated U6 snRNA is recognized by YTHDF2, leading to U6 decay. In addition, both YTHDF2 and U6 localize to endosomes via SIDT2 and the endocytosis pathway. YTHDF2 binds to m⁶A U6 to compete with TLR3 binding to m⁶A U6, thus inhibiting the expression of inflammatory modulators, cell proliferation, and tumor growth. In addition, UVB irradiation inhibits YTHDF2 phosphorylation at S39, which is critical for YTHDF2 interaction with CNOT1 and localization in endosomes. YTHDF2 is also downregulated by UVB irradiation at later time points through p62-mediated selective autophagy. YTHDF2 is reduced in human skin cancer, SLE, and type I diabetes, as compared with normal controls. Skin-specific YTHDF2 deletion sensitizes mice to UVB-induced inflammation and tumorigenesis. Our findings demonstrated a critical role of YTHDF2 in regulating self non-coding RNA in inflammation and tumorigenesis (Fig. 7P).

YTHDF2 was initially shown to be an mRNA-binding protein that recognizes m⁶A methylation in mRNA and promotes mRNA decay by directly interacting with CNOT1 and recruiting the CCR4–NOT deadenylase complex[27,49]. Here, we show that YTHDF2 binds to m⁶A-modified self U6 snRNA to mediate its decay. As a key component of the spliceosome, U6 binds to and interacts with many protein factors[13]. Upon m⁶A methylation by METTL16, U6 structure may be altered such that a sequence on or around the m⁶A site (A43) becomes available to preferentially bind to YTHDF2. Previously it has been shown that m⁶A RNA methylation regulates the RNA-structure-dependent accessibility of RNA binding motifs to affect RNA–protein interactions for mRNAs[7]. Thus, it is possible that m⁶A methylation of U6 induces U6 structural remodeling to permit YTHDF2 binding. In addition, the interaction between YTHDF2 and U6 snRNP proteins may also facilitate YTHDF2 access to U6 snRNA. Moreover, we also show that CNOT1, which is critical for the CCR4-NOT mRNA deadenylase activity in the decay of mRNAs[60], including m⁶A-methylated mRNAs[49], binds to U6 snRNA through interacting with YTHDF2 in an m⁶A-methylation dependent manner, as *YTHDF2* deletion or *METTL16* knockdown inhibited the CNOT1-U6 snRNA interaction. Previously, the 3' tail of a small fraction of U6 snRNA has been shown to be adenylated[61]. Although U6 is not known to be polyadenylated, recent studies have detected m⁶A methylated U6 snRNA in poly(A)⁺ RNA in worms[10]. While this may represent remnants left after poly(A)⁺ enrichment from total RNA, it is also possible that a fraction of U6 snRNA is polyadenylated in worms and other organisms, which could regulate U6 decay upon m⁶A methylation similar to poly(A)⁺ mRNAs via the YTHDF2-CNOT1 interaction[49]. It is possible that adenylation and/or polyadenylation cooperates with m⁶A methylation of U6 snRNA to regulate U6 snRNA turnover. Future investigation is warranted to elucidate the detailed mechanism by which YTHDF2 and CNOT1 regulate U6 snRNA decay. Nevertheless, our findings establish self U6 snRNA as a previously unrecognized non-coding RNA target of YTHDF2.

Intriguingly, previous studies have also shown that loss of *mett-10* in *C. elegans* increases U6 snRNA levels[10], while there is no m⁶A reader protein like YTHDF2 in worms. There are three possibilities: (1) m⁶A methylation of U6 may affect other U6 snRNA modifications that in turn regulate U6 snRNA stability[13], (2) m⁶A methylation of other *mett-10* target genes may regulate U6 snRNA transcription, maturation, and/or stability, or (3) worms may express other proteins that recognize m⁶A-methylated U6 snRNA to mediate U6 snRNA decay. Future investigations are required to test these possibilities. In contrast, a previous report by Warda and colleagues did not detect a change to U6 snRNA levels upon *METTL16* knockdown in HEK293 cells[19]. These findings, together with our own, suggest that METTL16's regulation of U6 snRNA levels may be context-dependent, warranting further investigation to elucidate the molecular basis for these differing effects across cell types and organisms.

Previous reports have suggested that RNA modifications, including m⁶A methylation, of in vitro-transcribed synthetic RNAs, inhibit activation of TLR3 and RIG-I[62,63]. Accordingly, m⁶A methylation has been shown to play critical roles in the immune recognition of circular RNAs[64] and non-coding RNAs generated by back splicing[65], inhibiting their binding to RIG-I and subsequent RIG-I activation. U6 is best known for its role in splicing and functions in the nucleus[13]. However, emerging evidence has shown that U6 can be localized in the cytoplasm and the nucleus[66]. Recently, several snRNAs including U6 snRNA have been shown to be glycosylated and localized at the cell surface[67], suggesting emerging biological roles of cytoplasm and surface U6 snRNA in RNA biology. In addition, free U6 snRNA has a double-stranded 5'-short stem, 3'-telestem, and internal stem-loop (ISL) regions (Supplementary Fig. S13B)[18,68], which may permit U6 binding to TLR3 as a self RNA pathogen-associated molecular pattern (PAMP). Our findings demonstrate that endogenous U6 snRNA is localized in endosomes and binds to TLR3 via the LRR21 motif, but not LRR20, the known motif that the synthetic ligand poly I:C binds, suggesting that TLR3 utilizes distinct LRR motifs for different RNA agonists. Our data also demonstrates that TLR3 prefers to bind to m⁶A U6 than U6 when YTHDF2 is absent (Fig. 4O), suggesting that the m⁶A motif and/or methylation may be critical for TLR3 binding. Intriguingly, we found that TLR3 preferentially binds to the U6 sequence containing the m⁶A

methylation site A43, which is also bound by YTHDF2 (Fig. 4M). Thus, YTHDF2 may compete with TLR3 in binding to m⁶A U6, thereby inhibiting TLR3 activation. Distinct from the role of U1 snRNA in UVB-induced inflammation[34], in vitro irradiation of U6 with UVB had no effect on cytokine expression, further supporting the critical role of the U6 m⁶A methylation sequence, but not UVB-induced de novo damage lesions of U6, in TLR3 activation. Under baseline conditions, *METTL16* knockdown increases both total and endosomal U6 abundance and cytokine expression in METTL16-low A431 cells (Fig. 4N), suggesting that non-m⁶A-modified U6 snRNA, which is not bound by YTHDF2, can also enter endosomes and be bound by TLR3 in a YTHDF2-independent manner.

Intriguingly we show that both YTHDF2 and U6 are localized in endosomes, permitting YTHDF2 binding to U6 in endosomes to regulate TLR3 activation. YTHDF2 is known to localize in P-bodies to mediate the decay of m⁶A-methylated mRNAs[27], while U6 is mainly localized in the nucleus as well as cytoplasm[66]. Transporting RNA from the nucleus to endosomes and other membrane locations requires RNAs to cross a membrane at least once. The mammalian *Sidt1* and *Sidt2* genes, two homologs of the *C. elegans sid-1* gene[69,70], have been shown to mediate the trafficking of RNAs across cellular or intracellular vesicle membranes such as lysosomes[71]. Recently SIDT2 has been shown to be localized in endosomes[47]. In addition, RNA transport by the *C. elegans* Sid-1 protein is shown to be specific for dsRNA[69]. Indeed, previous studies have shown that free U6 snRNA contains double-stranded regions (Supplementary Fig. S13B)[18,68], which may permit U6 to interact with SIDT2 and thus be transported to endosomes. Taken together, we show that both U6 snRNA and YTHDF2 are localized to endosomes to control TLR3 activation.

Specifically, our findings demonstrate that both U6 snRNA and YTHDF2 enter endosomes through SIDT2- and dynamin-dependent intracellular trafficking pathway, while YTHDF2 is transported into endosome by the RNA transporter SIDT2 through binding to m⁶A-methylated U6 snRNA. This model is supported by the following key findings (Fig. 5, supplementary Fig. S14): (1) *SIDT2* knockdown or dynamin inhibition reduces entry of both U6 snRNA and YTHDF2 to endosomes, (2) to our best knowledge, YTHDF2 does not have an N-terminal peptide signal sequence that would enable YTHDF2 access to the endosomal lumen, (3) U6 knockdown reduces YTHDF2 levels in endosomes, (4) YTHDF2-N, a mutant lacking the m6A-RNA binding domain, showed reduced endosomal level, (5) knockdown of *YTHDF2* or *METTL16* increased U6 snRNA levels in endosomes, indicating that neither YTHDF2 nor m⁶A methylation of U6 snRNA is required for U6 snRNA entry into endosomes, and (6) extracellular uptake for either U6 snRNA or YTHDF2 was not observed for endosome transportation. Taken together, these findings support a model that U6 snRNA is transported into endosomes at least in part by SIDT2 in a dynamin-dependent, m⁶A methylation-independent, and YTHDF2-independent manner, while YTHDF2 entry into endosomes is dependent on U6 snRNA.

UVB radiation causes damage to biomolecules including DNA, resulting in genomic instability. Besides genomic instability, UVB irradiation also induces inflammation, which manifests as sunburn, a critical enabling hallmark for tumorigenesis and aging. Our findings demonstrate that YTHDF2 inhibition enhances UVB-induced inflammation, leading to epidermal hyperplasia in mice. UV damage triggers a number of pathways such as the induction of COX-2[58] and proinflammatory cytokines[72] in epidermal keratinocytes. Previously, we and others have demonstrated that *COX-2* in keratinocytes acts as a proinflammatory and tumor-promoting factor in skin tumorigenesis and is upregulated in human skin cancer[51,73,74]. UV-induced COX-2 upregulation can initiate the synthesis of the principle inflammation mediator PGE₂[58] and thus increase keratinocyte proliferation and hyperplasia in the epidermis. Other cytokines such as TNF-α, IL-6, and IL-1β also play

critical roles by contributing to the induction of COX-2 and other genes involved in epidermal inflammation and hyperplasia. In this study, we found that YTHDF2 inhibition increases UV-induced COX-2 expression in primary keratinocytes and cultured human keratinocytes and epidermal hyperplasia, and that the COX-2 inhibitor decreased cell proliferation in cells with YTHDF2 knockdown, supporting a critical role of COX-2 in increased cell proliferation caused by YTHDF2 inhibition.

Our study also suggests that UVB might reduce YTHDF2 phosphorylation at S39 by promoting the interaction between YTHDF2 and MYPT1, which could be involved in the autophagic degradation of YTHDF2. Previous studies have shown that CHK1, a kinase activated by DNA damage and UV radiation[75], binds and phosphorylates MYPT1, leading to the recruitment of protein phosphatase 1β (PP1cβ) to dephosphorylate its substrate[76]. It remains to be investigated whether UV radiation-induced CHK1 activation mediates MYPT1 phosphorylation, thus leading to increased MYPT1 binding to YTHDF2. In addition, YTHDF2 S39/T381 phosphorylation by EGFR/SRC/ERK signaling has previously been shown to promote YTHDF2 protein stability[77]. It is possible that UVB-induced YTHDF2 dephosphorylation mediates YTHDF2 down-regulation by autophagy. In addition, we show that YTHDF2 S39 phosphorylation is critical for its interaction with CNOT1, U6 decay, and its endosomal trafficking, suggesting that YTHDF2's function in promoting RNA decay and endosomal localization is regulated by its phosphorylation at S39. Future investigation is needed to determine whether phosphorylation at other sites regulates YTHDF2 activity. In addition to stabilizing U6 by inhibiting YTHDF2, UVB further increases U6 snRNA levels in YTHDF2-depleted cells (Fig. 2I-J), suggesting that UVB also induces U6 snRNA transcription, leading to TLR3 activation independent of YTHDF2. These YTHDF2-dependent and -independent mechanisms may serve as a multilayer defense mechanism to ensure the induction of a robust inflammatory and innate immune response in the skin under UVB stress to prevent microbe infection and maintain barrier function. Future investigation is warranted to fully map the molecular machinery that regulates U6 m⁶A methylation, decay, and trafficking under baseline conditions, UVB stress, and other stresses. Taken together, our studies establish a critical role for YTHDF2 in recognizing m⁶A methylation of U6 snRNA, thus increasing U6 decay and competing with m⁶A U6 binding to TLR3 to suppress TLR3 activation in inflammation and tumorigenesis under baseline and UVB stress conditions. These findings may also open promising opportunities for targeting the YTHDF2/m⁶A U6 axis in the intervention and treatment of epithelial cancers and autoimmune diseases, such as SLE.

## Methods
All research complied with relevant ethical regulations. Human tissue samples were obtained from US Biomax under protocols approved by Institutional Review Boards, with informed consent obtained from the donors, as documented by the vendor. All animal experiments were conducted in accordance with institutional guidelines and were approved by the Institutional Animal Care and Use Committee (IACUC) of the University of Chicago.

### Human skin tumor samples
Human tissue microarrays were obtained from US Biomax (Derwood, MD). According to the vendor's documentation, all human tissues were collected under protocols approved by Institutional Review Boards, with informed consent obtained from the donors. The samples were provided in a fully de-identified form, ensuring that no personally identifiable information was accessible to the investigators. All research involving human-derived samples was conducted in accordance with the ethical principles set forth in the Declaration of Helsinki.

## Cell culture and treatment

Primary normal human epidermal keratinocytes (NHEK) were purchased from Invitrogen (Invitrogen, C-001-5 C) and Lonza (00192907). NHEK cells from Invitrogen were maintained in NHEK complete medium (Invitrogen, S-001-5) according to the manufacturer's instructions, and NHEK cells from Lonza maintained in KGM Gold keratinocyte growth basal medium (Lonza, #00192151) and KGM Gold keratinocyte growth medium supplements and growth factors (Lonza, # 00192152). HaCaT cells (human keratinocyte, kindly provided by Dr. Fusenig), A431 cells (human squamous carcinoma cells, ATCC, CRL-1555), HEK-293T cells (human embryonic kidney cells, ATCC, CRL-3216), HeLa cells (WT and YTHDF2 knockout [KO] cells, kindly provided by Dr. Chuan He), and CHL-1 cells (melanoma cells, provided by the Comprehensive Cancer Center Core Facilities at the University of Chicago) were maintained in Dulbecco's modified Eagle's medium (Invitrogen, Carlsbad, CA) supplemented with 10% fetal bovine serum (Gibco), 100U/ml penicillin, and 100 µg/ml streptomycin (Invitrogen, Carlsbad, CA). Cells were treated with the following agents: Dynasore (80 µM, 16 h. Sigma, D7693), human cGAS inhibitor G140 (5 µM, 24 h. InvivoGen), Pladienolide B (500 nM, 3 h. MedChemExpress, HY-16399), followed by sham or UVB irradiation (20 mJ/cm$^2$, 6 h, unless indicated otherwise), or actinomycin D (2 µM. Sigma, SBR00013) post-sham or UVB irradiation. Cells were irradiated with sham or UVB irradiation (20 mJ/cm$^2$ for HaCaT cells and 30 mJ/cm$^2$ for other cell lines that exbibit decreased sensitivity to UVB stress) and then collected for analysis at 6 h or 24 h for the delayed UVB stress response such as inflammatory gene expression and YTHDF2 protein down-regulation, or 1 h for the early or direct UVB stress response such as YTHDF2-interacting proteins and YTHDF2 phosphorylation, unless indicated otherwise.

## Mouse tumorigenesis and UVB treatment

All animal procedures were approved by the University of Chicago institutional animal care and use committee (IACUC). Athymic nude mice were obtained from Harlan Sprague-Dawley (now Envigo). For xenograft experiments, one million cells were injected subcutaneously into the right flanks of 6-week-old female nude mice. Tumor growth was monitored and measured weekly by a caliper, and tumor volume was calculated using the formula, Tumor volume (mm$^3$) = d$^2$ X D/2, where d and D are the shortest and the longest diameters, respectively.

Mice with wild type (*WT, Ythdf2*$^{flox/flox}$) & conditional skin-specific YTHDF2 homozygous knockout (*DF2 cKO, K14Cre;Ythdf2*$^{flox/flox}$), WT & ATG5 cKO (*K14Cre;Atg5*$^{flox/flox}$), WT & ATG7 cKO (*K14Cre;Atg7*$^{flox/flox}$), and WT & p62 cKO (*K14Cre;p62*$^{flox/flox}$) were generated and maintained on an SKH-1 hairless background[26,51,78], Both male and female mice (6–8 weeks old) were used for UVB irradiation experiments. The initial dose of UVB was 80 mJ/cm$^2$ for the first week, followed by a weekly 10% increase until it reached 100 mJ/cm$^2$. WT and cKO were irradiated with UVB every other day 3 times a week, and tumor formation was recorded[26]. For short UVB treatment, mice were irradiated with UVB irradiation (100 mJ/cm$^2$) 3 times every other day and skin samples were collected 24 h after the final UVB irradiation.

All mice were housed in the University of Chicago Animal Resources Center (ARC) under specific pathogen-free conditions. ARC follows standard operating conditions, including a 12-h light/12-h dark cycle, ambient temperature of 21–23 °C, and relative humidity of 40–60%, with food and water provided ad libitum.

## Primary mouse keratinocyte isolation and culture

Primary keratinocytes were isolated from neonatal mice as described previously in ref. 79. Briefly, WT (*Ythdf2*$^{flox/+}$) and DF2 cHet (K14*Cre;Ythdf2*$^{flox/+}$) neonatal mice were euthanized and rinsed briefly with sterile PBS containing penicillin-streptomycin, followed by a quick rinse with 70% ethanol and then submerging in fresh 70% ethanol for 10 minutes to allow thorough surface sterilization. Mice were then rinsed again with sterile PBS (with penicillin-streptomycin) and kept in fresh PBS on ice. Skin was collected on a 60 mm tissue culture dish containing 5 mL of cold sterile Ca$^{2+}$-free 0.25% trypsin solution without EDTA (Invitrogen, Cat#15090-046, 10x stock, use PBS to dilute), with dermis facing down, and incubated at 4 °C for 15 to 24 h. Skin was then transferred to a dry, sterile tissue culture plate and epidermis was separated from dermis using a sterile pasteur pipette. Epidermis is minced and suspended in 6 ml of growth medium (10% FBS/DMEM, HaCaT medium) in a 15 mL tube, following by pipetting up and down to release keratinocytes from the epidermis. The suspension was then filtered through a sterile, 70 µm nylon filter (Thermo fisher, #22363548) into a fresh 50 mL tube to remove cornified sheets. The samples were then rinsed with 5 mL fresh growth medium, filtered through the same filter into the same 50 mL tube, and centrifuged at 160 *g* for 5 min at room temperature. Cells were resuspended in KGM® Gold Keratinocyte Growth Medium (Lonza, 00192060), counted, and seeded at 0.5–1 × 10$^6$ cells in 35 mm dishes. 48–72 h after plating, medium was removed and replaced with fresh culture medium for UVB irradiation experiments. WT and DF2 cHet cells were irradiated with UVB (30 mJ/cm$^2$) and then samples were collected at 6 h post-sham or -UVB irradiation for m$^6$A-IP qPCR analysis.

## Lentiviral generation and infection

pLKO.1 plasmids of *shNC, shYTHDF2, shMETTL16, shATG5, shATG7, shp62*, and *shTLR3_GFP* were obtained from Sigma. pLenti plasmids of overexpression of YTHDF2 were purchased from Applied Biological Materials Inc. (Abm). Lentivirus was produced by co-transfection into HEK-293T (human embryonic kidney) cells with lentiviral constructs together with the pCMVdelta8.2 packaging plasmid and pVSV-G envelope plasmid using GenJet™ Plus DNA In Vitro Transfection Reagent (Signagen, Ijamsville, MD). Virus-containing supernatants were collected 24–48 h after transfection and used to infect recipients. Target cells were infected in the presence of Polybrene (8 µg/ml) (Sigma-Aldrich, St. Louis, MO) and selected with puromycin (Santa Cruz Biotechnology, Santa Cruz, CA) at 1 µg/ml for 6 days.

## Plasmids and generation of mutants

YTHDF2 (WT, N, and C) plasmids were kindly provided by Dr. Chuan He. The pCMV3-C-FLAG-TLR3 construct was obtained from Sino Biological (Catalog# HG10190-CF). Primers for this study were designed individually based on the specific deletions. The YTHDF2 mutants (S39A and S39D) and TLR3 deletion mutants were generated using a QuikChange Site-Directed Mutagenesis kit (Catalog# 200518), according to the manufacturer's instructions. Sequences of the primers for cloning of TLR3 or YTHDF2 mutants are shown in Supplementary Table S2. Newly generated plasmids are available upon request from the corresponding authors.

## siRNA and RNA Transfection

Cells were transfected with siRNA targeting negative control (siNC), an individual gene, or the combination of SIDT2, CNOT1, TLR3, TLR7, TLR8, MDA5, RIG-1, YTHDF2, U6[80], METTL16, TLR9, or MYPT1 (Dharmacon, Lafayette, CO). siU6, U6 snRNA, and m$^6$A-modified U6 snRNA (m$^6$A43) were purchased from Integrated DNA Technologies (IDT, Coralville, IA). tRNA (Roche, TRNABRE-RO SKU10109517001) was used as the control for structured RNA. RNAs were transfected using PepMute™ siRNA Transfection Reagent (Signagen, Ijamsville, MD), according to the manufacturer's instructions.

## Quantitative real-time PCR (qPCR)

Quantitative real-time PCR assays (qPCR) were performed using a CFX Connect real-time system (Bio-Rad, Hercules, CA) with Bio-Rad iQ SYBR Green Supermix (Bio-Rad, Hercules, CA). cDNAs were prepared by using iScript™ Reverse Transcription Supermix for RT-qPCR (Bio-Rad, Hercules, CA) for mRNAs, or miRNA All-In-One cDNA Synthesis Kit

(Applied Biological Materials Inc., BC, Canada) for non-coding RNAs. The threshold cycle number (CQ) for each sample was determined in triplicate or quadruplicate. The CQ for values for YTHDF2, TNF-α, IL-6, COX-2, IL-1β, GM-CSF, IL-8, MMP-9, VEGF, IL-16, IL-1α, FOS, JUN, SOX4, SOX9, CD74, U6 snRNA, ATG5, ATG7, p62, SIDT2, CNOT1, METTL16, and TLR3 were normalized against GAPDH, ACTB, 18S rRNA, or HPRT1, which are specifically indicated in figure legends. Sequences of the primers for qPCR are shown in Supplementary Table S3.

### RNA stability assay

RNA stability was assessed using the transcriptional inhibitor actinomycin D (2 μM). Cells were harvested at different time points after treatment with actinomycin D. Total RNA was isolated using an RNeasy plus mini kit (QIAGEN, Hilden, Germany) or TRIzol, following the manufacturer's protocol. The HPRT1 housekeeping gene was used as a loading control, as HPRT1 mRNA does not contain m⁶A modifications, is not bound by YTHDsF2, and is rarely affected by actinomycin D treatment[27].

### Immunofluorescence (IF), confocal imaging, immunohistochemistry (IHC), and hematoxylin and eosin (HE) staining

Immunofluorescence analysis was performed as described previously in refs. [78,81]. For immunofluorescence analysis of cells, cells were fixed with 4% paraformaldehyde/PBS for 30 min and permeabilized with 0.5% Triton X-100 (Sigma-Aldrich, T8787)/PBS for 20 min. Cells were then washed with PBS with 0.05% Triton X-100. The samples were then incubated with a blocking solution of 3% albumin from chicken egg white (Sigma-Aldrich, A5503) in PBS for 1 h. For immunofluorescence analysis of tissues, formalin-fixed, paraffin-embedded tissue sections were pre-treated by antigen retrieval and incubated with a blocking solution of 3% albumin from chicken egg white (Sigma-Aldrich, A5503) in PBS for 1 h. After removal of the blocking solution, samples were incubated with primary antibodies at 4 °C overnight, and then the samples were washed three times with 0.1% Triton X-100/PBS for 10 min at room temperature (RT). Samples were then incubated with fluorochrome-conjugated secondary antibodies for 1 h at RT and washed three times with 0.1% Triton X-100/PBS for 10 min at room temperature. Cells were then fixed in Prolong Gold Antifade with DAPI (Invitrogen, P36931), Fluoromount™ Aqueous Mounting Medium (Sigma-Aldrich, F4680), or Anti-Fade Fluorescence Mounting Medium (Abcam, ab104135), and were then observed under a fluorescence microscope (Olympus IX71, Olympus Life Science, Japan). For confocal microscopy, cells were imaged with the SoRa Marianas Spinning Disk Confocal microscope (Intelligent Imaging Innovations, 3i). Hematoxylin and eosin (HE) staining of tissues and immunohistochemical analysis of YTHDF2 (Proteintech, #247441-1-AP) were performed by the Human Tissue Resource Center (HTRC) core facility at the University of Chicago. Epidermal thickness was measured on HE stained sections, as described previously[78]. Two investigators independently scored the staining intensity blindly, as 3 (strong), 2 (medium), 1 (weak), and 0 (negative). ImageJ (NIH) was also used for analyzing cells and tissue. Information for the antibodies for the IF staining are shown in Supplementary Table S4.

### Fluorescence In-Situ Hybridization (FISH)

Cells were fixed with 4% paraformaldehyde (PFA) for 30 minutes at room temperature (RT). Permeabilization was performed using 0.5% Triton X-100 in PBS supplemented with RNase inhibitor (40 U/mL) for 30 min at 4 °C. Samples were then washed with Buffer A (20% Buffer A (Cat:SMF-WA1-60, BioSearch)+ 10% formamide) for 5 min at RT. Hybridization was carried out overnight at 37 °C using a 150 nM DIG-labeled U6 probe (U6 probe: 5′-CACGAATTTGCGTGTCATCCTT-3′, IDT) in hybridization solution (Cat:SMF-HB1-10, BioSearch) containing 10% formamide and the RNase inhibitor (40 U/mL). After hybridization, samples were thoroughly washed with PBS to prepare for immunofluorescence staining. Blocking was performed using

goat serum for 30 minutes at RT. The anti-Rab7 (ab137029) primary antibody was diluted in blocking buffer and incubated with the samples for 2 h at RT. After washing with PBS, DIG was detected using the anti-DIG Alexa Fluor 488 antibody (1:50), and Rab7 was visualized using an Alexa Fluor 594-conjugated secondary antibody by incubation for 1 h at RT. Finally, samples were washed with PBS, counterstained with DAPI, and then mounted for imaging. For mouse tissue, formalin-fixed, paraffin-embedded (FFPE) tissue sections were first subjected to antigen retrieval. After washing, U6 probe hybridization was performed following the same protocol as for cells. After hybridization, tissue sections were washed with 0.1% Triton X-100 in PBS. Blocking was then performed using 3% chicken egg white albumin (Sigma-Aldrich, A5503) in PBS for 1 h at RT. Primary antibodies were applied, followed by incubation with the corresponding Alexa Fluor-conjugated secondary antibodies. Tissue samples were then fixed in Prolong Gold Antifade with DAPI (Invitrogen, P36931). Cells and tissue samples were then imaged with the SoRa Marianas Spinning Disk Confocal microscope (Intelligent Imaging Innovations, 3i) and a fluorescence microscope (Olympus IX71, Olympus Life Science, Japan), respectively.

### Immunoblotting

Cells were washed twice with ice-cold PBS and lysed with Cell Lysis Buffer (Cell Signaling, #9803) or RIPA buffer (Cell Signaling, #9806) containing inhibitors for proteases and phosphatases. Cell lysates were resolved by SDS-PAGE and transferred onto nitrocellulose membranes followed by blotting. The information for the antibodies for immunoblotting is shown in Supplementary Table S5.

### Flow cytometric analysis

Single cell suspensions were prepared from spleen and skin tissues. Live/dead labeling was performed before the cell surface staining, using a Zombie NIR™ Fixable Viability Kit (Biolegend; catalog number 423106) diluted 1:1,000 in PBS for 10 min at room temperature in dark. Cells were labeled with specific fluorescence-conjugated antibodies to CD45, CD3, CD4, CD8, MHCII, CD11b, CD103, Ly6C, and/or Ly6G for 20 min on ice. Intracellular cytokine staining was performed using an eBioscience Intracellular Fixation & Permeabilization Buffer Set (catalog number 88-8824-00) according to the manufacturer's instructions. Antibodies against FOXP3 (clone; NRRF-30) were used for regulatory T cell staining. Flow cytometric analysis was performed on a Fortessa 4–15 (BD Biosciences) and Attune NxT (Thermo Fisher Scientific) with Flowjo V10.6.1 used for analysis. The information for the antibodies is shown in Supplementary Table S6. The gating strategy is shown in Supplementary Fig. S18.

### m⁶A immunoprecipitation (m⁶A IP)

100 μg to 150 μg total RNA was extracted from cells using TRIzol following the manufacturer's protocol. mRNA was purified using a Dynabeads mRNA DIRECT Kit following the manufacturer's protocols. 1 μg mRNA was sonicated to ~100 nt and m⁶A containing mRNA fragments was enriched with an EpiMark N6-Methyladenosine Enrichment Kit following the manufacturer's protocols. Finally, RNA was isolated using RNA Clean and Concentrator (Zymo Research) and subjected to library preparation with a TruSeq stranded mRNA sample preparation kit (Illumina).

### Quantitative analysis of the m⁶A levels in mRNA

The m⁶A enrichment in mRNA and U6 snRNA was analyzed using ultra–high-performance liquid chromatography-tandem mass spectrometry (UHPLC-MS/MS). Total RNA was extracted from cells using TRIzol following the manufacturer's protocol. U6 snRNA was isolated as described previously[42]. The m⁶A in polyadenylated RNA was quantified using an Agilent 6460 LC-MS/MS spectrometer as described previously[26,42]

## Sequencing data analysis

General pre-processing of reads: all samples were sequenced by Illumina Hiseq4000 with single end 80 bp read length. The adapters were removed by using cutadapt for m6A-seq. RefSeq Gene structure annotations were downloaded from the UCSC Table Browser. Reads were aligned to the reference genome (hg38) using HISAT2. DESeq2 was used for differential gene expression analysis. Lowly expressed genes were pre-filtered out using reads count threshold of 10. An adjusted $p$ value (DESeq2 default) threshold of 0.05 was used for all differential expression analysis. Aligned reads were extended to 150 bp (average fragment size) and converted from genome-based coordinates to isoform-based coordinates. The method used for peak calling was adapted from published work with modifications[28]. To call m6A peaks, the longest isoform of each gene was scanned using a 100 bp sliding window with a 10 bp step. To reduce bias from potential inaccurate gene structure annotation and the arbitrary usage of the longest isoform, windows with read counts of less than 1/20 of the top window in both m6A-IP and the input sample were excluded. For each gene, the read counts in each window were normalized to the median count of all windows of that gene. A Fisher exact test was used to identify the differential windows between IP and input samples. The window was called as positive if the FDR < 0.01 and log2(Enrichment Score) >= 1. Overlapping positive windows were merged. The following four numbers were calculated to obtain the enrichment score of each peak (or window): read counts of the IP samples in the current peak/window (a), median read counts of the IP sample in all 100 bp windows on the current mRNA (b), read counts of the input sample in the current peak/window (c), and median read counts of the input sample in all 100 bp windows on the current mRNA (d). The enrichment score of each window was calculated as $(a \times d)/(b \times c)$. The data have been deposited in the GEO repository with accession number GSE145925.

## Gene-specific m6A IP qPCR

Real-time quantitative PCR (qPCR) was performed to assess the relative m6A abundance of the selected U6 snRNA and mRNA in m6A antibody IP samples and input samples, as described previously[78,81]. Briefly, total RNA was isolated with an RNeasy mini kit or TRIzol. While 500 ng RNA was saved as an input sample, the rest of the RNA was used for m6A IP. 100 μg RNA was diluted into 500 μL IP buffer (150 mM NaCl, 0.1% NP-40, 10 mM Tris, pH 7.4, 100 U RNase inhibitor) and incubated with the m6A antibody (Synaptic Systems, Goettingen, Germany). The mixture was rotated at 4 °C for 2 hours, then Dynabeads® Protein A (Thermo Fisher Scientific, Waltham, MA) coated with BSA was added into the solution and rotated for an additional 2 hours at 4 °C. After washing with the IP buffer with RNase inhibitors four times, the m6A IP portion was eluted with elution buffer (5 mM Tris-HCL pH 7.5, 1 mM EDTA pH 8.0, 0.05% SDS, and 4.2 μl Proteinase K (20 mg/ml)). The final eluted RNA was concentrated with an RNA Clean & Concentrator-5 kit (Zymo Research, Irvine, CA). The same amount of the concentrated IP RNA or input RNA from each sample was used for the cDNA library. The RNA expression was determined by the number of amplification cycles (Cq). The relative m6A levels in genes were calculated by normalizing the m6A levels (m6A IP) against the expression of each gene (Input).

## Endosome immunoprecipitation (Endo-IP)

Endo-IP was performed as described previously[45]. Cells were harvested on ice by scraping in 2 ml DPBS and pelleted by centrifugation at 1000 x $g$ for 2 min at 4 °C. Supernatant was discarded, and cell pellet was washed once with 1 ml of KPBS buffer (25 mM KCl, 100 mM potassium phosphate, pH 7.2), followed by centrifugation at 1000xg for 2 min at 4 °C. Cells were then resuspended in 500 μL of KPBS supplemented with the protease inhibitors (Thermo Fisher, 78442) and lysed with 30 strokes with a 2 ml Dounce homogenizer on ice.

Lysed cells were then subject to centrifugation at 1000xg for 5 min at 4 °C, and the post-nuclear supernatants (PNS) were transferred to new tubes on ice. The α-FLAG M2 magnetic beads were washed with 1 mL KPBS buffer with the protease inhibitors and resuspend in the same buffer. Resuspended bead slurry was added to each PNS and incubated at 4 °C for 1 h with gentle rotation. Beads were eluted by addition of 120 μL 0.5% NP-40 in KBPS with the protease inhibitors and incubated for 30 min at 4 °C with gentle rotation, followed by immunoblot analysis.

## RNA immunoprecipitation (RIP)

The interaction between protein and RNA was assessed using the Magna RIP RNA-Binding Protein Immunoprecipitation Kit (Millipore Co., Burlington, MA), according to the manufacturer's instructions. Briefly, specific antibodies attached to magnetic beads were incubated with cell lysates. Then, the protein-RNA complexes were isolated, and the RNAs were extracted by the phenol-chloroform RNA extraction method. The relative interaction between the target proteins and RNAs was analyzed by qPCR with normalization against the corresponding input.

## Isolation of proteins and RNAs in endosomes

Endosomes were isolated using the Minute™ Endosome Isolation and Cell Fractionation Kit (Invent biotech, Plymouth, MN), according to the manufacturer's instructions. Briefly, the final isolated pellets were lysed with 1X Cell Lysis Buffer (Cat #9803, Cell Signaling, Danvers, MA) containing Invitrogen™ SUPERase•In™ RNase Inhibitor (Invitrogen, Carlsbad, CA). For RNA isolation from endosomes, TRizol or an RNA Clean & Concentrator-5 kit (Zymo Research, Irvine, CA) was used according to the manufacturer's instructions. The relative levels of target RNAs in endosomes were normalized against HPRT1 mRNA, 18S rRNA, or the total U6 snRNA in the input samples.

## Isolation of the nuclear and cytoplasmic RNAs or proteins

The nuclear, cytoplasmic, or total RNAs were isolated using the RNA Subcellular Isolation Kit (Active Motif, Carlsbad, CA) according to the manufacturer's instructions. The relative RNA levels were measured by qPCR analysis. The nuclear and cytoplasmic proteins were isolated using the NE-PER™ Nuclear and Cytoplasmic Extraction Reagents kit (Thermo Scientific, 78833) according to the manufacturer's instructions, followed by immunoblotting analysis.

## Biotin labeling and streptavidin immunoprecipitation (IP)

tRNA (Sigma-Aldrich, St. Louis, MO), U6 (IDT, Coralville, IA), m6A43 U6 (IDT, Coralville, IA), or Poly I:C (Sigma-Aldrich, St. Louis, MO) were labeled with an RNA 3' End Biotinylation Kit (Thermo Fisher Scientific, Waltham, MA). The quality and quantity of the biotin-labeled RNAs were confirmed using the Chemiluminescent Nucleic Acid Detection Module Kit (Thermo Fisher Scientific, Waltham, MA) according to the manufacturer's instructions. Streptavidin-coated magnetic beads, Dynabeads™ MyOne™ Streptavidin C1 (Thermo Fisher Scientific, Waltham, MA), were incubated with the prepared IP samples, including biotin-labeled target RNAs with or without recombinant proteins such as human YTHDF2 (Active Motif, Carlsbad, CA) or human TLR3 (Abcam, Waltham, MA). 1X Cell Lysis Buffer (Cat# 9803, Cell Signaling, Danvers, MA) was used as an IP reaction buffer for cell lysates or recombinant proteins. The relative binding affinity between biotin-labeled RNAs and target proteins was measured by immunoblotting.

## Analysis of YTHDF2-interacting proteins by Mass Spectrometry

All experiments were conducted using established cell lines and immunoprecipitation assays; ethical approval was not applicable.

To characterize proteins associated with YTHDF2, immunoprecipitates were analyzed by mass spectrometry. A total of three samples were submitted: one IgG control and two YTHDF2 immunoprecipitates

(treated and non-treated). Sample preparation, LC–MS/MS acquisition, and data processing were performed by MS Bioworks (MS Bioworks LLC, Ann Arbor, MI).

For sample preparation, half of each immunoprecipitate was subjected to SDS-PAGE using a 10% Bis-Tris NuPAGE gel (Invitrogen) with the MES buffer system. Gels were run to ~2 cm, and the corresponding region was excised into ten equal slices. Gel slices were processed for in-gel digestion on a ProGest robot (DigiLab) using the following protocol: washing with 25 mM ammonium bicarbonate and acetonitrile, reduction with 10 mM dithiothreitol at 60 °C, alkylation with 50 mM iodoacetamide at room temperature, digestion with sequencing-grade trypsin (Promega) at 37 °C for 4 h, and quenching with formic acid. Supernatants were analyzed directly without further cleanup.

Half of each digested sample was analyzed by nanoLC-MS/MS using a Waters M-Class HPLC system coupled to a Thermo Fisher Orbitrap Fusion Lumos mass spectrometer. Peptides were first loaded onto a trapping column and then separated on a 75 μm analytical column packed with Luna C18 resin (Phenomenex) at 350 nL/min. Elution was performed with a linear gradient of acetonitrile in 0.1% formic acid. The mass spectrometer was operated in data-dependent acquisition mode, with full MS scans acquired at 60,000 FWHM and MS/MS scans at 15,000 FWHM, using a 3 s cycle time.

Data analysis was performed using Mascot (Matrix Science) against the SwissProt Human database (concatenated forward and reverse sequences plus common contaminants). Search parameters were as follows: enzyme specificity set to trypsin/P with up to two missed cleavages; carbamidomethylation of cysteine as a fixed modification; oxidation of methionine, N-terminal acetylation, pyroglutamate formation (N-terminal glutamine), deamidation (N,Q), and phosphorylation (S,T,Y) as variable modifications; monoisotopic mass values; precursor ion tolerance of 10 ppm; and fragment ion tolerance of 0.02 Da. Mascot DAT files were parsed into Scaffold (Proteome Software) for validation and filtering. Data were filtered at 1% peptide and protein false discovery rate (FDR), requiring at least two unique peptides per protein

### Generation of the anti-YTHDF2 pS39 antibody

The anti-YTHDF2 pS39 antibody was generated by Abclonal Technology. One-and-a-half-year-old New Zealand rabbits (2.5 kg), housed under SPF conditions, were subcutaneously injected with 700 μg of a modified antigen peptide (EPYL(S-p)PQAR-C-KLH) emulsified with Complete Freund's Adjuvant (CFA) for the primary immunization, followed by five booster injections of 350 μg of the same peptide emulsified with Incomplete Freund's Adjuvant (IFA) at 1-, 2-, and 3-week intervals. Terminal bleeds were collected after the final immunization. Polyclonal antibodies were purified from the terminal bleeds by antigen affinity chromatography using a column conjugated with the modified peptide (EPYL(S-p)PQAR-C), followed by depletion using a column conjugated with the non-modified peptide (EPYLSPQAR-C).

### Pathway and Venn diagram analysis

Gene ontology (GO) enrichment and Kyoto Encyclopedia of Genes and Genomes (KEGG) pathway enrichment analyses were performed using Metascape (https://metascape.org/)[82], a web-based portal designed to provide a comprehensive gene list annotation and analysis resource, as described previously[78], using the default background gene list (all genes) provided by Metascape for initial pathway screening. Quantitative Venn diagrams were generated using the following web tool: https://www.deepvenn.com/[83]. Genes with an adjusted *p* value less than 0.05 by DESeq2 were used for pathway analysis and Venn diagram analysis.

### Migration assay

Migration assays were performed as described previously[81]. Briefly, $5 \times 10^4$ cells were suspended in 150 μl of serum-free medium and seeded onto 8-mm Pore Transwell Inserts (Corning, Corning, NY) for the migration assay. The lower chamber was filled with 900 μl of complete medium. Cells on the Transwell Inserts were then fixed with 4% paraformaldehyde/PBS for 30 min. Subsequently, fixed cells were stained with hematoxylin solution (Sigma-Aldrich, St. Louis, MO) for 1 h. After wiping off cells on the upper side of the filter on the Transwell Inserts using cotton swabs, microphotograms of the cells migrated onto the lower side of the filter were taken using light microscopy. Cells migrated onto the lower side of the filter were manually counted from the microphotograms. Mean cell numbers were quantified from about ten randomly selected squares (500 μm X 500 μm) per Transwell insert.

### Cell proliferation assay

Cell proliferation was assessed using the Cell Counting Kit-8 (CCK-8; Sigma-Aldrich, St. Louis, MO) according to the manufacturer's instructions, with minor modifications. Briefly, cells were seeded in 24-well plates and incubated with CCK-8 solution for 1 h. After incubation, the culture medium containing the CCK-8 reagent was transferred to 96-well plates, and absorbance was measured at 450 nm using a microplate reader. The use of 24-well plates, instead of the standard 96-well format recommended in the kit manual, helped minimize several technical issues encountered during long-term culture, including high cell density effects, difficulties in handling very small volumes, and uneven medium evaporation at the plate edges. In addition, the larger well format reduced the risk of cell detachment during medium changes, which can easily occur in smaller wells due to suction effects[78,81].

### Analysis of public RNA-Seq data sets for autoimmune diseases

RNA Seq or microarray data used to assess the role of YTHDF2 in autoimmune diseases were originally found using the ADEx: Autoimmune Diseases Explorer database (https://adex.genyo.es). Data files were accessed and downloaded using the Gene Expression Omnibus (GEO). Data was imported into Prism for statistical analyses. Datasets analyzed were GSE110169_SLE (microarray), GSE50772 (microarray), GSE72509 (RNA-Seq), GSE11907_T1D (microarray), GSE110169 RA (microarray), GSE56649 (microarray), GSE45291 (microarray), GSE89408 (RNA-seq), GSE104174 (RNA-Seq), GSE124073 (RNA-Seq), and GSE51092 (microarray).

### Statistics and reproducibility

Statistical analyses were performed using Prism v10 (GraphPad Software). Data are presented as the mean of at least three independent experiments and were analyzed using a two-tailed unpaired Student's *t*-test or a Mann–Whitney U test, as indicated. Error bars indicate the SDs or SEs of the means as specified. $P < 0.05$ was considered statistically significant. All experiments were independently repeated at least three times with similar results. Representative images (e.g., micrographs, blots) are shown from experiments that were repeated independently with comparable outcomes. Sample sizes (n) refer to biologically independent replicates (e.g., independent mice, cell cultures, or tissue samples) unless otherwise indicated.

### Reporting summary

Further information on research design is available in the Nature Portfolio Reporting Summary linked to this article.

## Data availability

The data supporting the findings of this study are available from the corresponding authors upon request. m[6]A IP sequencing and RNA sequencing data have been deposited in the GEO repository under accession code GSE145925. Mass spectrometry data have been deposited in the PRIDE repository under accession code PXD059422. Source data are provided with this paper. For some qPCR data, raw

data points are unavailable because they were acquired in 2018 on the Bio-Rad CFX Connect Real-Time System and processed with CFX Manager software v3.1 (Bio-Rad), which at that time retained only summary statistics; therefore, only summary values are shown and available for Figs. 1H–K, 1M–Q, 1S–U; 2G–I, 2P; 3A–C, 3N–W; 4A–G; and Supplementary Figs. 6B–C, 8A, 10D, 11D–E, 12A–J, 16J, and 16L. No restrictions apply to accessing the deposited sequencing or proteomics datasets, and they are publicly available. Source data are provided with this paper.

## Code availability

All the software and packages used in this article have been listed in the Reporting Summary and are publicly available.

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

## Acknowledgements

We thank the Human Tissue Resource Center (HTRC, RRID: SCR_019199), the Cytometry and Antibody Technology core facility (CAT, RRID: SCR_017760), and the Integrated Light Microscopy Core (RRID: SCR_019197) at the University of Chicago for their assistance. We thank Dr. Masaaki Komatsu for providing the Atg7flox/flox and p62flox/flox mice, Dr. Noboru Mizushima for providing the Atg5flox/flox mice, Dr. Norbert Fusenig for providing the HaCaT cells, and Dr. Ann Motten for a critical reading of the manuscript. The schematic was created with BioRender.com. This work was supported in part by NIH grants ES031693 (Y.Y.H.), ES024373 (Y.Y.H.), the University of Chicago Comprehensive Cancer Center P30 CA014599, the CTSA (UL1 TR000430), the CACHET (ES027792), and the University of Chicago Friends of Dermatology Endowment Fund. C.H. is an investigator of the Howard Hughes Medical Institute (HHMI).

## Author contributions

Y.Y.H. and S.Y. conceived the project and designed the original studies. S.Y., Y.H.C., and H.L. performed most of the experiments with the help from M.V., T.N., and L.L. Supervised by C.H., J.W. performed the $m^6A$ IP seq experiment and participated in discussions, and X.C. performed most of the bioinformatics analysis with the input from J.W., S.Y., C.H., and Y.Y.H. E.W. and H.L. performed the data analysis of autoimmune diseases. G.P. performed the flow cytometric analysis of immune cells from mice. M.S. generated mutant YTHDF2 and TLR3 plasmids. T.Y.R., J.C., and M.B. participated in discussions. S.Y., Y.H.C., H.L., and Y.Y.H. wrote the manuscript with the input from J.W., M.V., E.W., M.S., J.C., M.B., and C.H. All authors approved the final manuscript.

## Competing interests

C.H. is a scientific founder, a member of the scientific advisory board and equity holder of Aferna Bio and Ellis Bio, a scientific cofounder and equity holder of Accent Therapeutics, and a member of the scientific advisory board of Rona Therapeutics and Element Biosciences. The other authors declare no competing interests.
