## [Transparent Peer Review file · Nature Communications]

YTHDF2 regulates self non-coding RNA metabolism to control inflammation and tumorigenesis

Corresponding Author: Dr Yu-Ying He

Version 0:

Reviewer comments:

Reviewer #1

(Remarks to the Author)

In this study, Yang et al., investigated the function of YTHDF2, a m6A reader protein, in regulating U6 snRNA stability and in skin inflammation and tumorigenesis. Using multiple mouse models and cultured human cells, they have presented an extensive study on the YTHDF2-U6-TLR3 axis for its role in inducing inflammation and promoting tumorigenesis. They have examined the regulatory effect of YTHDF2 on U6 levels and downstream induction of inflammatory genes upon UVB irradiation, the interaction between YTHDF2 and m6A modified U6 and subsequently the binding of U6 by TLR3 in endosomes, the autophagy-mediated mechanism of YTHDF2 degradation upon UVB irradiation and the role of YTHDF2 and TLR3 in UVB-induced skin cancer. The experiments, esp. the ones involved in cultured cells and molecular studies, are generally well controlled and nicely executed. However, due to the ambitious scope of this study, many details, esp. the ones involved in animal studies, are unclear. They also went back and forth in many different cell lines w/o offering convincing rationale how each different cell line provides important and unique insights into the YTHDF2-U6-TLR3 biology. I would encourage the authors carefully identifying the core advance they wish to bring to the field and re-build their studies around the knowledge. Specific comments are provided below:

1. In Fig. 1, they provided evidence that DF2 suppresses inflammatory gene expression based on cultured human cell lines. They also used DF2 skin cKO model (Fig. 1C-E) to provide important insights into how the loss of DF2 leads to epidermal hyperplasia and causes immune cell changes. However, all detailed studies of gene expression changes in epithelial cells were performed in cultured human cells and no mouse KO keratinocytes were used. It would be important to perform RNAseq experiments or at least qPCR experiments to show that similar genes are altered in mouse keratinocytes as observed in cultured human cells. In addition, some general information for KO mouse should be provided. For example, what may be the mechanism of epidermal hyperplasia upon DF2 KO w/o UVB radiation? When was the analysis done in Fig. 1C?

2. In Fig. 2, they examined the binding of m6A modified U6 by DF2 and the upregulation of U6 upon UVB irradiation upon DF2 KD/KO in cultured cells and mouse skin. They should perform U6 in situ in these cells and mouse skin to provide 1) orthogonal validation of increased U6 levels; and 2) visualization of perturbed U6 localization to endosome as they showed later. In Fig. 2O, they used ActD to inhibit U6 transcription and examined its half life in various conditions. ActD is not a very specific inhibitor to RNA Pol III, and it seems the measured half-life of U6 is ~4 hours, much shorter than the general recognized half-life of ~24 hours. They should comment on these issues.

3. In Fig. 3, they showed that KD U6 RNA by ~60% strongly suppressed DF2 downregulation-mediated inflammatory gene expression. However, since U6 is elevated upon DF2 KD and they later showed that endosomal U6 interacts with TLR3 to activate the downstream gene expression changes, they should 1) quantify the downregulation of U6 in DF2 KD cells by siU6; 2) use U6 in situ to determine the extent to which endosomal U6 is reduced. Here are also examples they used HaCaT, A431 and HeLa cells for experiments. They should clearly state the rationale of using these different cell lines and identify these cells in text and/or figure legends.

4. In Fig. 4, they examined the interaction between DF2 and U6 and the mechanism of U6 binding to TLR3. In key figures, such as Fig. 4H, K, L and P, quantification of binding will be useful.

5. In Fig. 5, they showed the co-localization of U6 and DF2 in endosome upon UVB irradiation. As I commented before, in situ hybridization of U6, ideally with endosomal markers, will be important to confirm U6 localization.

6. In Fig. 6, they studied CCR4-NOT1 mediated U6 binding and degradation. Although U6 can be adenylated, it is generally

mutually exclusive from uridylation and, more importantly, adenylated U6 appears to be a very small fraction. Furthermore, previous studies, showing DF2-mediated mRNA decay through CCR4-NOT1, all focus on polyA containing mRNAs. Thus, it is important to show more convincing data for how U6 RNA, upon DF2 binding, is degraded.

7. In Fig. 7, they used the DF2 KO mouse model and cultured human cells to show DF2 suppressing tumorigenesis. First, UVB irradiation, in contrast to the DMBA/TPA carcinogenesis model, is not a widely used tumorigenesis model for skin cancer esp. for developing into advanced, invasive tumor. While UVB causes mutations, the mutational spectrum is not well characterized. W/o further boosting, it takes a long time for tumor, but not benign papilloma, to develop. They only used a few graphs (Fig. 7F-I) to show the number of "tumors" present in their WT and DF2 KO mice. At minimum, some pathological images, such as morphology, simple IF characterization, should be provided to demonstrate whether these tumors are benign papilloma or SCC.

Reviewer #2

(Remarks to the Author)

Yang et al. demonstrate that YTHDF2 plays an immunosuppressive role by degrading m6A-modified U6 snRNA and consequently regulating TLR3-mediated pro-inflammatory signaling. The authors first show that YTHDF2 knockdown (KD) leads to increased inflammatory gene expression especially in the context of UVB irradiation. Interestingly, KD of U6 snRNA attenuates the inflammatory phenotype in YTHDF2 KD models, and KD of METTL16 reduces U6 snRNA binding to YTHDF2, suggesting that YTHDF2 binds m6A-modified U6 snRNA to regulate inflammation. Yang et al. then performed a KD screen of RNA sensors and found that TLR3 KD greatly reduced inflammatory gene expression in the context of YTHDF2 KD, suggesting that YTHDF2 inhibits TLR3 signaling to regulate inflammation. Pulldown assays confirmed TLR3 binds to U6 snRNA in vitro and in vivo. Yang et al. then demonstrate that both YTHDF2 and U6 snRNA enter endosomes, where U6 snRNA may activate TLR3 signaling. They further show that degradation of m6A-modified U6 snRNA requires phosphorylation of S39 on YTHDF2 to recruit CNOT1. Conversely, UVB radiation may inhibit YTHDF2 by either inducing dephosphorylation of YTHDF2 via MYPT1 or by targeting YTHDF2 for autophagy in a p62-dependent manner. Importantly, in murine tumor models, Yang et al. show that YTHDF2 KD in tumor cells enhances tumor growth. Silencing U6 snRNA and TLR3 attenuates tumor growth in YTHDF2 KD settings, suggesting that YTHDF2 may inhibit tumor growth by suppressing the U6-TLR3 signaling axis which otherwise supports tumor growth. Overall, the authors provide compelling evidence for their proposed model of YTHDF2/U6/TLR3-regulation of inflammation, although there may be other non-mutually exclusive mechanisms. We recommend the manuscript to be accepted with moderate revisions to clarify their model (consider alternative models) and also clarify some aspects of their computational analyses.

Major comments:

Comment #1:

Previously, deposition of m6A on polyadenylated RNAs by METTL3 has been shown to suppress immunostimulatory dsRNA levels (PMID: 32497523, PMID: 37548590). When METTL16 is downregulated, could the decrease in total m6A deposition lead to overall increased dsRNA levels? METTL16 could be controlling inflammation by controlling U6 RNA as the authors propose, but there may be additional mechanisms by which METTL16 controls inflammation. It would be great to perform dsRNA staining in METTL16 knockdown cells with or without U6 snRNA knockdown to see if METTL16 knockdown significantly alters total dsRNA levels and if U6 snRNA comprise a significant fraction of dsRNA levels using anti-dsRNA antibodies as previously shown.

Comment #2:

The authors propose that, when YTHDF2 is deficient, over-abundant U6 snRNA activates TLR3. However, is it possible that disruption of the splicing machinery (either by YTHDF2 or U6 snRNA depletion) leads to abnormal accumulation of various RNAs that activate TLR3 (or other TLRs) instead of just U6 snRNA directly activating TLR3? Could a splicing-inhibiting drug help distinguish between these two possibilities?

Comment #3:

Can SIDT2 transport the U6-YTHDF2 complex altogether, or does U6 and YTHDF2 enter endosomes via independent mechanisms? Is there a way to experimentally delineate this? If not, the author should at least discuss the relative possibilities between these two mechanisms.

Comment #4:

Since UVB can also induce DNA damage, could a DNA sensing mechanism be involved in upregulating inflammation following UVB treatment? For example, could activation of TLR9 or cGAS-STING also contribute to the observed upregulation in inflammation?

Comment #5:

The role of MYPT1 is confusing. I think the model is that UVB-irradiated YTHDF2 interacts with MYPT1, which dephosphorylates YTHDF2 to prevent it from binding CNOT1. To prove MYPT1 plays a role in regulating U6 snRNA stability, the authors need to downregulate or overexpress MYPT1 and evaluating its effect on U6 snRNA levels and stability. Otherwise, the interpretation of MYPT1's role should be toned down.

Comment #6:

In Figure 6C-D, it is clear that phosphomimetic YTHDF2 binds MYPT1 and CNOT1, and Supplementary Figure S15E shows that UVB radiation promotes binding of YTHDF2 to MYPT1. However, it is unclear how YTHDF2 preferentially interacts with MYPT1 upon UVB treatment. Supplementary Figure S15E suggests that MYPT1 expression levels do not change with UVB treatment, but what about CNOT1 expression levels? If there is no difference in MYPT1 and CNOT1 expression levels between sham and UVB conditions, the authors should at least discuss potential mechanisms through which cells respond to UVB radiation to promote MYPT1-YTHDF2 interaction over CNOT1-YTHDF2 interaction.

Comment #7:

In Figure 5, the authors show that YTHDF2 and U6 snRNA enter endosomes. However, the functional significance of this as it pertains to their proposed model is unclear (especially with regards to YTHDF2). The authors should confirm that U6 snRNA entry into the endosome is required for upregulation of inflammatory genes and whether or not YTHDF2 entry into the endosome affects inflammatory gene expression.

Comments on computational analysis: #8 ~ #11

Comment #8:

The details behind the bulk RNA-seq analyses (Figures 1A-B, 2E, S1A-B, S1D, and S7A-C) need to be clarified in the Methods (there is currently no mention of how expression differences and their statistical significances were obtained per gene). For example, which tools were used to calculate differential expression? How were lowly expressed genes filtered out before testing for differential expression? How were thresholds for differential expression determined? In Figures 1A-B, 2E, S1A-B, there also appears to be filtering thresholds for how genes were selected for pathway analyses, but this is not clearly indicated in the figures. If there are indeed filtering thresholds, they should be clearly indicated as such in the figure legends or the Methods. Additionally, how the top genes were selected in Supplementary Figures S7B-C should be clearly indicated. For example, were the genes sorted by expression or by significance?

Comment #9:

For all bulk RNA-seq analyses and pathway analyses (Figures 1A-B, 2E, S1A-B, and S7A-C), it appears that the authors did not calculate adjusted p-values for multiple hypothesis testing as all significant results are reported as $p < 0.05$, not FDR, q , or adjusted $p < 0.05$. If the authors did perform multiple hypothesis testing, the adjustment method should be reported in figure legends and the Methods, and measures of statistical significance should be appropriately named in figure axes. If the authors did not perform multiple hypothesis testing, the analyses should be redone taking adjusted p-values into account when selecting for genes as input into Metascape analysis as well as reporting significantly enriched pathways from Metascape. It is well-known that multiple hypothesis testing is critical for reporting accurate results from expression and pathway analyses derived from RNA-seq data as it greatly reduces the frequency of false positives generated by these analyses.

Comment #10:

The analyses in Figure 1F and Supplementary Figure S5: Firstly, there is no mention of any normalization methods used in the Methods or the figure legends, and the y-axis of these box plots read "YTHDF2 level", which is not a specific quantification of RNA-seq data. Secondly, t-tests were used to determine statistical significances of expression differences. It is well-known that RNA-seq data follows a negative binomial distribution, not a normal distribution (which is assumed by t-tests). Thirdly, the p-values are not adjusted for multiple hypothesis testing, which is critical for accurately reporting statistical significance from RNA-seq data. I recommend the authors perform differential expression analysis on each dataset and report the results for YTHDF2 obtained from those analyses. If the appropriate analytical pipelines differ between datasets, the authors should also clearly report these different pipelines in the Methods. For example, GSE45291 was done on a microarray, whereas GSE51092 was done via RNA-seq. Additionally, the text currently states that all public datasets examined from ADEx are from RNA-seq datasets. Since the pool of datasets represents both RNA-seq and microarray datasets, it should be specified which dataset comes from which type of analysis as microarray data has been shown to be less accurate than RNA-seq data.

Comment #11:

For all pathway analyses (Fig 1A-B, 2E, S1A-B), the methods need to be clarified on how pathway analysis was conducted in Metascape. For example, what type of defaults (background gene lists) were used? This would help readers interpret the results of the pathway enrichment analysis as the background gene list has been shown to heavily influence the statistical significance of results by inflating false positives if the background list includes far more genes than is actually sequenced (10.1038/s41596-018-0103-9).

Minor comments:

Comment #1:

For all qPCR analyses, the housekeeping gene used as controls are not listed in the legends. The Methods specify that either GAPDH, ACTB, or HPRT1 were used throughout the paper, but the figure legends should specifically specify which gene was used for which figure panel.

Comment #2:

The authors should elaborate on why inflammation decreased in YTHDF2-depleted cells following UVB treatment for 24 hours compared to 6 hours.

Comment #3:

In Figure S1A, the title states that these pathways are “Among the top 30 enriched pathways”, but 4 out of 6 pathways indicated in Figure 1A, (hsa05200_Pathways in cancer, R-HAS-449147_Signaling by Interleukins, hsa04657_IL-17 signaling pathway, and R-HAS-168164_Toll Like Receptor 3 (TLR3) Cascade) are not found in Supplementary Figure S1A, which represents the full list of top 30 enriched pathways. Can the authors explain this discrepancy?

Comment #4:

Figure 3F is too confusing to read. The authors should highlight only the relevant comparisons made to back-up the claims stated in the main text on page 10.

Comment #5:

There are a few typos in the figures and figure legends:

- Figure 4J: the figure legend reads “YTHDF2-Flag overexpression.” Shouldn't this read “TLR3-Flag overexpression”?
- Figure 6F: the x-axis reads “IP:CONT1” when it should read “IP: CNOT1”.
- Supplementary Figure S1A: there is no label for the x-axis.
- Supplementary Figure S16G: there is no explanation for the color key in the figure legend.

Comment #6:

Dynasore inhibits dynamin, which is localized to the plasma membrane to facilitate endocytosis, and the authors show that dynasore may reduce U6 snRNA stability. However, the mechanism behind this is unclear, especially as to how U6 snRNA packaging into endosomes appears to be dynamin-dependent. Although the authors show YTHDF2 is not secreted and taken up by other cells in Figure 5O, is it possible that U6 snRNA is secreted and taken up by other cells?

Comment #7:

In Figure 5K, inhibition of SIDT2 blocks YTHDF2 entry into endosomes. How does inhibiting an RNA transporter also prevent YTHDF2 from entering endosome?

Comment #8:

The authors should elaborate on the identity of the unmarked bands above 25 kDa in Figure 5N. Are these bands a result of nonspecific antibody binding? If so, please clarify in the legend.

Comment #9:

The authors should describe in the Methods how they generated their custom antibody for phosphorylated YTHDF2 at S39 (Figure 6B).

Comment #10:

In Figure 4K: To clarify the molecular binding dynamics between TLR3, YTHDF2, and U6 snRNA, a better method is to perform electrophoretic mobility shift assays to obtain dissociation constants (Kd) for interactions between U6-TLR3 vs. U6-YTHDF2.

Reviewer #3

(Remarks to the Author)

Reviewer #4

(Remarks to the Author)

Yang et al NCOMMS-24-83544

The authors describe truly novel findings. The RNA modification m6A is shown to be present on a variety of RNA molecules. The m6A reader protein YTHDF2 is shown to bind modified mRNAs and mediate their decay via recruitment of the deadenylase CCR4-NOT1 complex. In this study, the authors demonstrate that YTHDF2 binds the spliceosomal U6 snRNA. This is new, as such an interaction was never demonstrated before. They also show that this interaction regulates RNA levels of U6 snRNA where YTHDF2 interacts with deadenylase component CNOT1, which may mediate decay of m6A-modified U6 snRNA. The authors describe the role of m6A reader YTHDF2 in the inflammatory response through its regulation of the U6 snRNA. They provide data from several different cell lines and a skin-specific conditional deletion mouse model. The study provides evidence that human YTHDF2 binds to and regulates the stability of U6, modulating the expression of inflammatory regulators via TLR3. Finally, they show that UVB irradiation leads to autophagy-mediated downregulation of YTHDF2, and that deletion of YTHDF2 in mouse skin cells increases sensitivity to UVB radiation. Overall, the findings are novel, unexpected and relevant for the RNA modification field.

General comments

1. Analysis of gene expression changes and changes in m6A (specifically changes in inflammatory genes, changes in U6

levels, U6 m6A status etc.) from skin of cKO YTHDF2 mice would have added a lot to the hypothesis that YTHDF2 regulation of U6 is responsible for histological observations

2. In the text, it is often not stated which cell line, or even system, is being referred to or studied

a. E.g. in the abstract the authors do not state that they studied YTHDF2 in human cells, and “skin-specific deletion” does not mention the mouse model

b. Several different cell lines (HaCaT, HeLa, A431, 293T, NHEK, CHL-1) are used in the study, but the text rarely states which result was obtained from which cell line, or the rationale for doing particular experiments in a particular cell line

i. Text should refer to cell lines, e.g. “Our RNA-seq analysis showed that YTHDF2 knockdown...” should add “in HaCaT cells”

ii. Figure panels should also show clearly which cell line the result comes from. This is done for some panels (Fig 5C, 5G etc.), but most are unlabelled

iii. Some figure legends do not mention cell line used

1. E.g. Fig. 5D. Panel includes label 293T, but not stated in figure legend – “D. Endo-IP analysis of YTHDF2 protein levels in endosomes.”

2. Fig. 5K. No label in panel, and no mention of cell line in figure legend. Also error in figure legend “K. Immunoblot analysis of YTHDF2 in whole cell lysates (WCL) and endosomes in cells as in K.”

3. Fig. 4M. No label in panel, no cell line mentioned in legend.

3. Most figures contain many panels showing qPCR results for different mRNAs in different conditions. It is sometimes unclear why certain mRNAs were or were not tested under specific conditions, and in the different cell lines used. Clearer labelling in figures and some explanation of the rationale in the text would be useful.

4. Different durations of UVB exposure are used for different experiments with different cell lines – justification in the text would be helpful

5. METTL16 section in introduction should refer to *C. elegans* METTL16 ortholog, and cite Mendel et al 2021

a. This paper confirms *mett-10* as the *C. elegans*, and also shows a slight increase in U6 snRNA levels upon *mett-10* deletion, which is relevant to this study. It is interesting in this context, as there is no m6A reader protein (like YTHDF2) in worms.

b. In addition, Warda et al., 2017, is cited, but this paper showed no change to U6 snRNA levels upon METTL16 knockdown, in contrast to the result presented here. This should be mentioned.

6. “In parallel, we also observed the U6-dependent YTHDF2-TLR3 interaction, which may be formed either as an intermediate to remove TLR3 from binding to m6A U6 or as an independent complex to maintain m6A U6-bound TLR3 in an inactive state.”

a. There is no reference to a figure, but I assume this refers to figure 4L, which shows pull-down of TLR3 and YTHDF2 with m6A U6, but not necessarily a YTHDF2-TLR3 interaction

7. The authors present evidence that degradation of the U6 snRNA may be mediated by the CCR4-NOT deadenylase component CNOT1, but this is limited to interaction between CNOT1 and U6 snRNA by RIP and increased stability of U6 snRNA upon CNOT1 knockdown in a single cell line (HeLa).

a. This is surprising, as U6 is not polyadenylated. In the discussion, the authors point out that the 3' tail of U6 can be adenylated, but as far as I know this adenylation has not been implicated in U6 decay. Perhaps the authors could include some more thoughts on how the mRNA deadenylase complex could regulate U6 snRNA decay.

Specific comments

Page 6 – Acronym SLE is not explained in the text

Page 8 – “YTHDF2 knockdown slightly increased m6A enrichment in the 5'UTR and 3'UTR...” Fig. 2C shows no change (-0.03%) in proportion of m6A in 3'UTR and extremely small (+0.08%) increase in 5'UTR, compared with a 0.4% decrease in the CDS upon shDF2

Page 10 and Fig. 3F – “While both U6 and m6A U6 induced IL-6 expression, U6 lead to a higher induction than m6A U6 (Fig. 3F).” Fig. 3F shows non-significant change to IL-6 levels with m6A U6

Page 11 – “Indeed, a small-scale siRNA screening for RNA sensors, including TLR3, TLR7, TLR8, MDA5, and RIG-I (Akira et al., 2006; Fitzgerald and Kagan, 2020)...” This reads as if the cited publications contain the small-scale siRNA screen

Page 12 – Many references to U6 snRNA “binding to” proteins (TLR3, YTHDF2) rather than being bound by them

Page 13 – “we assessed whether U6 and YTHDF2 also localize in endosomes...” – this question is asked but not answered, with the next relevant sentence being “Second, UVB irradiation increased both total level and endosomal proportion of U6 snRNA...”

Page 13 – “and confocal imaging analysis showed that YTHDF2 is also localized in endosomes (Fig. 5C-E)” It would be useful to mention the Rab7 endosomal marker in the text

Page 18 – “In addition, UVB irradiation inhibits YTHDF2 phosphorylation at S39, which is critical for YTHDF2 interaction with CNOT1 and localization in endosomes and is inhibited by UVB irradiation.” Redundancy – inhibition by UVB irradiation written twice.

Page 19 – “In addition, the interaction between YTHDF2 and U6 snRNPs...” Should be snRNP proteins or components

Fig. 3M – “However, dual knockdown of both METTL16 and YTHDF2 mimicked the effect of singular YTHDF2 knockdown (Fig. 3M-Q).” Panel M is COX-2 immunoblot and does not include YTHDF2 knockdown. Fig. 3M is not referenced elsewhere in text

Fig. 6F – Label error, “CONT1” instead of CNOT1

Fig. S16G – No mention of scoring system (3,2,1,0) for YTHDF2 levels in figure or legend

Version 1:

Reviewer comments:

Reviewer #1

(Remarks to the Author)

The revised manuscript is strengthened with new data, analysis and clarification. Here are few remaining issues that should be addressed.

1. It is a bit odd that they chose to quantify Cox2 mRNA in DF2 Het but not KO in Fig. S6D. Typically, primary keratinocytes are derived from neonatal pups, which take a few weeks to produce from Het breeding. Nevertheless, Cox2 mRNA changes are mild. Do they observe epidermal hyperproliferation in DF2 Het without or with UVB radiation? These results may add the validity for using Het for quantification. Alternatively, they could do Cox2 in situ or IF staining to demonstrate Cox2 increase upon DF2 KO in the mouse model.
2. In new Fig. S14G, they performed U6 in situ and Rab7 staining to corroborate cell culture studies. The fluorescent signals appear to be over-exposed for U6 in DF2 cKO and Rab7 in both sham and UVB conditions. They should also provide higher magnification images, comparable to their cell culture results e.g. S14H, to convincingly demonstrate co-localization of U6 and Rab7 in endosomes upon UVB radiation.

Reviewer #2

(Remarks to the Author)

Overall, the authors addressed our comments thoughtfully. We have two remaining questions on comments #5 and #7. We request more careful interpretation of data, and make suggestions on how to make more solid and clear conclusions.

Comment #5:

The role of MYPT1 is confusing. I think the model is that UVB-irradiated YTHDF2 interacts with MYPT1, which dephosphorylates YTHDF2 to prevent it from binding CNOT1. To prove MYPT1 plays a role in regulating U6 snRNA stability, the authors need to downregulate or overexpress MYPT1 and evaluating its effect on U6 snRNA levels and stability. Otherwise, the interpretation of MYPT1's role should be toned down.

Response: Thank you for the constructive comment. To determine whether MYPT1 regulates U6 snRNA stability, we assessed the effect of MYPT1 knockdown. Our new data showed that MYPT1 knockdown increased U6 snRNA levels and stability (Fig. S15F-S15G), supporting a role of MYPT1 in regulating U6 snRNA decay.

Reviewer response to rebuttal: In figures S15F and S15G, phenotypes are very weak, and only shown in Hela cells. Additionally, as pointed out by reviewer #1 it does concern me that authors interchangeably use so many cell lines. The

experiment conducted in S15F and S15G, should be repeated in other cell lines (e.g., A431) before authors conclude that MYPT1 knockdown increased U6 snRNA levels and stability, especially since the phenotype in HeLa cells is very weak.

Comment #7:

In Figure 5, the authors show that YTHDF2 and U6 snRNA enter endosomes. However, the functional significance of this as it pertains to their proposed model is unclear (especially with regards to YTHDF2). The authors should confirm that U6 snRNA entry into the endosome is required for upregulation of inflammatory genes and whether or not YTHDF2 entry into the endosome affects inflammatory gene expression.

Response: Thank you for the constructive comment. This is a great question and suggestion. To determine the importance of U6 snRNA entry into the endosome in the increased expression of inflammatory genes, we assessed the effect of Dynasore, which inhibits U6 entry into endosomes (Fig. 5G), and the role of the interaction U6 with TLR3, an endosomal RNA sensor. Our new data showed that Dynasore reduced expression of inflammatory genes induced by U6 or YTHDF2 knockdown (Fig. 5R and 5S). To further determine whether the effect of YTHDF2 inhibition on proinflammatory response is mediated by U6/TLR3, we assessed the effect of U6 knockdown, TLR3 knockdown, and the combination. Our new data showed that either U6 knockdown or TLR3 knockdown decreased IL6 expression and that the combination of double knockdown of TLR3 and U6 has similar effect to knockdown of either U6 or TLR3 singularly (Fig. 4H-4J), suggesting that the effect of U6 is mediated by the TLR3 pathway. Together with our findings that YTHDF2 endosomal entry is dependent on U6, while U6 endosomal entry is independent of YTHDF2 (Fig. 5M-O), our data demonstrate the U6 entry into endosomes is critical for upregulation of inflammatory genes.

Reviewer response to rebuttal: In Figure 5S, dynasore treatment seemed to have reduced expression of inflammatory genes in both control knockdown and DF2 knockdown cells. This suggests that U6 may enter endosomes to induce inflammation independently of DF2 knockdown. Hence, the results may not well support the author's model that DF2 knockdown leads to enhanced U6 entry into endosomes. For fig. 5S, more replicates are needed and statistical analysis between the black bars (vehicle vs. dynasore) are also needed for more conclusive interpretation. Also repeating the 5S assay in other cell lines (HeLa, HaCat) will also help. Or were the assays not tried in other cell lines because DF2 knockdown in only A431 leads to inflammation, and not in other cell types? Such clarification will help readers assess the significance of the finding, and the rationale for switching between cell lines.

Reviewer #3

(Remarks to the Author)

Reviewer #4

(Remarks to the Author)

I commend the authors for the revised version and I am happy to note the improvements made.

Version 2:

Reviewer comments:

Reviewer #2

(Remarks to the Author)

I am satisfied with the revisions made by the authors! The careful revisions have much improved the clarity and focus of the study. Thank you!

REVIEWER COMMENTS

Reviewer #1 (Remarks to the Author):

In this study, Yang et al., investigated the function of YTHDF2, a m6A reader protein, in regulating U6 snRNA stability and in skin inflammation and tumorigenesis. Using multiple mouse models and cultured human cells, they have presented an extensive study on the YTHDF2-U6-TLR3 axis for its role in inducing inflammation and promoting tumorigenesis. They have examined the regulatory effect of YTHDF2 on U6 levels and downstream induction of inflammatory genes upon UVB irradiation, the interaction between YTHDF2 and m6A modified U6 and subsequently the binding of U6 by TLR3 in endosomes, the autophagy-mediated mechanism of YTHDF2 degradation upon UVB irradiation and the role of YTHDF2 and TLR3 in UVB-induced skin cancer. The experiments, esp. the ones involved in cultured cells and molecular studies, are generally well controlled and nicely executed. However, due to the ambitious scope of this study, many details, esp. the ones involved in animal studies, are unclear. They also went back and forth in many different cell lines w/o offering convincing rationale how each different cell line provides important and unique insights into the YTHDF2-U6-TLR3 biology. I would encourage the authors carefully identifying the core advance they wish to bring to the field and re-build their studies around the knowledge. Specific comments are provided below:

Response: Thank you for taking the time to review our manuscript and provide us with insightful comments, all of which have been considered carefully in the preparation of the revision (please see our point-by-point responses below). All revisions have been highlighted with red text and submitted as supplementary information. To specifically address all the comments, we have performed a number of new experiments and added in new data (shown in Fig. S6D, S10B, S10C, S11A, S11F, S11G, 4H, 4I, 4J, S12K, S12L, S12M, 5C, 5N, 5O, 5R, 5S, S14F, S14G, S14H, S14I, S14O, 6H, 6I, 6J, S15F, S15G, S15M, S15N, S15O, S15P, S16D- immunoblots, and S17C). In addition, to address concerns in the original data analysis, we have revised the figure panels shown in Fig. 1A, 1B, S1A, and 2E (changing to log₁₀(q)), Fig. 1F and S5 (removed statistical analysis for all RNA-seq analysis), and Fig. 3F (reformatting the panels and simplify comparison group for clarity), as well as Fig. 1C and 5A (changing violin plot to bar plot). Furthermore, for conciseness and the interest of space, we have removed the original Fig. S1B and S1C, as they were redundant with the original Fig. S1D (now Fig. S1B). With these new data added, changes made, and new information and discussion added, we believe our revised manuscript has significantly improved with all the reviewers' suggestions.

1. In Fig. 1, they provided evidence that DF2 suppresses inflammatory gene expression based on cultured human cell lines. They also used DF2 skin cKO model (Fig. 1C-E) to provide important insights into how the loss of DF2 leads to epidermal hyperplasia and causes immune cell changes. However, all detailed studies of gene expression changes in epithelial cells were performed in cultured human cells and no mouse KO keratinocytes were used. It would be important to perform RNAseq experiments or at least qPCR experiments to show that similar genes are altered in mouse keratinocytes as observed in cultured human cells. In addition, some general information for KO mouse should be provided. For example, what may be the mechanism of epidermal hyperplasia upon DF2 KO w/o UVB radiation? When was the analysis done in Fig. 1C?

Response: Thank you for the constructive comments. First, to validate our findings in cultured human cells, we performed qPCR experiments to show whether similar genes are altered in mouse keratinocytes with or without YTHDF2 inhibition. Unfortunately, we could not obtain sufficient cKO mice within the time allowed for the revision (3 months), as we lost the cKO colony during transition when the first author, Dr. Yang, left the lab in 2023. Therefore, we have focused on assessing the effect of heterozygous YTHDF2 deletion in the mouse skin (DF2 cHet) on UVB-induced inflammatory gene expression. Indeed, primary keratinocytes with heterozygous YTHDF2 deletion showed increased expression of the inflammatory genes and m⁶A level in U6 snRNA as compared with their WT counterparts (Fig. S6B and S10C), consistent our observation in

cultured human cells (**Fig. 1H-1K, 1M-1Q, 2K, 2Q**). Together with our original data, our new data supporting a critical role of YTHDF2 in suppressing UVB-induced inflammation. We have added these new data (Fig. S6D and S10C) and the following statements in our revised manuscript.

“Furthermore, primary mouse keratinocytes isolated from mice with skin-specific heterozygous YTHDF2 deletion (DF2 cHet) showed increased expression of $TNF\alpha$, $IL-1\beta$, and COX-2 as compared with their WT counterparts under baseline condition or UVB stress, while IL-6 expression was not affected by either YTHDF2 inhibition or UVB stress (Supplementary Fig. S6D), likely due to the difference in genetic or molecular context between mouse primary keratinocytes and human keratinocyte cell line or NHEK cells.” (Page 7).

“...and primary mouse keratinocytes (Fig. 2K, Supplementary Fig. S10C).” (Page 9).

Second, we have added information for the KO mouse. For example, we have added discussion on the potential mechanism of epidermal hyperplasia upon DF2 KO with or without UVB irradiation. It is possible that the increase in inflammatory gene expression increased proliferation and hyperplasia in the epidermis through the production of the inflammation mediator prostaglandin E2 (PGE_2) mediated by induced COX-2 expression (Bowden, 2004). UV damage triggers a number of pathways such as the induction of COX-2 (Bowden, 2004) and proinflammatory cytokines (Kim and He, 2014) in epidermal keratinocytes. Previously we and others have demonstrated that COX-2 in keratinocytes acts as a pro-inflammatory and tumor-promoting factor in skin tumorigenesis and is upregulated in human skin cancer (Buckman et al., 1998), (Jiao et al., 2014; Qiang et al., 2017). UV-induced COX-2 up-regulation can initiate the synthesis of the principle inflammation mediator PGE_2 (Bowden, 2004) and thus increases keratinocyte proliferation and hyperplasia in the epidermis. Other cytokines such as $TNF\alpha$, IL6, and $IL1\beta$ also play critical roles by contributing to the induction of COX-2 or other genes involved in epidermal inflammation and hyperplasia. In this study, we found that YTHDF2 inhibition increases UV-induced COX-2 expression in primary keratinocytes and cultured human keratinocytes and epidermal hyperplasia, and that the COX-2 inhibitor decreased cell proliferation in cells with YTHDF2 knockdown, supporting a critical role of COX-2 in increased cell proliferation by YTHDF2 inhibition. We have added the following statements in the Discussion section.

“Our findings demonstrate that YTHDF2 inhibition enhances UVB-induced inflammation, leading to epidermal hyperplasia in mice. UV damage triggers a number of pathways such as the induction of COX-2 (Bowden, 2004) and proinflammatory cytokines (Kim and He, 2014) in epidermal keratinocytes. Previously, we and others have demonstrated that COX-2 in keratinocytes acts as a pro-inflammatory and tumor-promoting factor in skin tumorigenesis and is upregulated in human skin cancer (Buckman et al., 1998), (Jiao et al., 2014; Qiang et al., 2017). UV-induced COX-2 upregulation can initiate the synthesis of the principle inflammation mediator PGE_2 (Bowden, 2004) and thus increase keratinocyte proliferation and hyperplasia in the epidermis. Other cytokines such as $TNF\alpha$, IL6, and $IL1\beta$ also play critical roles by contributing to the induction of COX-2 and other genes involved in epidermal inflammation and hyperplasia. In this study, we found that YTHDF2 inhibition increases UV-induced COX-2 expression in primary keratinocytes and cultured human keratinocytes and epidermal hyperplasia, and that the COX-2 inhibitor decreased cell proliferation in cells with YTHDF2 knockdown, supporting a critical role of COX-2 in increased cell proliferation caused by YTHDF2 inhibition.” (**Page 27-28**).

Third, for Fig. 1C, skin samples were collected at 24 h post-the final sham or UVB irradiation. This time point was chosen based on our prior observations regarding timing of skin inflammation and hyperplasia in our previous work (Qiang et al 2017. PMID: 28933598). For short UVB treatment, mice were irradiated with UVB irradiation (100 mJ/cm^2) 3 times every other day and skin samples were collected 24 h after the final

UVB irradiation). We have added the information in the figure legend for Fig. 1C and the Method section (Page 30).

2. In Fig. 2, they examined the binding of m6A modified U6 by DF2 and the upregulation of U6 upon UVB irradiation upon DF2 KD/KO in cultured cells and mouse skin. They should perform U6 in situ in these cells and mouse skin to provide 1) orthogonal validation of increased U6 levels; and 2) visualization of perturbed U6 localization to endosome as they showed later. In Fig. 2O, they used ActD to inhibit U6 transcription and examined its half life in various conditions. ActD is not a very specific inhibitor to RNA Pol III, and it seems the measured half-life of U6 is ~4 hours, much shorter than the general recognized half-life of ~24 hours. They should comment on these issues.

Response: Thank you for the constructive comment. We have performed Fluorescence In Situ Hybridization (FISH) analysis of U6 snRNA in cells and mouse skin to orthogonally validate altered U6 levels and localization in endosomes. Our new data demonstrated that (a) UVB irradiation and/or YTHDF2 inhibition increased U6 in cells and mouse skin, (b) U6 localized to endosomes, and (c) U6 knockdown decreased U6 snRNA level in the nucleus, cytoplasm, and endosomes (Fig. S10B, 5C, S14F, S14G, S14H, S14I), which provides orthogonal validation of increased U6 levels and U6 localization in endosomes, which are the focus for Fig. 5. We have added the new data (Fig. S10B, 5C, S14F, S14G, S14H, S14I) and the following statement in our revised manuscript.

“Third, we performed RNA fluorescence in situ hybridization (FISH) of U6 in combination with immunofluorescence analysis of the endosome marker Rab7 to confirm U6 localization in endosomes. While U6 snRNA was mainly detected in the nucleus, we observed that UVB increases U6 snRNA levels in the cytoplasm and U6 colocalization with the late endosome marker Rab7 in HaCaT cells, HeLa cells, and mouse skin (Fig. 5C, Supplementary Fig. S14F-S14G), supporting U6 localization in endosomes. YTHDF2 inhibition in HaCaT cells, mouse skin, or A431 cells increased U6 snRNA levels in the nucleus and cytoplasm and increased U6 colocalization with Rab7 (Fig. 5C, Supplementary Fig. S14F-S14I), while U6 knockdown decreased them in A431 and HaCaT cells (Supplementary Fig. S14H-S14I). These orthogonal validations further suggest a role for endosomal U6 snRNA in the inflammatory response.” (Page 16).

In addition, we agree with the reviewer. Actinomycin D (ActD) is not specific for RNA Pol III, as it inhibits all three eukaryotic polymerases (I, II, and III) by binding to DNA. Therefore, the shorter half-life observed in our data suggest that the effect of ActD on other RNA polymerases also affects the U6 level and stability, as other RNA polymerases may modulate the upstream regulators for U6 transcription, maturation, and/or stability. We have added the following statements in the Discussion.

“Notably, the half-life of U6 in control cells is approximately 4 hours, substantially shorter than the ~24-hour half-life reported previously (Sauterer et al., 1988; Terns et al., 1993). It is possible that actinomycin D (ActD) is not specific for RNA Pol III, as it inhibits all three eukaryotic polymerases (I, II, and III) by binding to DNA (Bensaude, 2011). Therefore, the shorter half-life observed in our data suggests that the effect of ActD on other RNA polymerases might influence U6 levels and stability, as other RNA polymerases may modulate upstream regulators involved in U6 transcription, maturation, and/or stability.” (Page 10).

3. In Fig. 3, they showed that KD U6 RNA by ~60% strongly suppressed DF2 downregulation-mediated inflammatory gene expression. However, since U6 is elevated upon DF2 KD and they later showed that endosomal U6 interacts with TLR3 to activate the downstream gene expression changes, they should 1) quantify the downregulation of U6 in DF2 KD cells by siU6; 2) use U6 in situ to determine the extent to

which endosomal U6 is reduced. Here are also examples they used HaCaT, A431 and HeLa cells for experiments. They should clearly state the rationale of using these different cell lines and identify these cells in text and/or figure legends.

Response: Thank you for the constructive comment. To address these important questions, we have performed new experiments. Our new data showed that (1) siU6 decreases U6 in YTHDF2 KD cells by siU6 by qPCR (Fig. S11A) and (2) U6 decrease in endosome by U6 FISH (Fig. S14H and S14I). We have added the new data (Fig. S11A, S14H, and S14I) and the following statements.

“Third, we performed RNA fluorescence in situ hybridization (FISH) of U6 in combination with immunofluorescence analysis of the endosome marker Rab7 to confirm U6 localization in endosomes. While U6 snRNA was mainly detected in the nucleus, we observed that UVB increases U6 snRNA levels in the cytoplasm and U6 colocalization with the late endosome marker Rab7 in HaCaT cells, HeLa cells, and mouse skin (Fig. 5C, Supplementary Fig. S14F-S14G), supporting U6 localization in endosomes. YTHDF2 inhibition in HaCaT cells, mouse skin, or A431 cells increased U6 snRNA levels in the nucleus and cytoplasm and increased U6 colocalization with Rab7 (Fig. 5C, Supplementary Fig. S14F-S14I), while U6 knockdown decreased them in A431 and HaCaT cells (Supplementary Fig. S14H-S14I). These orthogonal validations further suggest a role for endosomal U6 snRNA in the inflammatory response.” (Page 16).

In this study, we elected to utilize different cell lines, in order to (1) ensure the robustness of the observed role of the YTHDF2/U6 axis in modulating inflammatory response and (2) achieve technical feasibility as some cell lines such as HaCaT and A431 cells are more difficult to transfect and yield lower amount of cell lysates compared with model cell lines, such as HeLa or 293T cells. The details on each cell line are provided below.

- NHEK cells are primary human keratinocytes which exhibit UVB-induced inflammatory responses.
- HaCaT cells are an immortalized keratinocyte cell line and a model for non-tumorigenic keratinocytes, which respond to UVB damage by inducing an inflammatory response that is augmented by YTHDF2 inhibition.
- A431 cells are a human epidermoid cell carcinoma cell line, in which YTHDF2 knockdown augments baseline inflammatory response, possibly due to the increased sensitivity to YTHDF2 inhibition in these tumorigenic cells.
- HeLa cells are a human cervical cancer cell line, a model cell line instrumental for biomedical research and widely used due to its efficient transfection efficiency and fast growth. We choose to use HeLa cells due to the follow reasons: (1) HeLa cells are epithelial cancer cells, (2) HeLa cells grow faster than many other cell lines and can generate larger amount of proteins and RNAs for less sensitive assays, such as endosome fractionation, (3) HeLa cells have much higher transfection efficiency, making them more feasible to study the role of gene overexpression, knockdown, or knockout, and (5) HeLa cells exhibit UVB-induced inflammatory responses.
- HEK 293T (293T) cells are an epithelial-like immortalized cell line expressing the SV40 large T antigen. Similar to HeLa cells, 293T cells exhibit higher transfection efficiency, faster growth, and enhanced protein production, making them suitable for less sensitive assays such as endosome fractionation.
- Mouse primary keratinocytes were isolated from mice with or without skin-specific heterozygous YTHDF2 knockout during the revision to validate our findings observed in cultured human cells. We have added these new data (Fig. S6D and S10C). Our new data in these cells are consistent with data from the human cell lines, validating the critical role of YTHDF2 in controlling the inflammatory response.
- CHL-1 cells are a human melanoma cell line and were used to determine whether the effect of UVB irradiation is specific to keratinocytes or applies to other cell types such as melanoma cells.

We have added the justifications for using different cell lines in the Results section.

“..., non-tumorigenic human keratinocytes...” (Page 5).

“...in HeLa cells, a cell line with high transfection efficiency, ...” (Page 11).

“To explore the role of m⁶A methylation of U6 in YTHDF2’s function in skin cancer and confirm our findings in HaCaT cells, we next assessed the consequence of METTL16 inhibition on the expression of cytokines in the human A431 skin carcinoma cells” (Page 11).

“in 293T cells in order to achieve high plasmid transfection and protein expression” (Page 16).

“...in multiple cells and primary keratinocytes including HaCaT, CHL-1 (melanoma cells), HeLa, and NHEK cells ...” (Page 19).

4. In Fig. 4, they examined the interaction between DF2 and U6 and the mechanism of U6 binding to TLR3. In key figures, such as Fig. 4H, K, L and P, quantification of binding will be useful.

Response: Thank you for the constructive comment. We have quantified these figure panels with ratio or fold changes shown under each band (now Fig. 4K, N, O, and S) (Original Fig. 4H, K, L and P).

5. In Fig. 5, they showed the co-localization of U6 and DF2 in endosome upon UVB irradiation. As I commented before, in situ hybridization of U6, ideally with endosomal markers, will be important to confirm U6 localiation.

Response: Thank you for the constructive comment. To confirm U6 localization in endosomes, we have performed RNA fluorescence in situ hybridization (FISH) of U6 in combination with immunofluorescence analysis of the endosome marker Rab7. Our new data show that while U6 snRNA is mainly detected in the nucleus, we also observed colocalization of U6 snRNA with the late endosome marker Rab7, supporting U6 localization in endosomes. These orthogonal validations further demonstrate a role for endosomal U6 snRNA in the inflammatory response. We have these new data (Fig. 5C, S14F, S14G, S14H, and S14I) and the following statements in our revised manuscript.

“Third, we performed RNA fluorescence in situ hybridization (FISH) of U6 in combination with immunofluorescence analysis of the endosome marker Rab7 to confirm U6 localization in endosomes. While U6 snRNA was mainly detected in the nucleus, we observed that UVB increases U6 snRNA levels in the cytoplasm and U6 colocalization with the late endosome marker Rab7 in HaCaT cells, HeLa cells, and mouse skin (Fig. 5C, Supplementary Fig. S14F-S14G), supporting U6 localization in endosomes. YTHDF2 inhibition in HaCaT cells, mouse skin, or A431 cells increased U6 snRNA levels in the nucleus and cytoplasm and increased U6 colocalization with Rab7 (Fig. 5C, Supplementary Fig. S14F-S14I), while U6 knockdown decreased them in A431 and HaCaT cells (Supplementary Fig. S14H-S14I). These orthogonal validations further suggest a role for endosomal U6 snRNA in the inflammatory response.” (Page 16).

6. In Fig. 6, they studied CCR4-NOT1 mediated U6 binding and degradation. Although U6 can be adenylated, it is generally mutually exclusive from uridylation and, more importantly, adenylated U6 appears to be a very small fraction. Furthermore, previous studies, showing DF2-mediated mRNA decay through CCR4-NOT1, all focus on polyA containing mRNAs. Thus, it is important to show more convincing data for how U6 RNA, upon DF2 binding, is degraded.

Response: Thank you for the constructive comment. We agree with the reviewer that adenylated U6 seems to be a very small fraction of U6 snRNA. Previous studies have shown that YTHDF2 mediates polyA-

containing mRNAs through binding with CCR4-NOT1 (Du et al., 2016). Our original data (Fig. 2, 6F, 6H, 6I) showed that both YTHDF2 and CNOT1 bind to U6 snRNA and are critical for U6 snRNA decay. To address this important question and further elucidate the mechanism of YTHDF2-mediated U6 decay, we performed new experiments to determine the role of YTHDF2 and METTL16-mediated m⁶A methylation in CNOT1 binding to U6 snRNA. Our new data showed that either YTHDF2 deletion or METTL16 knockdown reduces CNOT1 binding to U6 (Fig. 6I, 6J), supporting a critical role of YTHDF2 and U6 m⁶A methylation in CNOT1 binding to U6 snRNA. Our new data demonstrate that CNOT1 binds to U6 snRNA and mediates U6 decay, through interacting with the YTHDF2-m⁶A U6 complex. The role of U6 adenylation cannot be excluded and warrants future investigations. We have added the new data and the following statements in the Results and Discussion section.

“To elucidate the mechanism of YTHDF2-mediated U6 decay, we examined the role of YTHDF2 and METTL16-mediated m⁶A methylation in CNOT1 binding to U6 snRNA. We found that either YTHDF2 deletion or METTL16 knockdown drastically reduces CNOT1 binding to U6 in HeLa cells (Fig. 6I-6J and Supplementary Fig. S15N-S15O), supporting a critical role of YTHDF2 and U6 m⁶A methylation in CNOT1 binding to U6 snRNA. To determine whether UVB irradiation affects CNOT1 expression, we assessed the effect of UVB irradiation on CNOT1 protein levels. We found that UVB irradiation had no effect on CNOT1 protein expression in either HeLa or HaCaT cells (Supplementary Fig. S15P). These results demonstrate that CNOT1 binds to U6 snRNA and mediates U6 decay through interacting with YTHDF2 in an m⁶A-methylation dependent manner.” (Page 19).

“Previously, the 3' tail of a small fraction of U6 snRNA has been shown to be adenylated (Chen et al., 2000). Although U6 is not known to be polyadenylated, recent studies have detected m⁶A methylated U6 snRNA in poly(A)⁺ RNA in worms (Mendel et al., 2021). While this may represent remnants left after poly(A)⁺ enrichment from total RNA, it is also possible that a fraction of U6 snRNA is polyadenylated in worms and other organisms, which could regulate U6 decay upon m⁶A methylation similar to poly(A)⁺ mRNAs via the YTHDF2-CNOT1 interaction (Du et al., 2016). It is possible that adenylation and/or polyadenylation cooperates with m⁶A methylation of U6 snRNA to regulate U6 snRNA turnover. Future investigation is warranted to elucidate the detailed mechanism by which YTHDF2 and CNOT1 regulate U6 snRNA decay.” (Page 24).

7. In Fig. 7, they used the DF2 KO mouse model and cultured human cells to show DF2 suppressing tumorigenesis. First, UVB irradiation, in contrast to the DMBA/TPA carcinogenesis model, is not a widely used tumorigenesis model for skin cancer esp. for developing into advanced, invasive tumor. While UVB causes mutations, the mutational spectrum is not well characterized. W/o further boosting, it takes a long time for tumor, but not benign papilloma, to develop. They only used a few graphs (Fig. 7F-I) to show the number of “tumors” present in their WT and DF2 KO mice. At minimum, some pathological images, such as morphology, simple IF characterization, should be provided to demonstrate whether these tumors are benign papilloma or SCC.

Response: Thank you for the constructive comment. We agree with the reviewer. We did observe large tumors and squamous cell carcinoma development in the YTHDF2 cKO mice, likely due to increased inflammation. We have added pathological images, such as morphology (HE staining) and IF characterization using Keratin 14 and Keratin 10 as markers of tumor differentiation status. We have added the new data (Fig. S17C) and the following statement in our revised manuscript.

“Histological analysis showed that UVB irradiation induces skin tumor formation in both WT and DF2 cKO mice, while squamous skin carcinoma was detected in the DF2 cKO mice (Supplementary Fig. S17C). This was supported by immunofluorescence analysis that indicated expression of the basal keratinocyte marker Keratin 14 (K14) in both basal layer of mouse skin and mouse tumors, while the differentiation marker Keratin 10 (K10) was only detected in mouse skin, hair follicles, tumor from WT mice, but not in the tumor from DF2 cKO mice (Supplementary Fig. S17C).” (Page 22).

Reviewer #2 (Remarks to the Author):

Yang et al. demonstrate that YTHDF2 plays an immunosuppressive role by degrading m6A-modified U6 snRNA and consequently regulating TLR3-mediated pro-inflammatory signaling. The authors first show that YTHDF2 knockdown (KD) leads to increased inflammatory gene expression especially in the context of UVB irradiation. Interestingly, KD of U6 snRNA attenuates the inflammatory phenotype in YTHDF2 KD models, and KD of METTL16 reduces U6 snRNA binding to YTHDF2, suggesting that YTHDF2 binds m6A-modified U6 snRNA to regulate inflammation. Yang et al. then performed a KD screen of RNA sensors and found that TLR3 KD greatly reduced inflammatory gene expression in the context of YTHDF2 KD, suggesting that YTHDF2 inhibits TLR3 signaling to regulate inflammation. Pulldown assays confirmed TLR3 binds to U6 snRNA in vitro and in vivo. Yang et al. then demonstrate that both YTHDF2 and U6 snRNA enter endosomes, where U6 snRNA may activate TLR3 signaling. They further show that degradation of m6A-modified U6 snRNA requires phosphorylation of S39 on YTHDF2 to recruit CNOT1. Conversely, UVB radiation may inhibit YTHDF2 by either inducing dephosphorylation of YTHDF2 via MYPT1 or by targeting YTHDF2 for autophagy in a p62-dependent manner. Importantly, in murine tumor models, Yang et al. show that YTHDF2 KD in tumor cells enhances tumor growth. Silencing U6 snRNA and TLR3 attenuates tumor growth in YTHDF2 KD settings, suggesting that YTHDF2 may inhibit tumor growth by suppressing the U6-TLR3 signaling axis which otherwise supports tumor growth. Overall, the authors provide compelling evidence for their proposed model of YTHDF2/U6/TLR3-regulation of inflammation, although there may be other non-mutually exclusive mechanisms. We recommend the manuscript to be accepted with moderate revisions to clarify their model (consider alternative models) and also clarify some aspects of their computational analyses.

Response: Thank you for taking the time to review our manuscript and provide us with insightful comments, all of which have been considered carefully in the preparation of the revision (please see our point-by-point responses below). All revisions have been highlighted with red text and submitted as supplementary information. To specifically address all the comments, we have performed a number of new experiments and added in new data (shown in Fig. S6D, S10B, S10C, S11A, S11F, S11G, 4H, 4I, 4J, S12K, S12L, S12M, 5C, 5N, 5O, 5R, 5S, S14F, S14G, S14H, S14I, S14O, 6H, 6I, 6J, S15F, S15G, S15M, S15N, S15O, S15P, S16D- immunoblots, and S17C). In addition, to address concerns in the original data analysis, we have revised the figure panels shown in Fig. 1A, 1B, S1A, and 2E (changing to log₁₀ (q)), Fig. 1F and S5 (removed statistical analysis for all RNA-seq analysis), and Fig. 3F (reformatting the panels and simplify comparison group for clarity), as well as Fig. 1C and 5A (changing violin plot to bar plot). Furthermore, for conciseness and the interest of space, we have removed the original Fig. S1B and S1C, as they were redundant with the original Fig. S1D (now Fig. S1B). With these new data added, changes made, and new information and discussion added, we believe our revised manuscript has significantly improved with all the reviewers' suggestions.

Major comments:

Comment #1:

Previously, deposition of m6A on polyadenylated RNAs by METTL3 has been shown to suppress immunostimulatory dsRNA levels (PMID: 32497523, PMID: 37548590). When METTL16 is downregulated, could the decrease in total m6A deposition lead to overall increased dsRNA levels? METTL16 could be controlling inflammation by controlling U6 RNA as the authors propose, but there may be additional mechanisms by which METTL16 controls inflammation. It would be great to perform dsRNA staining in METTL16 knockdown cells with or without U6 snRNA knockdown to see if METTL16 knockdown significantly alters total dsRNA levels and if U6 snRNA comprise a significant fraction of dsRNA levels using

anti-dsRNA antibodies as previously shown.

Response: Thank you for the constructive comment. This is a great suggestion. To determine whether METTL16 inhibition affects the dsRNA levels as reported for inhibition of METTL3 (PMID: 32497523, PMID: 37548590) and whether U6 snRNA comprises a significant fraction of dsRNA levels, we assessed the difference in the dsRNA levels in A431 cells between control, METTL16 knockdown, and/or U6 knockdown by immunofluorescence analysis using the anti-J2 antibody. However, either METTL16 knockdown, U6 knockdown, or the combination had no effect on the dsRNA levels (Fig. S11F), suggesting that METTL16 regulates the inflammatory response via a distinct mechanism from METTL3 and that the effect of METTL16 knockdown or U6 depletion on inflammatory gene expression is not mediated by secondary dsRNA fragment accumulation. We have added the new data (Fig. S11F) and the following statement in our revised manuscript.

“Finally, to determine whether METTL16 inhibition affects dsRNA levels, as previously observed following METTL3 inhibition, (Gao et al., 2020; Guirguis et al., 2023) and to assess whether U6 snRNA contributes significantly to the dsRNA pool, we compared dsRNA levels in A431 cells under control, METTL16 knockdown, and/or U6 knockdown conditions using immunofluorescence analysis with the anti-J2 antibody. However, METTL16 knockdown, U6 knockdown, or the combined knockdown had no effect on dsRNA levels in A431 cells (Fig. S11F-S11G), suggesting that METTL16 regulates inflammatory response via a distinct mechanism from METTL3 and that the effect of U6 depletion on inflammatory gene expression is not mediated by secondary dsRNA fragment accumulation.” (Page 12).

Comment #2:

The authors propose that, when YTHDF2 is deficient, over-abundant U6 snRNA activates TLR3. However, is it possible that disruption of the splicing machinery (either by YTHDF2 or U6 snRNA depletion) leads to abnormal accumulation of various RNAs that activate TLR3 (or other TLRs) instead of just U6 snRNA directly activating TLR3? Could a splicing-inhibiting drug help distinguish between these two possibilities?

Response: Thank you for the constructive comment. This is a great suggestion. U6 snRNA is best known for its function in splicing (PMID: 29367453). To determine whether splicing plays a role in inflammation regulated by YTHDF2, which controls U6 snRNA stability and abundance, we have performed new experiments to determine whether inhibiting splicing affects the proinflammatory response augmented by YTHDF2 inhibition, due to U6 upregulation. Our new data showed that the splicing inhibitor Pladienolide B (PlaB) decreased the expression of the proinflammatory cytokines in both control and YTHDF2-knockdown cells (Fig. S12K). In parallel, PlaB also decreased the U6 snRNA level, which correlates with the decrease in cytokine expression caused by PlaB (Fig. S12K). It is possible that splicing inhibition by PlaB reduces global transcription thus leading to decreased expression of inflammatory genes and U6 snRNA.

To further determine whether the effect of YTHDF2 inhibition on proinflammatory response is mediated by U6/TLR3, we assessed the effect of U6 knockdown, TLR3 knockdown, and the combination. Our new data showed that either U6 knockdown or TLR3 knockdown decreased IL6 expression and that the combination of double knockdown of TLR3 and U6 has similar effect to knockdown of either U6 or TLR3 singularly (Fig. 4H-4J), suggesting that the effect of U6 is mediated by the TLR3 pathway.

Furthermore, to determine whether METTL16 inhibition affects the dsRNA levels as reported for inhibition of METTL3 (PMID: 32497523, PMID: 37548590) and whether U6 snRNA comprises a significant fraction of dsRNA levels, we assessed the difference in the dsRNA levels in A431 cells between control, METTL16 knockdown, and/or U6 knockdown by immunofluorescence analysis using the anti-J2 antibody. However, either METTL16 knockdown, U6 knockdown, or the combination, had no effect on the dsRNA levels (Fig. S11F), suggesting that METTL16 regulates inflammatory response via a distinct mechanism from METTL3 and that the effect of U6 depletion on inflammatory gene expression is not mediated by secondary dsRNA fragment accumulation.

We have added the new data (Fig. S12K, 4H-4J, and S11F) and the following statements in our revised manuscript.

“To further determine whether the effect of YTHDF2 inhibition on the inflammatory response is mediated through U6/TLR3, we assessed the effect of U6 knockdown, TLR3 knockdown, and their combined depletion. We found that either U6 knockdown or TLR3 knockdown decreased IL6 expression, and that combined depletion of TLR3 and U6 had a similar effect to single knockdown of either U6 or TLR3 in A431 cells (Fig. 4H-4J), suggesting that the effect of U6 on inflammation is mediated through TLR3 signaling. In addition, U6 knockdown had no effect on the dsRNA level in A431 cells (Fig. S11F), suggesting that U6 depletion does not affect secondary dsRNA fragment accumulation in regulating inflammatory response.” (Page 13).

“Given the well-established role of U6 snRNA in regulation of splicing (Didychuk et al., 2018), we aimed to assess whether splicing inhibition affects the inflammatory response driven by YTHDF2 loss through U6 upregulation. We found that the splicing inhibitor Pladienolide B (PlaB) decreased the expression of the proinflammatory cytokines in both control and YTHDF2-knockdown cells (Supplementary Fig. S12K), mimicking the effect of U6 knockdown (Fig. 3). In parallel, PlaB also decreased the U6 snRNA level, which correlates with the decrease in cytokine expression caused by PlaB (Supplementary Fig. S12K). These findings raise the possibility that splicing inhibition by PlaB reduces global transcription or alters RNA metabolism, leading to decreased expression of inflammatory genes and U6 snRNA.” (Page 13).

“Finally, to determine whether METTL16 inhibition affects dsRNA levels, as previously observed following METTL3 inhibition, (Gao et al., 2020; Guirguis et al., 2023) and to assess whether U6 snRNA contributes significantly to the dsRNA pool, we compared dsRNA levels in A431 cells under control, METTL16 knockdown, and/or U6 knockdown conditions using immunofluorescence analysis with the anti-J2 antibody. However, METTL16 knockdown, U6 knockdown, or the combined knockdown had no effect on dsRNA levels in A431 cells (Fig. S11F-S11G), suggesting that METTL16 regulates inflammatory response via a distinct mechanism from METTL3 and that the effect of U6 depletion on inflammatory gene expression is not mediated by secondary dsRNA fragment accumulation.” (Page 12).

Comment #3:

Can SIDT2 transport the U6-YTHDF2 complex altogether, or does U6 and YTHDF2 enter endosomes via independent mechanisms? Is there a way to experimentally delineate this? If not, the author should at least discuss the relative possibilities between these two mechanisms.

Response: Thank you for the constructive comment. This is a great question. Our original data demonstrate that U6 knockdown decreased YTHDF2 levels in endosomes (Fig. 5M, S14M; original Fig. 5I, S14H), indicating that U6 is required for YTHDF2 entry into endosomes. To further determine whether YTHDF2 or U6 m⁶A methylation is important for U6 entry into endosomes, we assessed the effect of YTHDF2 knockdown or METTL16 knockdown. Our new data showed that either YTHDF2 knockdown or METTL16 knockdown increased both endosomal U6 level and total U6 level (Fig. 5N and 5O), indicating that either YTHDF2 or m⁶A methylation is not required for U6 entry into endosomes. Together our data demonstrate that SIDT2 transport either the U6-YTHDF2 complex altogether or U6 alone into endosomes, while YTHDF2 entry into endosomes requires U6 snRNA.

“To examine whether YTHDF2 or U6 enters endosomes as a complex or individually, we assessed the effect of knockdown of U6, YTHDF2, or METTL16 on endosomal localization. Knockdown of U6 reduced the YTHDF2 levels in endosomes in A431 cells (Fig. 5M and supplementary Fig. S14M), indicating that U6 is required for YTHDF2 localization to endosomes. We found that knockdown of either YTHDF2 or

METTL16 increased both endosomal U6 levels and total U6 levels in HeLa cells (Fig. 5N and 5O), indicating that neither YTHDF2 nor m⁶A methylation is required for U6 entry into endosomes.” (Page 17).

“For the first time, we show that both U6 snRNA and YTHDF2 are localized to endosomes to control TLR3 activation.

Specifically, our findings demonstrate that both U6 snRNA and YTHDF2 enter endosomes through SIDT2- and dynamin-dependent intracellular trafficking pathway, while YTHDF2 is transported into endosome by the RNA transporter SIDT2 through binding to m⁶A-methylated U6 snRNA. This model is supported by the following key findings (Fig. 5, supplementary Fig. S14): (1) SIDT2 knockdown or dynamin inhibition reduces entry of both U6 snRNA and YTHDF2 to endosomes, (2) to our best knowledge, YTHDF2 does not have an N-terminal peptide signal sequence that would enable YTHDF2 access to the endosomal lumen, (3) U6 knockdown reduces YTHDF2 levels in endosomes, (4) YTHDF2-N, a mutant lacking the m⁶A-RNA binding domain, showed reduced endosomal level, (5) knockdown of YTHDF2 or METTL16 increased U6 snRNA levels in endosomes, indicating that neither YTHDF2 nor m⁶A methylation of U6 snRNA is required for U6 snRNA entry into endosomes, and (6) extracellular uptake for either U6 snRNA or YTHDF2 was not observed for endosome transportation. Taken together, these findings support a model that U6 snRNA is transported into endosomes at least in part by SIDT2 in a dynamin-dependent, m⁶A methylation-independent, and YTHDF2-independent manner, while YTHDF2 entry into endosomes is dependent on U6 snRNA.” (Page 26-27).

Comment #4:

Since UVB can also induce DNA damage, could a DNA sensing mechanism be involved in upregulating inflammation following UVB treatment? For example, could activation of TLR9 or cGAS-STING also contribute to the observed upregulation in inflammation?

Response: Thank you for the constructive comment. This is a great question. To determine whether UVB-induced DNA damage also contributes to the induction of inflammation, we have performed new experiments to assess the effect of TLR9 knockdown or cGAS inhibition. Our new data showed that either TLR9 knockdown or the cGAS inhibitor G140 has no effect on UVB-induced expression of inflammatory genes (Fig. S12L-S12M). We have added the new data and the following statements in the Results section.

“To determine whether UVB-induced DNA damage also contributes to the induction of inflammation, we assessed the effect of TLR9 knockdown or cGAS inhibition. We found that either TLR9 knockdown or inhibition of cGAS had no effect on UVB-induced expression of inflammatory genes (Fig. S12L-S12M). These data suggest that UVB-induced inflammation is not mediated by UVB-induced DNA damage.” (Page 14).

Comment #5:

The role of MYPT1 is confusing. I think the model is that UVB-irradiated YTHDF2 interacts with MYPT1, which dephosphorylates YTHDF2 to prevent it from binding CNOT1. To prove MYPT1 plays a role in regulating U6 snRNA stability, the authors need to downregulate or overexpress MYPT1 and evaluating its effect on U6 snRNA levels and stability. Otherwise, the interpretation of MYPT1's role should be toned down.

Response: Thank you for the constructive comment. To determine whether MYPT1 regulates U6 snRNA stability, we assessed the effect of MYPT1 knockdown. Our new data showed that MYPT1 knockdown increased U6 snRNA levels and stability (Fig. S15F-S15G), supporting a role of MYPT1 in regulating U6 snRNA decay.

“To determine whether MYPT1 regulates U6 snRNA stability, we assessed the effect of MYPT1 knockdown. We found that MYPT1 knockdown increased U6 snRNA levels and stability in HeLa cells (Fig. S15F-S15G), supporting a role for MYPT1 in regulating U6 snRNA decay.” (Page 18).

Comment #6:

In Figure 6C-D, it is clear that phosphomimetic YTHDF2 binds MYPT1 and CNOT1, and Supplementary Figure S15E shows that UVB radiation promotes binding of YTHDF2 to MYPT1. However, it is unclear how YTHDF2 preferentially interacts with MYPT1 upon UVB treatment. Supplementary Figure S15E suggests that MYPT1 expression levels do not change with UVB treatment, but what about CNOT1 expression levels? If there is no difference in MYPT1 and CNOT1 expression levels between sham and UVB conditions, the authors should at least discuss potential mechanisms through which cells respond to UVB radiation to promote MYPT1-YTHDF2 interaction over CNOT1-YTHDF2 interaction.

Response: Thank you for the constructive comment. To determine whether UVB irradiation affects CNOT1 expression, we assessed that the effect of UVB irradiation. Our new data showed that UVB irradiation has no effect on CNOT1 protein levels (Fig. S15P). We have added the new data (Fig. S15P) and the following statements in the Results section and the Discussion section.

“To determine whether UVB irradiation affects CNOT1 expression, we assessed the effect of UVB irradiation on CNOT1 protein levels. We found that UVB irradiation had no effect on CNOT1 protein expression in either HeLa or HaCaT cells (Supplementary Fig. S15P). These results demonstrate that CNOT1 binds to U6 snRNA and mediates U6 decay through interacting with YTHDF2 in an m⁶A-methylation dependent manner.” (Page 19).

“Previous studies have shown that Chk1, a kinase activated by DNA damage and UV radiation (Liu et al., 2000), binds and phosphorylates MYPT1, leading to the recruitment of protein phosphatase 1 β (PP1c β) to dephosphorylate its substrate (Hu et al., 2018). It is possible that UV radiation activates Chk1 and thus MYPT1 phosphorylation, leading to MYPT1 binding to YTHDF2.” (Page 28).

Comment #7:

In Figure 5, the authors show that YTHDF2 and U6 snRNA enter endosomes. However, the functional significance of this as it pertains to their proposed model is unclear (especially with regards to YTHDF2). The authors should confirm that U6 snRNA entry into the endosome is required for upregulation of inflammatory genes and whether or not YTHDF2 entry into the endosome affects inflammatory gene expression.

Response: Thank you for the constructive comment. This is a great question and suggestion. To determine the importance of U6 snRNA entry into the endosome in the increased expression of inflammatory genes, we assessed the effect of Dynasore, which inhibits U6 entry into endosomes (Fig. 5G), and the role of the interaction U6 with TLR3, an endosomal RNA sensor. Our new data showed that Dynasore reduced expression of inflammatory genes induced by U6 or YTHDF2 knockdown (Fig. 5R and 5S). To further determine whether the effect of YTHDF2 inhibition on proinflammatory response is mediated by U6/TLR3, we assessed the effect of U6 knockdown, TLR3 knockdown, and the combination. Our new data showed that either U6 knockdown or TLR3 knockdown decreased IL6 expression and that the combination of double knockdown of TLR3 and U6 has similar effect to knockdown of either U6 or TLR3 singularly (Fig. 4H-4J), suggesting that the effect of U6 is mediated by the TLR3 pathway. Together with our findings that YTHDF2 endosomal entry is dependent on U6, while U6 endosomal entry is independent of YTHDF2 (Fig. 5M-O), our data demonstrate the U6 entry into endosomes is critical for upregulation of inflammatory genes.

“Lastly, to determine whether U6 snRNA entry into endosomes is required for the increased expression of inflammatory genes, we assessed the effect of Dynasore, an inhibitor of U6 endosomal entry (Fig. 5G), on cytokine expression. We found that Dynasore reduced expression of inflammatory genes induced by U6 in HeLa cells or YTHDF2 knockdown in A431 cells (Fig. 5R and 5S).” (Page 17).

“To further determine whether the effect of YTHDF2 inhibition on the inflammatory response is mediated through U6/TLR3, we assessed the effect of U6 knockdown, TLR3 knockdown, and their combined depletion. We found that either U6 knockdown or TLR3 knockdown decreased IL6 expression, and that combined depletion of TLR3 and U6 had a similar effect to single knockdown of either U6 or TLR3 in A431 cells (Fig. 4H-4J), suggesting that the effect of U6 on inflammation is mediated through TLR3 signaling. In addition, U6 knockdown had no effect on the dsRNA level in A431 cells (Fig. S11F), suggesting that U6 depletion does not affect secondary dsRNA fragment accumulation in regulating inflammatory response.” (Page 13).

Comments on computational analysis: #8 ~ #11

Comment #8:

The details behind the bulk RNA-seq analyses (Figures 1A-B, 2E, S1A-B, S1D, and S7A-C) need to be clarified in the Methods (there is currently no mention of how expression differences and their statistical significances were obtained per gene). For example, which tools were used to calculate differential expression? How were lowly expressed genes filtered out before testing for differential expression? How were thresholds for differential expression determined? In Figures 1A-B, 2E, S1A-B, there also appears to be filtering thresholds for how genes were selected for pathway analyses, but this is not clearly indicated in the figures. If there are indeed filtering thresholds, they should be clearly indicated as such in the figure legends or the Methods. Additionally, how the top genes were selected in Supplementary Figures S7B-C should be clearly indicated. For example, were the genes sorted by expression or by significance?

Response: Thank you for the constructive comment. For RNA-seq analysis, we used HISAT2 for reads alignment, and DESeq2 for differential gene expression analysis. Lowly expressed genes were pre-filtered out using reads count threshold of 10. We used an adjusted p value (DESeq2 default) threshold of 0.05 for all differential expression analysis. For genes in Supplementary Fig. S7B-C, we selected following 3 criteria as stated in our manuscript: (1) increased RNA level, (2) increased m⁶A enrichment, (3) related to inflammation or tumorigenesis. The selected genes were then sorted using adjusted p values to identify top 30 candidates. We have added these details to the Methods section.

“Reads were aligned to the reference genome (hg38) using HISAT2. DESeq2 was used for differential gene expression analysis. Lowly expressed genes were pre-filtered out using reads count threshold of 10. An adjusted p value (DESeq2 default) threshold of 0.05 was used for all differential expression analysis.” (Page 36).

Comment #9:

For all bulk RNA-seq analyses and pathway analyses (Figures 1A-B, 2E, S1A-B, and S7A-C), it appears that the authors did not calculate adjusted p-values for multiple hypothesis testing as all significant results are reported as $p < 0.05$, not FDR, q , or adjusted $p < 0.05$. If the authors did perform multiple hypothesis testing, the adjustment method should be reported in figure legends and the Methods, and measures of statistical significance should be appropriately named in figure axes. If the authors did not perform multiple hypothesis testing, the analyses should be redone taking adjusted p-values into account when selecting for genes as input into Metascape analysis as well as reporting significantly enriched pathways from

Metascope. It is well-known that multiple hypothesis testing is critical for reporting accurate results from expression and pathway analyses derived from RNA-seq data as it greatly reduces the frequency of false positives generated by these analyses.

Response: Thank you for the constructive comment. We used DESeq2 with default settings to calculate differentially expressed genes, so all functional analyses were based on genes with adjusted p value < 0.05. We have added the method details in the Method section. In addition, we have prepared the pathway analysis figure panels using FDR, q or adjusted p < 0.05. We have removed the original figure panels (Fig. 1A, 1B, S1A-C, 2E) and added in the revised figure panels (Fig. 1A, 1B, S1A, 2E). We have removed the original Fig. S1B-S1C, as we realized that Fig. S1B and S1C are a bit redundant with the original Fig. S1D. We have added the details in the figure legends.

“Reads were aligned to the reference genome (hg38) using HISAT2. DESeq2 was used for differential gene expression analysis. Lowly expressed genes were pre-filtered out using reads count threshold of 10. An adjusted p value (DESeq2 default) threshold of 0.05 was used for all differential expression analysis.” (Page 36).

“Genes with an adjusted p value less than 0.05 by DESeq2 were used for pathway analysis and Venn diagram analysis.” (Page 41, the Methods section and legends for Fig. 1A, 1B, S1A, and 2E).

Comment #10:

The analyses in Figure 1F and Supplementary Figure S5: Firstly, there is no mention of any normalization methods used in the Methods or the figure legends, and the y-axis of these box plots read “YTHDF2 level”, which is not a specific quantification of RNA-seq data. Secondly, t-tests were used to determine statistical significances of expression differences. It is well-known that RNA-seq data follows a negative binomial distribution, not a normal distribution (which is assumed by t-tests). Thirdly, the p-values are not adjusted for multiple hypothesis testing, which is critical for accurately reporting statistical significance from RNA-seq data. I recommend the authors perform differential expression analysis on each dataset and report the results for YTHDF2 obtained from those analyses. If the appropriate analytical pipelines differ between datasets, the authors should also clearly report these different pipelines in the Methods. For example, GSE45291 was done on a microarray, whereas GSE51092 was done via RNA-seq. Additionally, the text currently states that all public datasets examined from ADEx are from RNA-seq datasets. Since the pool of datasets represents both RNA-seq and microarray datasets, it should be specified which dataset comes from which type of analysis as microarray data has been shown to be less accurate than RNA-seq data.

Response: Thank you for the constructive comment. We apologize for the errors. As the reviewer pointed out, not all datasets were from RNA-seq analysis. In the revised manuscript, we have indicated RNA-seq or Microarray for each dataset in the figure panels. After discussing with Dr. Xiaolong Cui, an experienced bioinformatician and our collaborator, for RNA-seq data analysis, we have removed statistical analysis and only showed the trend. For microarray data, we used Student’s t-test. Although microarray data has been shown to be less accurate than RNA-seq data, the trend shows that YTHDF2 is lower in diseased groups than health control, which is intriguing and consistent with the inhibitory role of YTHDF2 in inflammatory gene expression. We have revised the statistical analysis used for these figures in the figure legends (Fig. 1F and S5) and added “Microarray” or “RNA-seq” in each figure panel and the list in the Methods section.

“Datasets analyzed were GSE110169_SLE (microarray), GSE50772 (microarray), GSE72509 (RNA-Seq), GSE11907_T1D (microarray), GSE110169 RA (microarray), GSE56649 (microarray), GSE45291 (microarray), GSE89408 (RNA-seq), GSE104174 (RNA-Seq), GSE124073 (RNA-Seq), and GSE51092 (microarray).” (Page 42).

Comment #11:

For all pathway analyses (Fig 1A-B, 2E, S1A-B), the methods need to be clarified on how pathway analysis was conducted in Metascape. For example, what type of defaults (background gene lists) were used? This would help readers interpret the results of the pathway enrichment analysis as the background gene list has been shown to heavily influence the statistical significance of results by inflating false positives if the background list includes far more genes than is actually sequenced (10.1038/s41596-018-0103-9).

Response: Thank you for the constructive comment. We used the default background gene list (all genes) provided by Metascape. We agree with the reviewer that the background gene list has been shown to heavily influence the statistical significance of results by inflating false positives if the background list contains far more genes than is actually sequenced (10.1038/s41596-018-0103-9.). As an initial screening, we reckon using the default background gene list provided by Metascape is appropriate. We follow up on the top pathway and validated the importance of the inflammatory pathway in the role of YTHDF2. We have added the following information in the Methods section.

“...using the default background gene list (all genes) provided by Metascape for initial pathway screening” (Page 41).

Minor comments:

Comment #1:

For all qPCR analyses, the housekeeping gene used as controls are not listed in the legends. The Methods specify that either GAPDH, ACTB, or HPRT1 were used throughout the paper, but the figure legends should specifically specify which gene was used for which figure panel.

Response: Thank you for the constructive comment. We have added the specific housekeeping gene used for each figure panel in figure legends.

Comment #2:

The authors should elaborate on why inflammation decreased in YTHDF2-depleted cells following UVB treatment for 24 hours compared to 6 hours.

Response: Thank you for the constructive comment. UVB irradiation induces an acute inflammatory response as observed by increased expression of inflammatory genes. To determine whether the inflammatory response correlates with YTHDF2 protein abundance, we performed a new experiment to assess YTHDF2 protein levels at different time points. Indeed, as compared with sham-irradiation, YTHDF2 protein level is decreased at 6 h post-UVB irradiation, while it was recovered at 24 h post-UVB irradiation (Fig. S16D), likely due to increased YTHDF2 mRNA levels (Fig. S16D). Such temporal YTHDF2 protein level is correlated with the increases in inflammation at 6 h followed by a decrease at 24 h post-UVB irradiation. We have added the following statements in the Results section.

“UVB irradiation induces an acute inflammatory response as observed by increased expression of inflammatory genes (Fig. 1). To determine whether the inflammatory response correlates with YTHDF2 protein abundance, we assessed YTHDF2 protein levels at different time points. Indeed, as compared with sham-irradiation, YTHDF2 protein level is decreased at 6 h post-UVB irradiation, while it was recovered at 24 h post-UVB irradiation in HaCaT cells (Supplementary Fig. S16D), likely due to increased YTHDF2 mRNA levels (Supplementary Fig. S16D). The temporal regulation of YTHDF2 protein expression by UVB irradiation negatively correlated with the increases in U6 snRNA levels (Fig. 2I) as well as inflammatory

gene expression (Qiang et al., 2017) at 6 h followed by a decrease at 24 h post-UVB irradiation, further supporting an inhibitory role of YTHDF2 in UVB-induced inflammation.” (Page 20).

Comment #3:

In Figure S1A, the title states that these pathways are “Among the top 30 enriched pathways”, but 4 out of 6 pathways indicated in Figure 1A, (hsa05200_Pathways in cancer, R-HAS-449147_Signaling by Interleukins, hsa04657_IL-17 signaling pathway, and R-HAS-168164_Toll Like Receptor 3 (TLR3) Cascade) are not found in Supplementary Figure S1A, which represents the full list of top 30 enriched pathways. Can the authors explain this discrepancy?

Response: Thank you for the constructive comment. We apologize for the error. We went back to check the cause of the issue and found that the data in S1A was generated by using Metascape, while Fig. 1A was created using a Top 30 graph from a raw Excel file (GO All) derived from the Metascape ZIP files in 2023 when we prepared the figure panels, and we failed to identify the error. We have now updated Fig. 1A, S1A, 1B, and 2E, to correct this error, and to use q (or FDR) instead of p values.

Comment #4:

Figure 3F is too confusing to read. The authors should highlight only the relevant comparisons made to back-up the claims stated in the main text on page 10.

Response: Thank you for the constructive comment. We have removed several unimportant comparisons to simplify the figure panel and emphasize the most important comparisons in Fig. 3F.

Comment #5:

There are a few typos in the figures and figure legends:

- *Figure 4J: the figure legend reads “YTHDF2-Flag overexpression.” Shouldn’t this read “TLR3-Flag overexpression”?*
- *Figure 6F: the x-axis reads “IP:CONT1” when it should read “IP: CNOT1”.*
- *Supplementary Figure S1A: there is no label for the x-axis.*
- *Supplementary Figure S16G: there is no explanation for the color key in the figure legend.*

Response: Thank you for the constructive comments. We have corrected all the errors in these figure panels and legends (Fig. 4J legend, Fig. 6F, Fig. S1A, and Fig. S16G legend).

Comment #6:

Dynasore inhibits dynamin, which is localized to the plasma membrane to facilitate endocytosis, and the authors show that dynasore may reduce U6 snRNA stability. However, the mechanism behind this is unclear, especially as to how U6 snRNA packaging into endosomes appears to be dynamin-dependent. Although the authors show YTHDF2 is not secreted and taken up by other cells in Figure 5O, is it possible that U6 snRNA is secreted and taken up by other cells?

Response: Thank you for the constructive comment and suggestion. To determine whether U6 snRNA is secreted and taken up by other cells, we have performed a new experiment. When we treated cells with U6 snRNA knockdown with conditioned medium from cells with or without U6 knockdown, we did not observe a significant difference (Fig. S14O), suggesting that the endosomal U6 snRNA is not from U6 secretion followed by endocytosis. These data together with our original data (Fig. 5G-M) suggest that U6 snRNA enters into endosomes through endocytosis pathway dependent on the dynamin pathway and SIDT2. We have added the new data (Fig. S14O) and the following statement in the Results section.

“Next, to characterize whether YTHDF2 or U6 snRNA is trafficked intracellularly or extracellularly to endosomes, we assessed whether cells uptake either YTHDF2 or U6 snRNA secreted by other cells. However, when cultured with conditioned medium from control cells, no YTHDF2 or U6 snRNA was detected in endosomes of HeLa cells with YTHDF2 deletion or A431 cells with U6 snRNA knockdown, respectively (Fig. S14N-S14O), suggesting that intracellular trafficking pathways deliver YTHDF2 and U6 snRNA into endosomes.” (Page 17).

Comment #7:

In Figure 5K, inhibition of SIDT2 blocks YTHDF2 entry into endosomes. How does inhibiting an RNA transporter also prevent YTHDF2 from entering endosome?

Response: Thank you for the constructive comment. This is a great question. Based on our original data and new data shown in Fig. 5 and S14, we propose that YTHDF2 is transported into endosome by the RNA transporter through binding to m⁶A-methylated U6 snRNA. This model is supported by our findings: (1) SIDT2 knockdown or dynamin inhibition reduces both entry of U6 snRNA and YTHDF2 to endosomes. (2) To our best knowledge, YTHDF2 does not have an N-terminal peptide signal sequence that enables YTHDF2 access to the endosomal lumen. (3) U6 knockdown reduces YTHDF2 levels in endosomes. (4) YTHDF2-N, a mutant lacking the m⁶A-RNA binding domain, showed reduced endosomal level. (5) Knockdown of YTHDF2 or METTL16 increased U6 snRNA levels in endosomes, indicating that neither YTHDF2 nor m⁶A methylation of U6 snRNA is required for U6 snRNA entry into endosomes. (6) U6 snRNA and YTHDF2 enter endosomes through intracellular trafficking pathways instead of secretion followed by uptake. Taken together these findings support a model that U6 snRNA is transported into endosomes at least in part by SIDT2 in a dynamin dependent manner, while YTHDF2 entry into endosomes is mediated by its binding to m⁶A-methylated U6 snRNA. We have added the new data (Fig. 5R-5S) and the following statements in the discussion.

“For the first time, we show that both U6 snRNA and YTHDF2 are localized to endosomes to control TLR3 activation.

Specifically, our findings demonstrate that both U6 snRNA and YTHDF2 enter endosomes through SIDT2- and dynamin-dependent intracellular trafficking pathway, while YTHDF2 is transported into endosome by the RNA transporter SIDT2 through binding to m⁶A-methylated U6 snRNA. This model is supported by the following key findings (Fig. 5, supplementary Fig. S14): (1) SIDT2 knockdown or dynamin inhibition reduces entry of both U6 snRNA and YTHDF2 to endosomes, (2) to our best knowledge, YTHDF2 does not have an N-terminal peptide signal sequence that would enable YTHDF2 access to the endosomal lumen, (3) U6 knockdown reduces YTHDF2 levels in endosomes, (4) YTHDF2-N, a mutant lacking the m⁶A-RNA binding domain, showed reduced endosomal level, (5) knockdown of YTHDF2 or METTL16 increased U6 snRNA levels in endosomes, indicating that neither YTHDF2 nor m⁶A methylation of U6 snRNA is required for U6 snRNA entry into endosomes, and (6) extracellular uptake for either U6 snRNA or YTHDF2 was not observed for endosome transportation. Taken together, these findings support a model that U6 snRNA is transported into endosomes at least in part by SIDT2 in a dynamin-dependent, m⁶A methylation-independent, and YTHDF2-independent manner, while YTHDF2 entry into endosomes is dependent on U6 snRNA.” (Page 26-27).

Comment #8:

The authors should elaborate on the identity of the unmarked bands above 25 kDa in Figure 5N. Are these bands a result of nonspecific antibody binding? If so, please clarify in the legend.

Response: Thank you for the constructive comment. We confirm that the unmarked bands above 25 kDa in Figure 5N are results of nonspecific antibody binding. We have added the information in the figure legend for Fig. 5Q (original Fig. 5N).

Comment #9:

The authors should describe in the Methods how they generated their custom antibody for phosphorylated YTHDF2 at S39 (Figure 6B).

Response: Thank you for the constructive comment. We have added the information on how this YTHDF2 S39 antibody was generated in the Methods section.

“One-and-a-half-year-old New Zealand rabbits (2.5 kg), housed under SPF conditions, were subcutaneously injected with 700 µg of a modified antigen peptide (EPYL(S-p)PQAR-C-KLH) emulsified with Complete Freund’s Adjuvant (CFA) for the primary immunization, followed by five booster injections of 350 µg of the same peptide emulsified with Incomplete Freund’s Adjuvant (IFA) at 1-, 2-, and 3-week intervals. Terminal bleeds were collected after the final immunization. Polyclonal antibodies were purified from the terminal bleeds by antigen affinity chromatography using a column conjugated with the modified peptide (EPYL(S-p)PQAR-C), followed by depletion using a column conjugated with the non-modified peptide (EPYLSPQAR-C).” (Page 40-41).

Comment #10:

In Figure 4K: To clarify the molecular binding dynamics between TLR3, YTHDF2, and U6 snRNA, a better method is to perform electrophoretic mobility shift assays to obtain dissociation constants (Kd) for interactions between U6-TLR3 vs. U6-YTHDF2.

Response: Thank you for the constructive comment. We attempted to perform electrophoretic mobility shift assays (EMSA) assay to the interactions between U6-TLR3 vs U6-YTHDF2 using recombinant TLR3 and YTHDF2 and biotin-labeled U6 snRNA, as our lab does not have a radioactive protocol. However, we were unable to detect any signal of binding for either U6-TLR3 or U6-YTHDF2, although we were able to detect the binding using pulldown assays (Fig. 4N, original Fig. 4K). Future investigation using more sensitive methods, such as radioactive assays and/or increased concentrations of the U6 probe and/or recombinant YTHDF2 and TLR3 proteins, is needed to assess dissociation constants (Kd) for the interactions using the EMSA assay.

Method used is as follows:

“The ability of biotinylated m⁶A U6 snRNA to bind to YTHDF2 or TLR3 in vitro was tested using the LightShift™ Chemiluminescent RNA EMSA Kit (Thermo, 20158) following the manufacturer’s protocol. Briefly, Biotin-labeled m⁶A U6 snRNA (2 nM) was mixed with or without recombinant proteins such as 3 µl human YTHDF2 (770 nM), 4 µl human TLR3 (480 nM), or 3 µl YTHDF2 plus 4 µl TLR3 in binding buffer (10 × Binding buffer 2 µl, DTT 1 mM, glycerol 5%) and incubated at room temperature for 30 min. 5 × loading buffer was then added, and the complex was separated on a 6% Novex™ TBE gel (Thermo, EC6265BOX) at 100 voltage for 40 min. The binding reactions were transferred onto a nylon membrane or incubated the reaction gel with the diluted streptavidin-HRP (1: 10000) probe for 1 h, add ECL exposure solution after washing, and image.”

Reviewer #3 (Remarks to the Author):

I co-reviewed this manuscript with one of the reviewers who provided the listed reports. This is part of the

Response: Thank you for taking the time to review our manuscript and provide us with insightful comments, all of which have been considered carefully in the preparation of the revision (please see our point-by-point responses below). All revisions have been highlighted with red text and submitted as supplementary information. To specifically address all the comments, we have performed a number of new experiments and added in new data (shown in Fig. S6D, S10B, S10C, S11A, S11F, S11G, 4H, 4I, 4J, S12K, S12L, S12M, 5C, 5N, 5O, 5R, 5S, S14F, S14G, S14H, S14I, S14O, 6H, 6I, 6J, S15F, S15G, S15M, S15N, S15O, S15P, S16D- immunoblots, and S17C). In addition, to address concerns in the original data analysis, we have revised the figure panels shown in Fig. 1A, 1B, S1A, and 2E (changing to log₁₀ (q)), Fig. 1F and S5 (removed statistical analysis for all RNA-seq analysis), and Fig. 3F (reformatting the panels and simplify comparison group for clarity), as well as Fig. 1C and 5A (changing violin plot to bar plot). Furthermore, for conciseness and the interest of space, we have removed the original Fig. S1B and S1C, as they were redundant with the original Fig. S1D (now Fig. S1B). With these new data added, changes made, and new information and discussion added, we believe our revised manuscript has significantly improved with all the reviewers' suggestions.

Reviewer #4 (Remarks to the Author):

Yang et al NCOMMS-24-83544

The authors describe truly novel findings. The RNA modification m6A is shown to be present on a variety of RNA molecules. The m6A reader protein YTHDF2 is shown to bind modified mRNAs and mediate their decay via recruitment of the deadenylase CCR4-NOT1 complex. In this study, the authors demonstrate that YTHDF2 binds the spliceosomal U6 snRNA. This is new, as such an interaction was never demonstrated before. They also show that this interaction regulates RNA levels of U6 snRNAm where YTHDF2 interacts with deadenylase component CNOT1, which may mediate decay of m6A-modified U6 snRNA. The authors describe the role of m6A reader YTHDF2 in the inflammatory response through its regulation of the U6 snRNA. They provide data from several different cell lines and a skin-specific conditional deletion mouse model. The study provides evidence that human YTHDF2 binds to and regulates the stability of U6, modulating the expression of inflammatory regulators via TLR3. Finally, they show that UVB irradiation leads to autophagy-mediated downregulation of YTHDF2, and that deletion of YTHDF2 in mouse skin cells increases sensitivity to UVB radiation. Overall, the findings are novel, unexpected and relevant for the RNA modification field.

Response: Thank you for taking the time to review our manuscript and provide us with insightful comments, all of which have been considered carefully in the preparation of the revision (please see our point-by-point responses below). All revisions have been highlighted with red text and submitted as supplementary information. To specifically address all the comments, we have performed a number of new experiments and added in new data (shown in Fig. S6D, S10B, S10C, S11A, S11F, S11G, 4H, 4I, 4J, S12K, S12L, S12M, 5C, 5N, 5O, 5R, 5S, S14F, S14G, S14H, S14I, S14O, 6H, 6I, 6J, S15F, S15G, S15M, S15N, S15O, S15P, S16D- immunoblots, and S17C). In addition, to correct the issues in the original data analysis, we have revised the figure panels shown in Fig. 1A, 1B, S1A, and 2E (changing to log₁₀ (q)), Fig. 1F and S5 (removed statistical analysis for all RNA-seq analysis), and Fig. 3F (reformatting the panels and simplify comparison group for clarity), as well as Fig. 1C and 5A (changing violin plot to bar plot). Furthermore, for conciseness and the interest of space, we have removed the original Fig. S1B and S1C, as they were redundant with the original Fig. S1D (now Fig. S1B). With these new data added, changes made, and new

information and discussion added, we believe our revised manuscript has significantly improved with all the reviewers' suggestions.

General comments

1. Analysis of gene expression changes and changes in m⁶A (specifically changes in inflammatory genes, changes in U6 levels, U6 m⁶A status etc.) from skin of cKO YTHDF2 mice would have added a lot to the hypothesis that YTHDF2 regulation of U6 is responsible for histological observations

Response: Thank you for the constructive comment. This is a great suggestion. To determine the gene expression changes and changes in m⁶A, we have attempted to isolate RNA from mouse skin for several times. However, we were unable to obtain enough RNA from frozen skin samples to perform these analyses. Alternatively, we have performed immunofluorescence and immunohistochemistry staining of COX-2 protein levels in mouse skin. Unfortunately, the anti-COX-2 antibody was not specific for these analyses as non-specific staining was also observed. To address this question indirectly, as a complimentary approach, we were able to isolate primary mouse epidermal keratinocytes from mice with wild-type or skin-specific deletion of YTHDF2. Unfortunately, we could not obtain sufficient cKO mice within the time allowed for the revision (3 months), as we lost the cKO colony during transition when the first author, Dr. Yang, left the lab in 2023. Therefore, we have focused on assessing the effect of heterozygous YTHDF2 deletion in the mouse skin (DF2 cHet) on UVB-induced inflammatory gene expression. Indeed, primary keratinocytes with heterozygous YTHDF2 deletion showed increased expression of the inflammatory genes and m⁶A level in U6 snRNA as compared with their WT counterparts (Fig. S6B and S10C), consistent our observation in cultured human cells (**Fig. 1H-1K, 1M-1Q, 2K, 2Q**). Together with our original data, our new data supporting a critical role of YTHDF2 in suppressing UVB-induced inflammation. We have added these new data (Fig. S6B and S10C) and the following statements in our revised manuscript.

“Furthermore, primary mouse keratinocytes isolated from mice with skin-specific heterozygous YTHDF2 deletion (DF2 cHet) showed increased expression of TNF α , IL-1 β , and COX-2 as compared with their WT counterparts under baseline condition or UVB stress, while IL-6 expression was not affected by either YTHDF2 inhibition or UVB stress (Supplementary Fig. S6D), likely due to the difference in genetic or molecular context between mouse primary keratinocytes and human keratinocyte cell line or NHEK cells.” (Page 7).

“...and primary mouse keratinocytes (Fig. 2K, Supplementary Fig. S10C).” (Page 9).

2. In the text, it is often not stated which cell line, or even system, is being referred to or studied

Response: Thank you for the constructive comment. We have added the specific cell line or system in the text, the figure panels, and figure legends.

a. E.g. in the abstract the authors do not state that they studied YTHDF2 in human cells, and “skin-specific deletion” does not mention the mouse model

Response: Thank you for the constructive comment. We have added the information for human cells and the mouse model in the Abstract.

“...responses in human and mouse cells and mouse models.” (Page 3).

“... in human keratinocytes and mouse skin. Skin-specific deletion of YTHDF2 in mice...” (Page 3).

b. Several different cell lines (HaCaT, HeLa, A431, 293T, NHEK, CHL-1) are used in the study, but the text rarely states which result was obtained from which cell line, or the rationale for doing particular experiments in a particular cell line

Response: Thank you for the constructive comment. In this study, we elected to utilize different cell lines, in order to (1) ensure the robustness of the observed role of the YTHDF2/U6 axis in modulating inflammatory response and (2) achieve technical feasibility as some cell lines such as HaCaT and A431 cells are more difficult to transfect and yield lower amounts of cell lysates compared with model cell lines, such as HeLa or 293T cells. The details on each cell line are provided below.

- NHEK cells are primary human keratinocytes which exhibit UVB-induced inflammatory responses.
- HaCaT cells are an immortalized keratinocyte cell line and a model for non-tumorigenic keratinocytes, which respond to UVB damage by inducing an inflammatory response that is augmented by YTHDF2 inhibition.
- A431 cells are a human epidermoid cell carcinoma cell line, in which YTHDF2 knockdown augments baseline inflammatory response, possibly due to the increased sensitivity to YTHDF2 inhibition in these tumorigenic cells.
- HeLa cells are a human cervical cancer cell line, a model cell line instrumental for biomedical research and widely used due to its efficient transfection efficiency and fast growth. We choose to use HeLa cells due to the follow reasons: (1) HeLa cells are epithelial cancer cells, (2) HeLa cells grow faster than many other cell lines and can generate larger amount of proteins and RNAs for less sensitive assays, such as endosome fractionation, (3) HeLa cells have much higher transfection efficiency, making them more feasible to study the role of gene overexpression, knockdown, or knockout, and (5) HeLa cells exhibit UVB-induced inflammatory responses.
- HEK 293T (293T) cells are an epithelial-like immortalized cell line expressing the SV40 large T antigen. Similar to HeLa cells, 293T cells exhibit higher transfection efficiency, faster growth, and enhanced protein production, making them suitable for less sensitive assays such as endosome fractionation.
- Mouse primary keratinocytes were isolated from mice with or without skin-specific heterozygous YTHDF2 knockout during the revision to validate our findings observed in cultured human cells. We have added these new data (Fig. S6D and S10B). Our new data in these cells are consistent with data from the human cell lines, validating the critical role of YTHDF2 in controlling the inflammatory response.
- CHL-1 cells are a human melanoma cell line and were used to determine whether the effect of UVB irradiation is specific to keratinocytes or applies to other cell types such as melanoma cells.

We have added the justifications for using different cell lines in the Results section.

“..., non-tumorigenic human keratinocytes...” (Page 5).

“...in HeLa cells, a cell line with high transfection efficiency, ...” (Page 11).

“To explore the role of m⁶A methylation of U6 in YTHDF2’s function in skin cancer and confirm our findings in HaCaT cells, we next assessed the consequence of METTL16 inhibition on the expression of cytokines in the human A431 skin carcinoma cells” (Page 11).

“in 293T cells in order to achieve high plasmid transfection and protein expression” (Page 16).

“...in multiple cells and primary keratinocytes including HaCaT, CHL-1 (melanoma cells), HeLa, and NHEK cells ...” (Page 19).

b.i. Text should refer to cell lines, e.g. “Our RNA-seq analysis showed that YTHDF2 knockdown...” should add “in HaCaT cells”

Response: Thank you for the constructive comment. We have added “... in HaCaT cells...” in the text (Page 21).

b.ii. Figure panels should also show clearly which cell line the result comes from. This is done for some panels (Fig 5C, 5G etc.), but most are unlabelled

Response: Thank you for the constructive comment. In our revised manuscript, we have added specific cell line or cells used for each figure in all the figure panels.

b.iii. Some figure legends do not mention cell line used

Response: Thank you for the constructive comment. In our revised manuscript, we have added specific cell line or cells used for each figure in all the figure panels.

b.iii.1. E.g. Fig. 5D. Panel includes label 293T, but not stated in figure legend – “D. Endo-IP analysis of YTHDF2 protein levels in endosomes.”

Response: Thank you for the constructive comment. We have added “in 293T cells” in the figure legend for Fig. 5E (Original Fig. 5D).

b.iii.2. Fig. 5K. No label in panel, and no mention of cell line in figure legend. Also error in figure legend “K. Immunoblot analysis of YTHDF2 in whole cell lysates (WCL) and endosomes in cells as in K.”

Response: Thank you for the constructive comment. We have added “in A431 cells” in the figure panel and legend (Fig. 5K). We have corrected the error in Figure legend (Page 59).

b.iii.3. Fig. 4M. No label in panel, no cell line mentioned in legend.

Response: Thank you for the constructive comment. We have added “in DF2 KO HeLa cells” in the figure panel and legend (Fig. 4P; original Fig. 4M).

3. Most figures contain many panels showing qPCR results for different mRNAs in different conditions. It is sometimes unclear why certain mRNAs were or were not tested under specific conditions, and in the different cell lines used. Clearer labelling in figures and some explanation of the rationale in the text would be useful.

Response: Thank you for the constructive comment. We have added specific labeling of cell lines in all figure panels. For qPCR results for different mRNAs in different conditions, we elected to focus on these mRNAs based on our findings on the UVB effect and/or YTHDF2 knockdown in HaCaT cells. Although different cell lines and different treatments all induce expression of inflammatory genes, some genes show more drastic response and other genes show more moderate or little response, suggesting that cellular context plays a critical role. To better focus our effort and resources on addressing the underlying molecular mechanism by which YTHDF2 regulates U6 decay and interaction with TLR3 in UV-induced inflammation,

for some cellular models, such as A431 and Hela cells, we elected to test the representative mRNAs that were robustly induced by YTHDF2 inhibition and/or UVB irradiation. Moreover, using multiple cell lines allows us to validate our findings in keratinocytes in other cell lines, including model cell lines such as HeLa cells, which we believe can increase the rigor of our findings and expand the potential biological relevance to different cell types. We have added some justifications in our revised manuscript.

“...an inflammatory gene robustly upregulated by UVB irradiation and/or YTHDF2 inhibition (Fig. 2I and 3B)...” (Page 11).

“...IL6 and TNF α , two inflammatory genes robustly upregulated by UVB irradiation and/or YTHDF2 inhibition (Fig. 2I and 3B)...” (Page 11).

4. Different durations of UVB exposure are used for different experiments with different cell lines – justification in the text would be helpful

Response: Thank you for the constructive comment. We have used different incubation durations post UVB exposure for different experiments with different cell lines, based on literature and our previous published reports as well as our pilot experiments. For example, we focused on 6 h or 24 h post-UVB irradiation for experiments focusing on analyzing expression of inflammatory genes. We focused on 1h post-UVB irradiation for YTHDF2 Co-IP mass spectrometry and YTHDF2 phosphorylation, as we focused on early response to UVB damage, which likely occurs as a direct response to UVB stress rather than a downstream response of the direct early response. We have added the following justification in the text in the Method section.

“Cells were irradiated with sham or UVB irradiation (20 mJ/cm² for HaCaT cells and 30 mJ/cm² for other cell lines that exhibit decreased sensitivity to UVB stress) and then collected for analysis at 6 h or 24 h for the delayed UVB stress response such as inflammatory gene expression and YTHDF2 protein down-regulation, or 1 h for the early or direct UVB stress response such as YTHDF2-interacting proteins and YTHDF2 phosphorylation, unless indicated otherwise.” (Page 29-30).

5. METTL16 section in introduction should refer to C. elegans METTL16 ortholog, and cite Mendel et al 2021

Response: Thank you for the constructive comment. We have added the findings in “*C. elegans* METTL16 ortholog” and cited Mendel et al 2021 in the Introduction for METTL16.

“The METTL16 ortholog METT-10 in *C. elegans* has been shown to deposit m⁶A on SAM synthase to inhibit its proper splicing (Mendel et al., 2021).” (Page 4).

a. This paper confirms mett-10 as the C. elegans, and also shows a slight increase in U6 snRNA levels upon mett-10 deletion, which is relevant to this study. It is interesting in this context, as there is no m6A reader protein (like YTHDF2) in worms.

Response: Thank you for the constructive comment. This is an important point. We are also intrigued to see that loss of mett-10 in *C. elegans* increases U6 snRNA levels, while there is no m⁶A reader protein like YTHDF2 in worms, as pointed out by the reviewer. There are three possibilities. (1) m⁶A methylation of U6 may also regulate U6 stability by affecting U6 uridylation, which can stabilize U6 snRNA. (2) m⁶A methylation of other mett-10 target genes may regulate either U6 snRNA transcription, maturation, or stability. (3) Worms may express other proteins that recognize m⁶A-methylated U6 snRNA and thus mediate U6 snRNA decay. Future investigation is required to test the role of these possibilities. We have added the following statements in the Discussion section.

“Intriguingly, previous studies have also shown that loss of *mett-10* in *C. elegans* increases U6 snRNA levels (Mendel et al., 2021), while there is no m⁶A reader protein like YTHDF2 in worms. There are three possibilities: (1) m⁶A methylation of U6 may affect other U6 snRNA modifications that in turn regulate U6 snRNA stability (Didychuk et al., 2018), (2) m⁶A methylation of other *mett-10* target genes may regulate U6 snRNA transcription, maturation, and/or stability, or (3) worms may express other proteins that recognize m⁶A-methylated U6 snRNA to mediate U6 snRNA decay. Future investigations are required to test these possibilities.” (Page 24-25).

b. In addition, Warda et al., 2017, is cited, but this paper showed no change to U6 snRNA levels upon METTL16 knockdown, in contrast to the result presented here. This should be mentioned.

Response: Thank you for the constructive comment. We agree. The report by Warda and colleagues did not detect a change to U6 snRNA levels upon METTL16 knockdown in HEK293 cells. These findings, together with our findings in several epithelial cells, and prior studies in worms, suggest that the regulation of U6 snRNA levels by METTL16 may be cell-type or organism specific, which warrants further investigation to elucidate the molecule basis for these different effects of METTL16 inhibition in different cell types and organisms. We have added the following statements in the Discussion section.

“In contrast, a previous report by Warda and colleagues did not detect a change to U6 snRNA levels upon METTL16 knockdown in HEK293 cells (Warda et al., 2017). These findings, together with our own, suggest that METTL16’s regulation of U6 snRNA levels may be context-dependent, warranting further investigation to elucidate the molecular basis for these differing effects across cell types and organisms.” (Page 25).

6. “In parallel, we also observed the U6-dependent YTHDF2-TLR3 interaction, which may be formed either as an intermediate to remove TLR3 from binding to m⁶A U6 or as an independent complex to maintain m⁶A U6-bound TLR3 in an inactive state.”

Response: Thank you for the constructive comment. We apologize for this error. We elected to remove the data related to this statement, as we reckon that these data may indicate another pathway related to YTHDF2/TLR3. However, we failed to remove this statement. In the revised manuscript, we have removed this statement.

a. There is no reference to a figure, but I assume this refers to figure 4L, which shows pull-down of TLR3 and YTHDF2 with m⁶A U6, but not necessarily a YTHDF2-TLR3 interaction

Response: Thank you for the constructive comment. We apologize for this error. We elected to remove the data related to this statement, as we reckon that these data may indicate another pathway related to YTHDF2/TLR3. However, we failed to remove this statement. In the revised manuscript, we have removed this statement.

7. The authors present evidence that degradation of the U6 snRNA may be mediated by the CCR4-NOT deadenylase component CNOT1, but this is limited to interaction between CNOT1 and U6 snRNA by RIP and increased stability of U6 snRNA upon CNOT1 knockdown in a single cell line (HeLa).

Response: Thank you for the constructive comment. We have performed a new experiment in A431 cells. Our new data showed that CNOT1 knockdown increased U6 snRNA stability in A431 cells as well. We have added the new data (Fig. 6H).

“... CNOT1 knockdown inhibits U6 decay in both HeLa and A431 cells (Fig. 6G-6H, and Supplementary Fig. S15I-S15J).” (Page 19).

a. This is surprising, as U6 is not polyadenylated. In the discussion, the authors point out that the 3' tail of U6 can be adenylated, but as far as I know this adenylation has not been implicated in U6 decay. Perhaps the authors could include some more thoughts on how the mRNA deadenylase complex could regulate U6 snRNA decay.

Response: Thank you for the constructive comment. We agree with the reviewer that U6 is not polyadenylated. Interestingly previous studies have shown that YTHDF2 mediates polyA-containing mRNAs through binding with CCR4-NOT1 (Du et al., 2016). Our original data (Fig. 2, 5F, 5G, 5H) showed that both YTHDF2 and CNOT1 bind to U6 snRNA and are critical for U6 snRNA decay. To address this important question and further elucidate the mechanism of YTHDF2-mediated U6 decay, we performed new experiments to determine the role of YTHDF2 and METTL16-mediated m⁶A methylation in CNOT1 binding to U6 snRNA. Our new data showed that either YTHDF2 deletion or METTL16 knockdown drastically reduces CNOT1 binding to U6 (Fig. 6I, 6J), supporting a critical role of YTHDF2 and U6 m⁶A methylation in CNOT1 binding to U6 snRNA. Our new data demonstrate that CNOT1 binds to U6 snRNA, through interacting with the YTHDF2-m⁶A U6 complex. The role of U6 adenylation cannot be excluded and warrants future investigations. We have added the new data and the following statements in the Results and Discussion section.

“To elucidate the mechanism of YTHDF2-mediated U6 decay, we examined the role of YTHDF2 and METTL16-mediated m⁶A methylation in CNOT1 binding to U6 snRNA. We found that either YTHDF2 deletion or METTL16 knockdown drastically reduces CNOT1 binding to U6 in HeLa cells (Fig. 6I-6J and Supplementary Fig. S15N-S15O), supporting a critical role of YTHDF2 and U6 m⁶A methylation in CNOT1 binding to U6 snRNA. To determine whether UVB irradiation affects CNOT1 expression, we assessed the effect of UVB irradiation on CNOT1 protein levels. We found that UVB irradiation had no effect on CNOT1 protein expression in either HeLa or HaCaT cells (Supplementary Fig. S15P). These results demonstrate that CNOT1 binds to U6 snRNA and mediates U6 decay through interacting with YTHDF2 in an m⁶A-methylation dependent manner.” (Page 19).

“Previously, the 3' tail of a small fraction of U6 snRNA has been shown to be adenylated (Chen et al., 2000). Although U6 is not known to be polyadenylated, recent studies have detected m⁶A methylated U6 snRNA in poly(A)⁺ RNA in worms (Mendel et al., 2021). While this may represent remnants left after poly(A)⁺ enrichment from total RNA, it is also possible that a fraction of U6 snRNA is polyadenylated in worms and other organisms, which could regulate U6 decay upon m⁶A methylation similar to poly(A)⁺ mRNAs via the YTHDF2-CNOT1 interaction (Du et al., 2016). It is possible that adenylation and/or polyadenylation cooperates with m⁶A methylation of U6 snRNA to regulate U6 snRNA turnover. Future investigation is warranted to elucidate the detailed mechanism by which YTHDF2 and CNOT1 regulate U6 snRNA decay.” (Page 24).

Specific comments

Page 6 – Acronym SLE is not explained in the text

Response: Thank you for the constructive comment. We have added explanation for the acronym SLE in the revised text.

“... YTHDF2 expression was decreased in systemic lupus erythematosus (SLE)...” (Page 6).

Page 8 – “*YTHDF2 knockdown slightly increased m6A enrichment in the 5'UTR and 3'UTR...*” Fig. 2C shows no change (-0.03%) in proportion of m6A in 3'UTR and extremely small (+0.08%) increase in 5'UTR, compared with a 0.4% decrease in the CDS upon shDF2

Response: Thank you for the constructive comment. We agree. YTHDF2 knockdown had little effect on m⁶A enrichment across the gene regions, particularly in Fig. 2C. We have corrected these statements for Fig. 2B-C.

“YTHDF2 knockdown had little effect on m⁶A enrichment in the 5'UTR, 3'UTR, CDS, or the total peak distribution (Fig. 2B-C).” (Page 8).

Page 10 and Fig. 3F – “*While both U6 and m6A U6 induced IL-6 expression, U6 lead to a higher induction than m6A U6 (Fig. 3F).*” Fig. 3F shows non-significant change to IL-6 levels with m6A U6

Response: Thank you for the constructive comment. We recognize that Fig. 3F is very busy and the difference between U6 and m⁶A U6 in inducing IL-6 expression was buried by many comparisons. We have re-prepared the figure panel such that U6 and m⁶A U6 are next to each other. We also reduced comparisons, so the key comparison and differences can be highlighted in Fig. 3F. Indeed, we observed that U6 lead to a higher induction of IL-6 expression than m⁶A U6 in WT cells but not DF2 KO cells (Fig. 3F). We have revised the statement, so it reads as follows.

“In WT HeLa cells, while both U6 and m⁶A U6 induced IL-6 expression, an inflammatory gene robustly upregulated by UVB irradiation and/or YTHDF2 inhibition (Fig. 2I and 3B), U6 lead to a higher induction than m⁶A U6 (Fig. 3F).” (Page 11).

Page 11 – “*Indeed, a small-scale siRNA screening for RNA sensors, including TLR3, TLR7, TLR8, MDA5, and RIG-I (Akira et al., 2006; Fitzgerald and Kagan, 2020)...*” This reads as if the cited publications contain the small-scale siRNA screen

Response: Thank you for the constructive comment. We have revised the statement, so it reads as “...a small-scale siRNA screening for TLR3, TLR7, TLR8, MDA5, and RIG-I, which are all known sensors for RNA (Akira et al., 2006; Fitzgerald and Kagan, 2020), showed that knockdown...” (Page 12).

Page 12 – *Many references to U6 snRNA “binding to” proteins (TLR3, YTHDF2) rather than being bound by them*

Response: Thank you for the constructive comment. We have revised these phrases with “...as well as YTHDF2 binding to the mRNAs...” (Page 8) and “...binds with” (Page 14).

Page 13 – “*we assessed whether U6 and YTHDF2 also localize in endosomes...*” – this question is asked but not answered, with the next relevant sentence being “*Second, UVB irradiation increased both total level and endosomal proportion of U6 snRNA...*”

Response: Thank you for the constructive comment. We have revised our statement, so it reads as “...we assessed whether U6 and YTHDF2 also localize in the cytoplasm, particularly in endosomes...” (Page 15).

Page 13 – “*and confocal imaging analysis showed that YTHDF2 is also localized in endosomes (Fig. 5C-*

E)” *It would be useful to mention the Rab7 endosomal marker in the text*

Response: Thank you for the constructive comment. We have added Rab7 endosomal marker in the text. “...as shown by the colocalization of YTHDF2 with the late endosome marker Rab7...” (Page 16).

Page 18 – *“In addition, UVB irradiation inhibits YTHDF2 phosphorylation at S39, which is critical for YTHDF2 interaction with CNOT1 and localization in endosomes and is inhibited by UVB irradiation.”*
Redundancy – inhibition by UVB irradiation written twice.

Response: Thank you for the constructive comment. We have removed the redundancy.

Now the statement reads as “In addition, UVB irradiation inhibits YTHDF2 phosphorylation at S39, which is critical for YTHDF2 interaction with CNOT1 and localization in endosomes” (Page 23).

Page 19 – *“In addition, the interaction between YTHDF2 and U6 snRNPs...”* *Should be snRNP proteins or components*

Response: Thank you for the constructive comment. We have changed this statement to “In addition, the interaction between YTHDF2 and U6 snRNP proteins...” (Page 24).

Fig. 3M – *“However, dual knockdown of both METTL16 and YTHDF2 mimicked the effect of singular YTHDF2 knockdown (Fig. 3M-Q).”* *Panel M is COX-2 immunoblot and does not include YTHDF2 knockdown. Fig. 3M is not referenced elsewhere in text*

Response: Thank you for the constructive comment. We have added a separate statement and referenced Fig. 3M in the revised text.

“METTL16 knockdown increased UVB-induced COX-2 expression (Fig. 3M).” (Page 12).

Fig. 6F – *Label error, “CONT1” instead of CNOT1*

Response: Thank you for the constructive comment. We have corrected this error (Fig. 6F).

Fig. S16G – *No mention of scoring system (3,2,1,0) for YTHDF2 levels in figure or legend*

Response: Thank you for the constructive comment. We have added the scoring system in the figure legend (Fig. S16G) in the Supplemental Information.

“The staining intensity was scored as 3 (strong), 2 (medium), 1 (weak), and 0 (negative).” (Supple Information, Fig. S16G).

References

- Akira, S., Uematsu, S., and Takeuchi, O. (2006). Pathogen recognition and innate immunity. *Cell* 124, 783-801.
- Bensaude, O. (2011). Inhibiting eukaryotic transcription: Which compound to choose? How to evaluate its activity? *Transcription* 2, 103-108.
- Bowden, G.T. (2004). Prevention of non-melanoma skin cancer by targeting ultraviolet-B-light signalling. *Nat Rev Cancer* 4, 23-35.
- Buckman, S.Y., Gresham, A., Hale, P., Hruza, G., Anast, J., Masferrer, J., and Pentland, A.P. (1998). COX-2 expression is induced by UVB exposure in human skin: implications for the development of skin cancer. *Carcinogenesis* 19, 723-729.
- Chen, Y., Sinha, K., Perumal, K., and Reddy, R. (2000). Effect of 3' terminal adenylic acid residue on the uridylation of human small RNAs in vitro and in frog oocytes. *RNA* 6, 1277-1288.
- Didychuk, A.L., Butcher, S.E., and Brow, D.A. (2018). The life of U6 small nuclear RNA, from cradle to grave. *RNA* 24, 437-460.
- Du, H., Zhao, Y., He, J., Zhang, Y., Xi, H., Liu, M., Ma, J., and Wu, L. (2016). YTHDF2 destabilizes m(6)A-containing RNA through direct recruitment of the CCR4-NOT deadenylase complex. *Nat Commun* 7, 12626.
- Fitzgerald, K.A., and Kagan, J.C. (2020). Toll-like Receptors and the Control of Immunity. *Cell* 180, 1044-1066.
- Gao, Y., Vasic, R., Song, Y., Teng, R., Liu, C., Gbyli, R., Biancon, G., Nelakanti, R., Lobben, K., Kudo, E., *et al.* (2020). m(6)A Modification Prevents Formation of Endogenous Double-Stranded RNAs and Deleterious Innate Immune Responses during Hematopoietic Development. *Immunity* 52, 1007-1021 e1008.
- Guirguis, A.A., Ofir-Rosenfeld, Y., Knezevic, K., Blackaby, W., Hardick, D., Chan, Y.C., Motazedian, A., Gillespie, A., Vassiliadis, D., Lam, E.Y.N., *et al.* (2023). Inhibition of METTL3 Results in a Cell-Intrinsic Interferon Response That Enhances Antitumor Immunity. *Cancer Discov* 13, 2228-2247.
- Hu, X., Li, Z., Ding, Y., Geng, Q., Xiahou, Z., Ru, H., Dong, M.Q., Xu, X., and Li, J. (2018). Chk1 modulates the interaction between myosin phosphatase targeting protein 1 (MYPT1) and protein phosphatase 1beta (PP1cbeta). *Cell Cycle* 17, 421-427.
- Jiao, J., Mikulec, C., Ishikawa, T.O., Magyar, C., Dumlao, D.S., Dennis, E.A., Fischer, S.M., and Herschman, H. (2014). Cell-type-specific roles for COX-2 in UVB-induced skin cancer. *Carcinogenesis*.
- Kim, Y., and He, Y.Y. (2014). Ultraviolet radiation-induced non-melanoma skin cancer: Regulation of DNA damage repair and inflammation. *Genes Dis* 1, 188-198.
- Liu, Q., Guntuku, S., Cui, X.S., Matsuoka, S., Cortez, D., Tamai, K., Luo, G., Carattini-Rivera, S., DeMayo, F., Bradley, A., *et al.* (2000). Chk1 is an essential kinase that is regulated by Atr and required for the G(2)/M DNA damage checkpoint. *Genes Dev* 14, 1448-1459.
- Mendel, M., Delaney, K., Pandey, R.R., Chen, K.M., Wenda, J.M., Vagbo, C.B., Steiner, F.A., Homolka, D., and Pillai, R.S. (2021). Splice site m(6)A methylation prevents binding of U2AF35 to inhibit RNA splicing. *Cell* 184, 3125-3142 e3125.

Qiang, L., Sample, A., Shea, C.R., Soltani, K., Macleod, K.F., and He, Y.Y. (2017). Autophagy gene Atg7 regulates ultraviolet radiation-induced inflammation and skin tumorigenesis. *Autophagy* 13, 2086-2103.

Sauterer, R.A., Feeney, R.J., and Zieve, G.W. (1988). Cytoplasmic assembly of snRNP particles from stored proteins and newly transcribed snRNA's in L929 mouse fibroblasts. *Exp Cell Res* 176, 344-359.

Terns, M.P., Dahlberg, J.E., and Lund, E. (1993). Multiple cis-acting signals for export of pre-U1 snRNA from the nucleus. *Genes Dev* 7, 1898-1908.

Warda, A.S., Kretschmer, J., Hackert, P., Lenz, C., Urlaub, H., Hobartner, C., Sloan, K.E., and Bohnsack, M.T. (2017). Human METTL16 is a N(6)-methyladenosine (m(6)A) methyltransferase that targets pre-mRNAs and various non-coding RNAs. *EMBO Rep* 18, 2004-2014.

REVIEWER COMMENTS

Reviewer #1 (Remarks to the Author):

The revised manuscript is strengthened with new data, analysis and clarification. Here are few remaining issues that should be addressed.

Response: We thank you for taking the time to review our manuscript and providing us with insightful follow-up comments, all of which have been considered carefully in the preparation of this revision (please see our point-by-point responses below). All revisions have been highlighted with red text and submitted as supplementary information. To specifically address these comments, we have performed several new experiments and added in new data (shown in the new Fig. S6D, Fig. 5S-with more replicates added, Fig. S14G, Fig. S14P, and Fig. S14Q). To improve the focus of our manuscript, we elected to remove the original Fig. S6D and Fig. S15F-G. With these new data added, changes made, and new discussion added, we believe our revised manuscript has significantly improved with all the reviewers' suggestions.

1. It is a bit odd that they chose to quantify Cox2 mRNA in DF2 Het but not KO in Fig. S6D. Typically, primary keratinocytes are derived from neonatal pups, which take a few weeks to produce from Het breeding. Nevertheless, Cox2 mRNA changes are mild. Do they observe epidermal hyperproliferation in DF2 Het without or with UVB radiation? These results may add the validity for using Het for quantification. Alternatively, they could do Cox2 in situ or IF staining to demonstrate Cox2 increase upon DF2 KO in the mouse model.

Response: Thank you for your constructive comments. We agree. The change in COX-2 mRNA is modest, as YTHDF2 level is only partially reduced. Therefore, we did not continue to characterize the histological changes in the WT and DF2 cHet mice. As suggested by the reviewer, we have performed new experiments to determine COX-2 expression using immunofluorescence (IF) analysis in WT and DF2 cKO mice treated with sham or UVB irradiation as in the original Fig. 1C. The new analysis showed that COX-2 protein expression is increased by either UVB irradiation or YTHDF2 deletion alone in mouse skin and further increased by the combination of UVB irradiation and YTHDF2 deletion, supporting our findings in human keratinocytes. We did note that the effect of YTHDF2 deletion on COX-2 expression is modest. Future studies are needed to characterize the temporal regulation of COX-2 expression by YTHDF2 in mouse skin irradiated with different doses of UVB irradiation, acute UVB irradiation, and different repeated UVB irradiation, at different time points. To improve the focus of our manuscript, we elected to remove the original Fig. S6D using WT and YTHDF2 cHet primary mouse keratinocytes.

We have added the following statement in the Results section.

“Furthermore, we found that skin-specific YTHDF2 deletion increased COX-2 expression upon sham or UVB irradiation (Supplementary Fig. S6D), supporting our findings in human cells that YTHDF2 loss enhances UVB-induced inflammatory gene expression.” (Page 7).

2. In new Fig. S14G, they performed U6 in situ and Rab7 staining to corroborate cell culture studies. The fluorescent signals appear to be over-exposed for U6 in DF2 cKO and Rab7 in both sham and UVB conditions. They should also provide higher magnification images, comparable to their cell culture results e.g. S14H, to convincingly demonstrate co-localization of U6 and Rab7 in endosomes upon UVB radiation.

Response: Thank you for your insightful comments. We have provided higher magnification images for Fig. S14G. These higher magnification images did show colocalization of U6 and Rab7, which is most visible in the DF2 cKO mouse skin exposed to UVB irradiation.

Reviewer #2 (Remarks to the Author):

Overall, the authors addressed our comments thoughtfully. We have two remaining questions on comments #5 and #7. We request more careful interpretation of data, and make suggestions on how to make more solid and clear conclusions.

Response: We thank you for taking the time to review our manuscript and providing us with insightful follow-up comments, all of which have been considered carefully in the preparation of this revision (please see our point-by-point responses below). All revisions have been highlighted with red text and submitted as supplementary information. To specifically address these comments, we have performed several new experiments and added in new data (shown in the new Fig. S6D, Fig. 5S-with more replicates added, Fig. S14G, Fig. S14P, and Fig. S14Q). To improve the focus of our manuscript, we elected to remove the original Fig. S6D and Fig. S15F-G. With these new data added, changes made, and new discussion added, we believe our revised manuscript has significantly improved with all the reviewers' suggestions.

Comment #5:

The role of MYPT1 is confusing. I think the model is that UVB-irradiated YTHDF2 interacts with MYPT1, which dephosphorylates YTHDF2 to prevent it from binding CNOT1. To prove MYPT1 plays a role in regulating U6 snRNA stability, the authors need to downregulate or overexpress MYPT1 and evaluating its effect on U6 snRNA levels and stability. Otherwise, the interpretation of MYPT1's role should be toned down.

Response: Thank you for the constructive comment. To determine whether MYPT1 regulates U6 snRNA stability, we assessed the effect of MYPT1 knockdown. Our new data showed that MYPT1 knockdown increased U6 snRNA levels and stability (Fig. S15F-S15G), supporting a role of MYPT1 in regulating U6 snRNA decay.

Reviewer response to rebuttal: In figures S15F and S15G, phenotypes are very weak, and only shown in HeLa cells. Additionally, as pointed out by reviewer #1 it does concern me that authors interchangeably use so many cell lines. The experiment conducted in S15F and S15G, should be repeated in other cell lines (e.g., A431) before authors conclude that MYPT1 knockdown increased U6 snRNA levels and stability, especially since the phenotype in HeLa cells is very weak.

Response: Thank you for your constructive comments. During the project, we faced a number of technical challenges working with the HaCaT keratinocyte cell line, a difficult cell line to transfect for plasmids and to obtain large amounts of samples, which limits our ability to assess the molecular mechanisms by which YTHDF2 regulates inflammatory responses. Therefore, we tested and used several model cell lines, including HeLa, 293T, and A431 cells. In addition, we would also like to validate whether our findings in keratinocytes are specific for keratinocytes or are also observed in model cell lines, so we can demonstrate whether our findings on YTHDF2 regulation of inflammatory response is more broadly implicated. To address the question on the role of MYPT1 in regulating U6 metabolism in additional cell lines, we have

performed new experiments using both A431 and HaCaT cells. Our original data in Fig. S15F-G showed that MYPT1 knockdown slightly increased U6 snRNA stability. However, different from HeLa cells, in A431 and HaCaT cells, we found that MYPT1 knockdown slightly decreased U6 snRNA stability (See Figure A and B, C is the original Fig. S15F-G), consistent with our proposed model that MYPT1 regulates YTHDF2 dephosphorylation and YTHDF2 phosphorylation regulates the YTHDF2-CNOT1 interaction and thus U6 decay. In addition, these data together suggest a cell-type-dependent role of MYPT1 in regulating U6 snRNA stability. It is possible that in addition to YTHDF2 phosphorylation at serine 39, MYPT1 may have other downstream protein targets, some of which may regulate the stability or transcription of the U6 snRNA independent of YTHDF2. Moreover, the specific downstream targets of MYPT1 may be specific for different cell types, which may utilize distinct pathways to regulate U6 expression and metabolism.

A-C. qPCR analysis of U6 snRNA stability and the level of U6 snRNA and MYPT1 in A431 cells (A, new data), and HaCaT cells (B, new data) and HeLa cells (C, original Fig. S15F-G) with or without MYPT1 knockdown.

To improve the focus of our manuscript, we elected to leave out the original Fig. S15F-G and not to include our new data in A431 and HaCaT cells. We are currently investigating the detailed mechanism by which U6 metabolism is regulated by different molecular pathways as an independent project. In addition, we have tone down the interpretation of the role of MYPT1 by removing MYPT1 in the schematic summary of our proposed model and revised our statements in the Results and Discussion section.

“...suggesting a potential role of the MYPT1–phosphatase complex in dephosphorylating YTHDF2 at S39.” (Page 18-19).

“Our study also suggests that UVB might reduce YTHDF2 phosphorylation at S39 by promoting the interaction between YTHDF2 and MYPT1, which could be involved in the autophagic degradation of YTHDF2. Previous studies have shown that Chk1, a kinase activated by DNA damage and UV radiation (Liu et al., 2000), binds and phosphorylates MYPT1, leading to the recruitment of protein phosphatase 1β (PP1cβ) to dephosphorylate its substrate (Hu et al., 2018). It remains to be investigated whether UV radiation-induced Chk1 activation mediates MYPT1 phosphorylation, thus leading to increased MYPT1 binding to YTHDF2.” (Page 28).

However, we are open to additional suggestions from the reviewer and will be happy to make additional revisions if needed.

Comment #7:

In Figure 5, the authors show that YTHDF2 and U6 snRNA enter endosomes. However, the functional significance of this as it pertains to their proposed model is unclear (especially with regards to YTHDF2). The authors should confirm that U6 snRNA entry into the endosome is required for upregulation of inflammatory genes and whether or not YTHDF2 entry into the endosome affects inflammatory gene expression.

Response: Thank you for the constructive comment. This is a great question and suggestion. To determine the importance of U6 snRNA entry into the endosome in the increased expression of inflammatory genes, we assessed the effect of Dynasore, which inhibits U6 entry into endosomes (Fig. 5G), and the role of the interaction U6 with TLR3, an endosomal RNA sensor. Our new data showed that Dynasore reduced expression of inflammatory genes induced by U6 or YTHDF2 knockdown (Fig. 5R and 5S). To further determine whether the effect of YTHDF2 inhibition on proinflammatory response is mediated by U6/TLR3, we assessed the effect of U6 knockdown, TLR3 knockdown, and the combination. Our new data showed that either U6 knockdown or TLR3 knockdown decreased IL6 expression and that the combination of double knockdown of TLR3 and U6 has similar effect to knockdown of either U6 or TLR3 singularly (Fig. 4H-4J), suggesting that the effect of U6 is mediated by the TLR3 pathway. Together with our findings that YTHDF2 endosomal entry is dependent on U6, while U6 endosomal entry is independent of YTHDF2 (Fig. 5M-O), our data demonstrate the U6 entry into endosomes is critical for upregulation of inflammatory genes.

Reviewer response to rebuttal: In Figure 5S, dynasore treatment seemed to have reduced expression of inflammatory genes in both control knockdown and DF2 knockdown cells. This suggest that U6 may enter endosomes to induce inflammation independently of DF2 knockdown. Hence, the results may not well support the author's model that DF2 knockdown leads to enhanced U6 entry into endosomes. For fig. 5S, more replicates are need and statistical analysis between the black bars (vehicle vs. dynasore) are also needed for more conclusive interpretation. Also repeating the 5S assay in other cell lines (Hela, HaCat) will also help. Or were the assays not tried in other cell lines because DF2 knockdown in only A431 lead to inflammation, and not in other cell types? Such clarification will help readers assess the significance of the finding, and the rationale for switching between cell lines.

Response: Thank you for your insightful comments and suggestions. We have performed new experiments and added clarifications for our model. We have added three biological replicates for Fig. 5S and added the statistical analysis for vehicle vs Dynasore. In addition, we have performed new experiments in both HaCaT and Hela cells following exposure to UVB or sham irradiation and observed similar effect of Dynasore in UVB irradiated cells (Fig. S14P-Q). We agree with the reviewer. Dynasore seems to reduce inflammatory gene expression in YTHDF2-expressed cells as well (Fig. 5S), suggesting YTHDF2-independent effect. We think that such effect may be due to the reduction of non-m⁶A modified U6 entry into endosomes and subsequent TLR3 interaction and activation, which is YTHDF2-independent. Future investigation is needed to identify the molecular details and dynamics of U6 metabolism and function.

We have added the following statements in the Results and Discussion section.

“... or by YTHDF2 knockdown in HaCaT and HeLa cells followed by UVB irradiation (Fig. S14P-S14Q). In control cells with YTHDF2 expression, Dynasore also slightly reduced inflammatory gene expression in A431 cells and IL-6 expression in UVB-irradiated HaCaT cells, but not in UVB-irradiated HeLa cells (Fig. 5S and Fig. S14P-S14Q), suggesting a cell-type-dependent, YTHDF2-independent effect of Dynasore. Such effect of Dynasore may be due to the reduction of endosomal localization of non-m⁶A-modified U6, which can be YTHDF2-independent, as non-m⁶A-modified U6 is bound by TLR3 but not by YTHDF2 (Fig. 4K and 4O). Future investigation is needed to elucidate the specific molecular mechanism for regulating U6 snRNA metabolism, localization, and function.” (Page 17-18).

“Under baseline conditions, METTL16 knockdown increases both total and endosomal U6 abundance and cytokine expression in METTL16-low A431 cells (Fig. 4N), suggesting that non-m⁶A-modified U6 snRNA, which is not bound by YTHDF2, can also enter endosomes and be bound by TLR3 in a YTHDF2-independent manner.” (Page 26).

Reviewer #3 (Remarks to the Author):

I co-reviewed this manuscript with one of the reviewers who provided the listed reports. This is part of the Nature Communications initiative to facilitate training in peer review and to provide appropriate recognition for Early Career

Researchers who co-review manuscripts.

Response: Thank you.

Reviewer #4 (Remarks to the Author):

I commend the authors for the revised version and I am happy to note the improvements made.

Response: Thank you.

References

Hu, X., Li, Z., Ding, Y., Geng, Q., Xiahou, Z., Ru, H., Dong, M.Q., Xu, X., and Li, J. (2018). Chk1 modulates the interaction between myosin phosphatase targeting protein 1 (MYPT1) and protein phosphatase 1beta (PP1cbeta). *Cell Cycle* 17, 421-427.

Liu, Q., Guntuku, S., Cui, X.S., Matsuoka, S., Cortez, D., Tamai, K., Luo, G., Carattini-Rivera, S., DeMayo, F., Bradley, A., *et al.* (2000). Chk1 is an essential kinase that is regulated by Atr and required for the G(2)/M DNA damage checkpoint. *Genes Dev* 14, 1448-1459.

Yang et al NCOMMS-24-83544

The authors describe truly novel findings. The RNA modification m6A is shown to be present on a variety of RNA molecules. The m6A reader protein YTHDF2 is shown to bind modified mRNAs and mediate their decay via recruitment of the deadenylase CCR4-NOT1 complex. In this study, the authors demonstrate that YTHDF2 binds the spliceosomal U6 snRNA. This is new, as such an interaction was never demonstrated before. They also show that this interaction regulates RNA levels of U6 snRNA where YTHDF2 interacts with deadenylase component CNOT1, which may mediate decay of m6A-modified U6 snRNA. The authors describe the role of m6A reader YTHDF2 in the inflammatory response through its regulation of the U6 snRNA. They provide data from several different cell lines and a skin-specific conditional deletion mouse model. The study provides evidence that human YTHDF2 binds to and regulates the stability of U6, modulating the expression of inflammatory regulators via TLR3. Finally, they show that UVB irradiation leads to autophagy-mediated downregulation of YTHDF2, and that deletion of YTHDF2 in mouse skin cells increases sensitivity to UVB radiation. Overall, the findings are novel, unexpected and relevant for the RNA modification field.

General comments

1. Analysis of gene expression changes and changes in m6A (specifically changes in inflammatory genes, changes in U6 levels, U6 m6A status etc.) from skin of cKO YTHDF2 mice would have added a lot to the hypothesis that YTHDF2 regulation of U6 is responsible for histological observations
2. In the text, it is often not stated which cell line, or even system, is being referred to or studied
 - a. E.g. in the abstract the authors do not state that they studied YTHDF2 in human cells, and “skin-specific deletion” does not mention the mouse model
 - b. Several different cell lines (HaCaT, HeLa, A431, 293T, NHEK, CHL-1) are used in the study, but the text rarely states which result was obtained from which cell line, or the rationale for doing particular experiments in a particular cell line
 - i. **Text should refer to cell lines, e.g. “Our RNA-seq analysis showed that YTHDF2 knockdown...” should add “in HaCaT cells”**
 - ii. Figure panels should also show clearly which cell line the result comes from. This is done for some panels (Fig 5C, 5G etc.), but most are unlabelled
 - iii. Some figure legends do not mention cell line used
 1. **E.g. Fig. 5D. Panel includes label 293T, but not stated in figure legend – “D. Endo-IP analysis of YTHDF2 protein levels in endosomes.”**
 2. **Fig. 5K. No label in panel, and no mention of cell line in figure legend. Also error in figure legend “K. Immunoblot analysis of YTHDF2 in whole cell lysates (WCL) and endosomes in cells as in K.”**
 3. **Fig. 4M. No label in panel, no cell line mentioned in legend.**

3. Most figures contain many panels showing qPCR results for different mRNAs in different conditions. It is sometimes unclear why certain mRNAs were or were not tested under specific conditions, and in the different cell lines used. Clearer labelling in figures and some explanation of the rationale in the text would be useful.
4. Different durations of UVB exposure are used for different experiments with different cell lines – justification in the text would be helpful
5. METTL16 section in introduction should refer to *C. elegans* METTL16 ortholog, and cite Mendel et al 2021
 - a. This paper confirms *mett-10* as the *C. elegans*, and also shows a slight increase in U6 snRNA levels upon *mett-10* deletion, which is relevant to this study. It is interesting in this context, as there is no m6A reader protein (like YTHDF2) in worms.
 - b. In addition, Warda *et al.*, 2017, is cited, but this paper showed no change to U6 snRNA levels upon METTL16 knockdown, in contrast to the result presented here. This should be mentioned.
6. *“In parallel, we also observed the U6-dependent YTHDF2-TLR3 interaction, which may be formed either as an intermediate to remove TLR3 from binding to m6A U6 or as an independent complex to maintain m6A U6-bound TLR3 in an inactive state.”*
 - a. There is no reference to a figure, but I assume this refers to figure 4L, which shows pull-down of TLR3 and YTHDF2 with m6A U6, but not necessarily a YTHDF2-TLR3 interaction
7. The authors present evidence that degradation of the U6 snRNA may be mediated by the CCR4-NOT deadenylase component CNOT1, but this is limited to interaction between CNOT1 and U6 snRNA by RIP and increased stability of U6 snRNA upon CNOT1 knockdown in a single cell line (HeLa).
 - a. This is surprising, as U6 is not polyadenylated. In the discussion, the authors point out that the 3' tail of U6 can be adenylated, but as far as I know this adenylation has not been implicated in U6 decay. Perhaps the authors could include some more thoughts on how the mRNA deadenylase complex could regulate U6 snRNA decay.

Specific comments

Page 6 – Acronym SLE is not explained in the text

Page 8 – *“YTHDF2 knockdown slightly increased m6A enrichment in the 5'UTR and 3'UTR...”* **Fig. 2C shows no change (-0.03%) in proportion of m6A in 3'UTR and extremely small (+0.08%) increase in 5'UTR, compared with a 0.4% decrease in the CDS upon shDF2**

Page 10 and Fig. 3F – *“While both U6 and m6A U6 induced IL-6 expression, U6 lead to a higher induction than m6A U6 (Fig. 3F).”* **Fig. 3F shows non-significant change to IL-6 levels with m6A U6**

Page 11 – *“Indeed, a small-scale siRNA screening for RNA sensors, including TLR3, TLR7, TLR8, MDA5, and RIG-I (Akira et al., 2006; Fitzgerald and Kagan, 2020)...”* **This reads as if the cited publications contain the small-scale siRNA screen**

Page 12 – Many references to U6 snRNA “binding to” proteins (TLR3, YTHDF2) rather than being bound by them

Page 13 – “we assessed whether U6 and YTHDF2 also localize in endosomes...” – **this question is asked but not answered, with the next relevant sentence being** “Second, UVB irradiation increased both total level and endosomal proportion of U6 snRNA...”

Page 13 – “and confocal imaging analysis showed that YTHDF2 is also localized in endosomes (Fig. 5C-E)” **It would be useful to mention the Rab7 endosomal marker in the text**

Page 18 – “In addition, **UVB irradiation inhibits** YTHDF2 phosphorylation at S39, which is critical for YTHDF2 interaction with CNOT1 and localization in endosomes and is **inhibited by UVB irradiation.**” **Redundancy – inhibition by UVB irradiation written twice.**

Page 19 – “In addition, the interaction between YTHDF2 and U6 snRNPs...” **Should be snRNP proteins or components**

Fig. 3M – “However, dual knockdown of both METTL16 and YTHDF2 mimicked the effect of singular YTHDF2 knockdown (Fig. 3M-Q).” **Panel M is COX-2 immunoblot and does not include YTHDF2 knockdown. Fig. 3M is not referenced elsewhere in text**

Fig. 6F – **Label error, “CONT1” instead of CNOT1**

Fig. S16G – No mention of scoring system (3,2,1,0) for YTHDF2 levels in figure or legend